# Optimal Exploration for Model-Based RL in Nonlinear Systems

**Andrew Wagenmaker**[*]
Paul G. Allen School of Computer Science & Engineering
University of Washington
Seattle, WA 98195

**Guanya Shi**[†]
Robotics Institute
Carnegie Mellon University
Pittsburgh, PA 15213

**Kevin Jamieson**[‡]
Paul G. Allen School of Computer Science & Engineering
University of Washington
Seattle, WA 98195

## Abstract

Learning to control unknown nonlinear dynamical systems is a fundamental problem in reinforcement learning and control theory. A commonly applied approach is to first explore the environment (exploration), learn an accurate model of it (system identification), and then compute an optimal controller with the minimum cost on this estimated system (policy optimization). While existing work has shown that it is possible to learn a uniformly good model of the system [1], in practice, if we aim to learn a good controller with a low cost on the actual system, certain system parameters may be significantly more critical than others, and we therefore ought to focus our exploration on learning such parameters.

In this work, we consider the setting of nonlinear dynamical systems and seek to formally quantify, in such settings, (a) which parameters are most relevant to learning a good controller, and (b) how we can best explore so as to minimize uncertainty in such parameters. Inspired by recent work in linear systems [2], we show that minimizing the controller loss in nonlinear systems translates to estimating the system parameters in a particular, task-dependent metric. Motivated by this, we develop an algorithm able to efficiently explore the system to reduce uncertainty in this metric, and prove a lower bound showing that our approach learns a controller at a near-instance-optimal rate. Our algorithm relies on a general reduction from policy optimization to optimal experiment design in arbitrary systems, and may be of independent interest. We conclude with experiments demonstrating the effectiveness of our method in realistic nonlinear robotic systems[1].

## 1 Introduction

Controlling nonlinear dynamical systems is a core problem in robotics, cyber-physical systems, and beyond, and a significant body of work in both the control theory and reinforcement learning communities has sought to address this challenge [3–5]. In many real-world scenarios [6–9], the dynamics of the system of interest is unknown, or only a coarse model of them is available, which

---

[*]ajwagen@cs.washington.edu
[†]guanyas@andrew.cmu.edu
[‡]jamieson@cs.washington.edu

[1]Code: https://github.com/ajwagen/nonlinear_sysid_for_control

37th Conference on Neural Information Processing Systems (NeurIPS 2023).

significantly increases the challenge of control—not only must we control such systems, we must *learn* to control them. While a variety of methods exist to address this challenge, a commonly applied approach is to first perform *system identification*, learning an accurate model of the system's dynamics, and then use this model to obtain a controller. Despite its promising potential, there are still several fundamental questions that must be answered to make this approach practically effective.

***Which parameters are most relevant to learning a good controller?*** Beyond some special cases, little work has been done characterizing how the estimation error from system identification translates to end-to-end suboptimality in the resulting controller of our nonlinear systems. In particular, certain parameters of the system or regions of the state space may be irrelevant to learning a good controller, and coarse estimates of these parameters would suffice, while other parameters may be critical to learning a good controller, and we must therefore estimate these parameters very accurately in order to effectively control the system. In the context of this work, where nonlinearities are considered, the heterogeneity of the parameters is further accentuated. For instance, around a point of equilibrium, some system parameters might be completely inactive, having no impact on the dynamics (see the example in Section 1.1 for an illustration of this).

***How can we best explore so as to minimize uncertainty in relevant parameters?*** Even if we are able to determine which parameters are most important for obtaining a good controller on the true system, it is not obvious how to use this information. How can we direct our system identification phase in order to focus on learning these parameters as quickly as possible, without spending time estimating the parameters of the system less critical for control? This is fundamentally a question of *exploration*. While it is known in linear systems that random excitation will efficiently explore [10], exploration in nonlinear systems is significantly more challenging since, in order to excite all parameters of interest, non-trivial planning may be required to ensure all relevant states are reached (as is the case in the example considered in Section 1.1).

We address both these questions in a particular class of nonlinear systems parameterized as:
$$\boldsymbol{x}_{h+1} = A_\star \boldsymbol{\phi}(\boldsymbol{x}_h, \boldsymbol{u}_h) + \boldsymbol{w}_h. \tag{1.1}$$
Here $\boldsymbol{x}_h \in \mathbb{R}^{d_x}$ denotes the state of the system, $\boldsymbol{u}_h \in \mathcal{U} \subseteq \mathbb{R}^{d_u}$ the input, $\boldsymbol{w}_h \sim \mathcal{N}(0, \sigma_{\boldsymbol{w}}^2 \cdot I)$ random noise, $\boldsymbol{\phi}(\cdot, \cdot) \in \mathbb{R}^{d_\phi}$ a (possibly nonlinear, known) feature map, and $A_\star \in \mathbb{R}^{d_x \times d_\phi}$ the (unknown) system parameter. Systems of this form are able to model a variety of real-world settings [11–15][2], and have been the subject of recent attention in the reinforcement learning community [1, 17, 14], yet the aforementioned questions have remained unanswered. Towards addressing this, in this work we make the following contributions:

1. For systems of the form (1.1), given some cost of interest which we wish to find a controller to minimize, we (a) formally characterize how estimation error translates into suboptimality in the learned controller, under the *certainty equivalent* control rule and (b) provide a lower bound on the loss of *any* (sufficiently regular) control rule learned from $T$ rounds of interaction with (1.1).

2. Motivated by this characterization, we present an algorithm which achieves the *instance-optimal* rate, with controller loss matching our lower bound. To the best of our knowledge, this is the first statistically optimal algorithm in the setting of nonlinear dynamical systems. Our algorithm relies on a generic reduction from policy optimization to optimal exploration in *arbitrary* dynamical systems (not necessarily of the form (1.1)), which may be of independent interest.

3. We present numerical experiments on several realistic nonlinear systems which illustrate that our approach—efficiently exploring to reduce uncertainty in parameters most relevant to learning a controller—yields significant gains in practice.

To further motivate our approach, we consider the following example.

## 1.1 Motivating Example

To motivate the need for effective exploration, we consider a simple 1-D system with nonlinear dynamics given by:
$$\boldsymbol{x}_{h+1} = a_1 \boldsymbol{x}_h + a_2 \boldsymbol{u}_h + \sum_{i=1}^{10} a_{i+2} \boldsymbol{\phi}_i(\boldsymbol{x}_h) + \boldsymbol{w}_h$$

---

[2]In real-world settings, $\boldsymbol{\phi}$ is typically (1) from physics (i.e., the system structure is known but some parameters such as drag coefficient are unknown [3]), (2) learned using representation learning or meta-learning [12, 15], and/or (3) from random features (e.g., any sufficiently regular, smooth nonlinear system $\boldsymbol{f}(\boldsymbol{x}, \boldsymbol{u})$ can be modeled by (1.1) using $N$ random features up to a $1/\sqrt{N}$ error [16]).

where $\phi_i(\boldsymbol{x}) = \max\{1 - 100(\boldsymbol{x} - c_i)^2, 0\}$ for some $c_i$. We choose $a_1 = 0.8, a_2 = 1$, and $a_3 = \ldots = a_{12} = -3$. We assume $a_{1:12}$ are unknown, $(\phi_i)_{i=1}^{10}$ is known, and set

$$\text{cost}(\boldsymbol{x}, \boldsymbol{u}) = (\boldsymbol{x} - c_1)^2 + 100^{-1} \cdot \boldsymbol{u}^2.$$

With this choice of cost, the optimal controller will attempt to direct the state $\boldsymbol{x}$ to the equilibrium point $c_1$ and maintain this position. Note that, with our choice of $\phi_i$, $\phi_i(\boldsymbol{x}) \neq 0$ only when $\boldsymbol{x}$ is very close to $c_i$. This renders the parameters $a_{4:12}$ irrelevant to learning the optimal controller, since $\phi_2, \ldots, \phi_{10}$ will be inactive if we are playing optimally, but learning $a_{1:3}$ is critical to performing optimally. In particular, the coefficient of the first nonlinearity, $a_3$, must be learned, as its value significantly changes the dynamics at the goal state.

We illustrate the result of running on this system in Figure 1, comparing our proposed approach (`Task-Driven Exploration`, Algorithm 1) to the approach which chooses $\boldsymbol{u}_h \sim \mathcal{N}(0, \sigma_{\boldsymbol{u}}^2)$ (`Random Exploration`), and the approach proposed in [1] (`Uniform Exploration`) which seeks to explore so as to estimate $a_{1:12}$ uniformly well. As can be seen, neither of these latter two approaches are able to learn a good controller, while our approach easily finds a near-optimal controller. The failure modes of each of these approaches is somewhat different. Here `Random Exploration` fails since the chance of reaching the point $\boldsymbol{x}_h \sim c_1$ is extremely small if the input is random noise—reaching $c_1$ requires playing a particular sequence of actions which are very unlikely to be played if $\boldsymbol{u}_h$ is

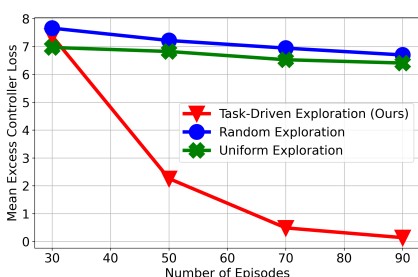

Figure 1: Performance on Motivating Example

chosen randomly. The `Uniform Exploration` approach does, in contrast, plan and, given enough time, is guaranteed to estimate all parameters accurately. However, as it aims to estimate all parameters uniformly well, it will attempt to estimate $a_{4:12}$ accurately despite their irrelevance to control, which will slow down the rate at which it is able to estimate $a_3$. Only our approach, which both plans and takes into account the cost while exploring, is able to reach $a_3$ enough times to efficiently estimate it, and learn a good controller.

This example illustrates that it is critical both to explore efficiently, and also to let the objective—learning a good controller—guide this exploration. We emphasize that the behavior in this example is only exhibited in nonlinear systems—though taking into account the task while exploring in linear systems is known to yield provable improvements [2], even playing random noise allows every direction to be learned in such systems. In nonlinear systems, however, this is not the case—one may fail to learn completely unless careful planning is performed.

## 2  Related Work

**Learning for control.**   Recently, there has been increased interest in studying control problems from a learning-theoretic perspective, largely for linear system settings such as online LQR or LQG with unknown dynamics [18, 10, 19–26]. In the nonlinear setting, [27–29] provide formal guarantees on system identification in several different classes of nonlinear systems, yet they only consider noiseless systems, or systems that are significantly easier to excite than (1.1). [17] study systems of the form (1.1), but consider only the regret minimization problem. While their bounds would yield a polynomial complexity via an online-to-batch conversion, the resulting guarantee would scale at a $\mathcal{O}(1/\sqrt{T})$ rate, significantly slower than our $\mathcal{O}(1/T)$ rate. Several additional works in reinforcement with general function approximation encompass nonlinear systems of the form (1.1) [30–32], yet these results also achieve a slow $\mathcal{O}(1/\sqrt{T})$ rate. The most relevant work [1] proposes an active learning approach to identify unknown parameters in (1.1), with the goal of minimizing the Euclidean distance in the parameter space. However, they do not provide end-to-end guarantees on learning controllers and, as shown in Section 1.1, this approach could be significantly worse than learning a model with the goal task in mind. Also very related to our work is [2], which seeks to answer a similar set of questions: performing system identification in order to learn a good controller. This work is restricted to the setting of linear dynamics, however, and does not address the additional complexities of nonlinear systems.

**System identification, dual control, and iterative learning control.**   There is a large body of classical work in system identification [7], and our work can be seen as an instance of *active* system

identification. While a variety of approaches have been proposed for active system identification [33–40], these tend to only consider linear systems, or lack rigorous theoretical guarantees. Recently deep learning approaches have also been applied in system identification [6, 8, 9, 41, 42]. In these works, the system identification phase is separate from the downstream controller design. Instead, in the control community, estimating parameters while simultaneously optimizing for performance has been formulated as a dual or iterative learning control problem [43–45], yet these settings focus on stability or asymptotic convergence whereas our work quantifies the end-to-end suboptimality gap.

**Model-based reinforcement learning.** This paper falls into the broad category of model-based reinforcement learning (MBRL), where an agent explores the environment to learn a model and then computes an optimal policy using the learned model. On the empirical side, deep MBRL has made exciting progress in many domains [46–48], and several task-aware methods have been designed to improve MBRL's performance [47–49], yet these works lack formal guarantees. On the theoretical side, a variety of different model-based approaches exist [50–55, 14]; however, the majority of these consider restricted settings such as tabular or linear MDPs. Of particular interest is the work of [14] which presents a result in systems of the form (1.1). While they show that polynomial sample complexity is possible, their guarantee scales at a $\mathcal{O}(1/T^{1/6})$ rate compared to our much faster $\mathcal{O}(1/T)$ rate.

**Adaptive nonlinear control.** Adaptive nonlinear control also seeks to control an unknown nonlinear system with parametric uncertainties [3, 4]. In particular, the key idea of model-reference adaptive control (MRAC) bears affinity to this paper, in that the adaptation law in MRAC adapts unknown parameters in a task-aware manner. There are two main differences between MRAC and our work. First, adaptive control does not explicitly optimize a cost function—the objective is typically tracking error convergence and Lyapunov stability, whereas our framework allows general cost functions— and the focus is typically on asymptotic convergence, while we give non-asymptotic optimality guarantees. Second, adaptive control has by and large been limited to specific system classes (e.g., fully-actuated systems [4, 15]) and policy classes (e.g., policy to directly cancel out the matched uncertainty [12, 13]), whereas our framework allows more general systems and policy classes.

## 3 Preliminaries

**Notation.** $\| \cdot \|_{\mathrm{op}}$ denotes the operator norm (matrix 2-norm), $\| \cdot \|_{\mathrm{F}}$ the Frobenius norm, and $\| \cdot \|_M$ the Mahalanobis norm, defined as $\|\boldsymbol{x}\|_M := \sqrt{\boldsymbol{x}^\top M \boldsymbol{x}}$ for $M \succeq 0$. $\mathrm{vec}(A)$ denotes the vectorization of matrix $A$. $\mathcal{B}_p(A; r) := \{A' : \|A - A'\|_p \leq r\}$. $[H] = \{1, 2, \ldots, H\}$. We let $\mathbb{E}_A[\cdot]$ denote the expectation over trajectories induced on system with parameter $A$, and $\mathbb{E}_{A,\pi}[\cdot]$ the expectation induced when policy $\pi$ is played. $\mathrm{poly}(\cdot)$ denotes some term that is polynomial in its arguments, with exponents absolute constants. We use $\lesssim$ informally to highlight key parameters in an inequality.

**Setting.** In this work, we are interested in systems of the form (1.1). We consider the episodic setting, where episodes are of length $H$, and assume that each episodes starts from a given state $\boldsymbol{x}_1$. We also assume $\|A_\star\|_{\mathrm{op}} \leq B_A$ for some known $B_A > 0$. We note that the setting considered here encompasses many real-world systems of interest in robotics and control (e.g., [12, 14, 11, 15] and Section 6).

The goal of the learner is to find a policy (controller) $\pi = (\pi_h)_{h=1}^H$ which achieves minimal cost on (1.1), for the cost defined by some (known) function $(\mathrm{cost}_h(\cdot, \cdot))_{h=1}^H$, with $\mathrm{cost}_h : \mathbb{R}^{d_x} \times \mathcal{U} \to \mathbb{R}_+$. For a given policy $\pi$, we define the expected cost on system $A$ as

$$\mathcal{J}(\pi; A) := \mathbb{E}_{A,\pi} \left[ \sum_{h=1}^H \mathrm{cost}_h(\boldsymbol{x}_h, \boldsymbol{u}_h) \right].$$

We consider the following interaction protocol:

1. Learner interacts with (1.1) for $T$ episodes, at each episode playing a policy $\pi_{\mathrm{exp}} \in \Pi_{\mathrm{exp}}$.
2. After $T$ episodes, the learner proposes a policy $\widehat{\pi}_T \in \Pi^\star$.
3. The learner suffers cost $\mathcal{J}(\widehat{\pi}_T; A_\star)$.

The goal of the learner is therefore first to explore and, after $T$ episodes of exploration, to propose its best guess at the optimal controller for (1.1), $\widehat{\pi}_T$. Here we take $\Pi_{\mathrm{exp}}$ to be a (known) set of admissible exploration policies (for example, policies with bounded input power), and $\Pi^\star$ a (known) set of admissible control policies. We assume that policies in $\Pi^\star$ are deterministic, but allow for randomized policies in $\Pi_{\mathrm{exp}}$. Policies may be either open- or closed-loop.

**System Notation.** We let $\mathcal{T}$ denote the space of all possible state trajectories, $\mathcal{T} \subseteq \mathbb{R}^{d_{\boldsymbol{x}} \times (H+1)}$, and, for any $\boldsymbol{\tau} \in \mathcal{T}$, let $\boldsymbol{\tau}_{1:h}$ denote the first $h$ states and inputs in $\boldsymbol{\tau}$. For any policy $\pi$, we denote

$$\boldsymbol{\Lambda}_{A,\pi} := \mathbb{E}_{A,\pi}\left[\sum_{h=1}^{H} \boldsymbol{\phi}(\boldsymbol{x}_h, \boldsymbol{u}_h)\boldsymbol{\phi}(\boldsymbol{x}_h, \boldsymbol{u}_h)^{\top}\right]$$

the expected covariance induced by playing $\pi$ on system $A$, $\boldsymbol{\Lambda}_{\pi} := \boldsymbol{\Lambda}_{A_\star, \pi}$, and $\check{\boldsymbol{\Lambda}} := I_{d_{\boldsymbol{x}}} \otimes \boldsymbol{\Lambda}$ the Kronecker product of $I_{d_{\boldsymbol{x}}}$ and $\boldsymbol{\Lambda}$. Finally, we let $\boldsymbol{\Omega}$ denote the convex hull of covariance matrices induced by $\Pi_{\exp}$, $\boldsymbol{\Omega} := \{\mathbb{E}_{\pi \sim \omega}[\boldsymbol{\Lambda}_\pi] : \omega \in \triangle_{\Pi_{\exp}}\}$, for $\triangle_{\Pi_{\exp}}$ the set of distributions over $\Pi_{\exp}$.

## 3.1 Regularity Assumptions

In order to make learning in (1.1) tractable, we need several regularity assumptions.

**Assumption 1** (Bounded Features). *For all $\boldsymbol{x} \in \mathbb{R}^{d_{\boldsymbol{x}}}$ and $\boldsymbol{u} \in \mathcal{U}$, we have $\|\boldsymbol{\phi}(\boldsymbol{x}, \boldsymbol{u})\|_2 \leq B_{\boldsymbol{\phi}}$.*

**Assumption 2** (Bounded Cost). *There exists some $r_{\mathrm{cost}}(A_\star) > 0$ such that, for all $A \in \mathcal{B}_{\mathrm{F}}(A_\star; r_{\mathrm{cost}}(A_\star))$ and all $\pi \in \Pi^\star$, we have $\mathbb{E}_{A,\pi}[(\sum_{h=1}^{H} \mathrm{cost}_h(\boldsymbol{x}_h, \boldsymbol{u}_h))^2] \leq L_{\mathrm{cost}}$.*

**Assumption 3** (Uniform Feature Excitation). *There exists $\omega \in \triangle_{\Pi_{\exp}}$ such that $\lambda_{\min}(\mathbb{E}_{\pi_{\exp} \sim \omega}[\boldsymbol{\Lambda}_{\pi_{\exp}}]) \geq \lambda_{\min}^\star$ for some $\lambda_{\min}^\star > 0$.*

We remark that these assumptions have appeared before in work on systems of the form (1.1) [1, 17]. In particular, Assumption 3 implies that every direction of $A_\star$ can, in principle, be excited, allowing it to be learned. In order to precisely quantify the optimal rates of learning, we require that our system satisfy certain smoothness assumptions. First, we require that $\boldsymbol{\phi}(\cdot, \cdot)$ is differentiable in its second argument.

**Assumption 4** (Smooth Nonlinearity). *For all $\boldsymbol{x} \in \mathbb{R}^{d_{\boldsymbol{x}}}$ and $\boldsymbol{u} \in \mathcal{U}$, $\boldsymbol{\phi}(\boldsymbol{x}, \boldsymbol{u})$ is four-times differentiable in $\boldsymbol{u}$. Furthermore, $\|\nabla_{\boldsymbol{u}}^{(i)} \boldsymbol{\phi}(\boldsymbol{x}, \boldsymbol{u})\|_{\mathrm{op}} \leq L_{\boldsymbol{\phi}}$, $\forall i \in \{1, 2, 3, 4\}$, $\boldsymbol{x} \in \mathbb{R}^{d_{\boldsymbol{x}}}$, and $\boldsymbol{u} \in \mathcal{U}$.*

We also require that the class of admissible control policies, $\Pi^\star$, has a parametric form, $\Pi^\star = \{\pi^{\boldsymbol{\theta}} : \boldsymbol{\theta} \in \mathbb{R}^{d_{\boldsymbol{\theta}}}\}$, and that the parameterization is smooth in the following sense.

**Assumption 5** (Smooth Controller Class). *$\pi_h^{\boldsymbol{\theta}}(\boldsymbol{\tau}_{1:h})$ is four-times differentiable in $\boldsymbol{\theta}$ for all $\boldsymbol{\tau} \in \mathcal{T}$ and $h \in [H]$. Furthermore, $\|\nabla_{\boldsymbol{\theta}}^{(i)} \pi_h^{\boldsymbol{\theta}}(\boldsymbol{\tau}_{1:h})\|_{\mathrm{op}} \leq L_{\boldsymbol{\theta}}$ for $\forall i \in \{1, 2, 3, 4\}$, $\boldsymbol{\theta} \in \mathbb{R}^{d_{\boldsymbol{\theta}}}$, and $\boldsymbol{\tau} \in \mathcal{T}$.*

Assumption 5 is satisfied for commonly considered classes of controllers, such as linear controllers, but is also satisfied by more complex classes such as neural network controllers. While the learner may propose any $\widehat{\pi}_T \in \Pi^\star$, we are particularly interested in the *certainty equivalence* decision rule (i.e., the learner decides $\widehat{\pi}_T$ as if the estimated system is the actual one), defined as:

$$\pi_\star(A) := \pi^{\boldsymbol{\theta}_\star(A)} \quad \text{for} \quad \boldsymbol{\theta}_\star(A) := \arg\min_{\boldsymbol{\theta} \in \mathbb{R}^{d_{\boldsymbol{\theta}}}} \mathcal{J}(\pi^{\boldsymbol{\theta}}; A). \tag{3.1}$$

To ensure that $\pi_\star(A)$ is well-defined and sufficiently regular, we make the following assumption.

**Assumption 6** (Unique Optimal Controller). *We assume that the global minimum of $\mathcal{J}(\pi^{\boldsymbol{\theta}}; A_\star)$, $\boldsymbol{\theta}_\star(A_\star)$, is unique, and that $\nabla_{\boldsymbol{\theta}}^2 \mathcal{J}(\pi^{\boldsymbol{\theta}}; A_\star)|_{\boldsymbol{\theta}=\boldsymbol{\theta}_\star(A_\star)} \succ 0$.*

In general, the policy optimization problem in (3.1) may not be computationally tractable. As we show in Appendix D, the globally optimal decision rule of (3.1) can be replaced with a locally optimal decision rule (i.e. $\pi_\star(A)$ a local minimum of $\mathcal{J}(\pi; A)$). Furthermore, Assumption 6 can be replaced by assuming the differentiability of $\boldsymbol{\theta}_\star(A)$ with respect to $A$ for $A$ near $A_\star$. For ease of exposition, in the main text we assume that Assumption 6 holds and that $\pi_\star(A)$ is defined as in (3.1). With these definitions and under Assumptions 1, 2, 4 and 5, we can show that $\mathcal{J}(\pi^{\boldsymbol{\theta}}; A_\star)$ is differentiable in $\boldsymbol{\theta}$ and, combined with Assumption 6, that $\boldsymbol{\theta}_\star(A)$ is differentiable in $A$, for $A \in \mathcal{B}_{\mathrm{F}}(A_\star; r_{\boldsymbol{\theta}}(A_\star))$ and some $r_{\boldsymbol{\theta}}(A_\star) > 0$. We let $L_{\pi_\star}$ denote an upper bound on the norm of the derivatives of $\boldsymbol{\theta}_\star(A)$.

## 4 Optimal Exploration in Nonlinear Systems

In this work, we are interested in characterizing the instance-optimal rates of learning a controller $\pi \in \Pi^\star$ which minimizes the loss $\mathcal{J}(\pi; A_\star)$. The following result, a generalization of Proposition 8.2 of [2] to nonlinear systems, is the starting point of our analysis.

**Proposition 1** (Informal). *Under Assumptions 1, 2 and 4 to 6 and on the system (1.1), we have*

$$\mathcal{J}(\pi_\star(\widehat{A}); A_\star) - \mathcal{J}(\pi_\star(A_\star); A_\star) = \|\mathrm{vec}(A_\star - \widehat{A})\|^2_{\mathcal{H}(A_\star)} + \mathcal{O}^\star(\|A_\star - \widehat{A}\|^3_{\mathrm{F}})$$

*for*

$$\mathcal{H}(A_\star) := \nabla^2_A \mathcal{J}(\pi_\star(A); A_\star)|_{A=A_\star}$$

*and where $\mathcal{O}^\star(\cdot)$ hides factors polynomial in the regularity parameters of Assumptions 1 to 6.*

The quantity $\mathcal{H}(A_\star) := \nabla^2_A \mathcal{J}(\pi_\star(A); A_\star)|_{A=A_\star}$, referred to as the *model-task Hessian* in [2], corresponds to the *curvature* of the loss of the certainty-equivalence controller $\pi_\star(A)$ around $A \leftarrow A_\star$. It precisely quantifies how estimation error in each coordinate of $A_\star$ translates into suboptimality of the controller—providing an answer to our question of which parameters are most relevant to learning a good controller—and reduces the problem of minimizing the controller loss to estimating $A_\star$ in a particular norm. The following result gives a bound on this estimation error, $\|\mathrm{vec}(A_\star - \widehat{A})\|^2_{\mathcal{H}(A_\star)}$.

**Proposition 2** (Informal). *Consider interacting with (1.1) for $T$ episodes, and let $\mathbf{\Lambda}_T = \sum_{t=1}^T \sum_{h=1}^H \boldsymbol{\phi}(\boldsymbol{x}_h^t, \boldsymbol{u}_h^t) \boldsymbol{\phi}(\boldsymbol{x}_h^t, \boldsymbol{u}_h^t)^\top$ denote the observed covariates and*

$$\widehat{A} = \arg\min_A \sum_{t=1}^T \sum_{h=1}^H \|\boldsymbol{x}_{h+1}^t - A\boldsymbol{\phi}(\boldsymbol{x}_h^t, \boldsymbol{u}_h^t)\|^2_2$$

*the least-squares estimate of $A_\star$. Recalling that $\check{\mathbf{\Lambda}}_T = I_{d_{\boldsymbol{x}}} \otimes \mathbf{\Lambda}_T$, we have, with high probability:*

$$\|\mathrm{vec}(A_\star - \widehat{A})\|^2_{\mathcal{H}(A_\star)} \lesssim \sigma^2_{\boldsymbol{w}} \cdot \mathrm{tr}(\mathcal{H}(A_\star) \check{\mathbf{\Lambda}}_T^{-1}).$$

## 4.1   Algorithm and Upper Bound

Proposition 2 motivates our algorithmic approach: explore to collect covariates $\mathbf{\Lambda}_T$ minimizing $\mathrm{tr}(\mathcal{H}(A_\star) \check{\mathbf{\Lambda}}_T^{-1})$. There are two primary challenges to achieving this: we do not know $\mathcal{H}(A_\star)$, as it depends on the (unknown) parameter $A_\star$ and, even if we did know $\mathcal{H}(A_\star)$, it is not clear how to explore so as to collect data minimizing $\mathrm{tr}(\mathcal{H}(A_\star) \check{\mathbf{\Lambda}}_T^{-1})$. We address both of these challenges in Algorithm 1.

---

**Algorithm 1** Optimal Exploration in Nonlinear Systems (informal)

---

1: **inputs:** episodes $T$, $(\mathrm{cost}_h)_{h=1}^H$, confidence $\delta$, control policies $\Pi^\star$, exploration policies $\Pi_{\mathrm{exp}}$
2: $\widehat{A}^1 \leftarrow 0$, $\ell_T \leftarrow \lceil \log_2 T/8 \rceil$, $T_\ell \leftarrow 2^\ell$
3: **for** $\ell = 1, 2, 3, \ldots, \ell_T$ **do**
4:     Compute estimate of model-task Hessian: $\mathcal{H}_\ell \leftarrow \mathcal{H}(\widehat{A}^\ell)$
5:     Run DYNAMICOED on $\Phi_\ell(\mathbf{\Lambda}) \leftarrow \mathrm{tr}(\mathcal{H}_\ell \cdot \mathbf{\Lambda}^{-1})$ to learn exploration policies $\Pi_\ell \subseteq \Pi_{\mathrm{exp}}$
6:     Rerun each policy in $\Pi_\ell$ $N_\ell = \lceil T_\ell/|\Pi_\ell| \rceil$ times, denote collected data $\mathfrak{D}_\ell$
7:     Estimate $A_\star$: $\widehat{A}^{\ell+1} = \arg\min_A \sum_{h=1}^H \sum_{(\boldsymbol{x}_{h+1}, \boldsymbol{u}_h, \boldsymbol{x}_h) \in \mathfrak{D}_\ell} \|\boldsymbol{x}_{h+1} - A\boldsymbol{\phi}(\boldsymbol{x}_h, \boldsymbol{u}_h)\|^2_2$
8: **return** $\widehat{\pi}_T \leftarrow \pi_\star(\widehat{A}^{\ell_T+1}) \in \Pi^\star$

---

Algorithm 1 proceeds in epochs of exponentially increasing length. At each epoch it first approximates $\mathcal{H}(A_\star)$ by computing the model-task Hessian of the estimated system, $\widehat{A}^\ell$. Using this approximatiom of $\mathcal{H}(A_\star)$, it seeks to explore to minimize $\mathrm{tr}(\mathcal{H}(\widehat{A}^\ell) \check{\mathbf{\Lambda}}_T^{-1})$. This exploration routine is encapsulated in the DYNAMICOED (dynamic optimal experiment design) function, an adaptive experiment-design routine inspired by recent work in reinforcement learning [56] and described in more detail in Section 5. DYNAMICOED returns a set of exploration policies, $\Pi_\ell$, which we run to collect data $\mathfrak{D}_\ell$. As we will show, the collected covariates, $\mathbf{\Lambda}_\ell := \sum_{h=1}^H \sum_{(\boldsymbol{u}_h, \boldsymbol{x}_h) \in \mathfrak{D}_\ell} \boldsymbol{\phi}(\boldsymbol{x}_h, \boldsymbol{u}_h) \boldsymbol{\phi}(\boldsymbol{x}_h, \boldsymbol{u}_h)^\top$, satisfy

$$\mathrm{tr}(\mathcal{H}(\widehat{A}^\ell) \check{\mathbf{\Lambda}}_\ell^{-1}) \lesssim T_\ell^{-1} \cdot \min_{\mathbf{\Lambda} \in \boldsymbol{\Omega}} \mathrm{tr}(\mathcal{H}(\widehat{A}^\ell) \check{\mathbf{\Lambda}}^{-1}),$$

which implies that DYNAMICOED collects data minimizing $\mathrm{tr}(\mathcal{H}(\widehat{A}^\ell) \check{\mathbf{\Lambda}}_\ell^{-1})$ at a near-optimal rate. Given the data $\mathfrak{D}_\ell$, we form the least-squares estimate of $A_\star$, $\widehat{A}^{\ell+1}$, and the process repeats. After running for $T$ episodes, the certainty-equivalence controller on the last estimate obtained, $\widehat{\pi}_T = \pi_\star(\widehat{A}^{\ell_T+1})$, is returned. The following result bounds the suboptimality of $\widehat{\pi}_T$ as compared to $\pi_\star(A_\star)$.

**Theorem 1.** *Under Assumptions 1 to 6, if $T \geq C_{\text{poly}} \cdot \max\{1, r_{\text{cost}}(A_\star)^{-2}, r_{\boldsymbol{\theta}}(A_\star)^{-2}\}$, then with probability at least $1 - \delta$, Algorithm 1 explores with policies in $\Pi_{\text{exp}}$ at every episode, runs for at most $T$ episodes, and returns $\widehat{\pi}_T \in \Pi^\star$ satisfying:*

$$\mathcal{J}(\widehat{\pi}_T; A_\star) - \mathcal{J}(\pi_\star(A_\star); A_\star) \leq \frac{\sigma_{\boldsymbol{w}}^2}{T} \cdot \min_{\boldsymbol{\Lambda} \in \boldsymbol{\Omega}} \text{tr}\left(\mathcal{H}(A_\star)\check{\boldsymbol{\Lambda}}^{-1}\right) \cdot C \log \frac{6 d_{\boldsymbol{x}} d_{\boldsymbol{\phi}}}{\delta} + \frac{C_{\text{poly}}}{T^{3/2}}$$

*where we recall $\boldsymbol{\Omega}$ is the set of possible expected covariates on (1.1), $C$ is a universal constant, and*

$$C_{\text{poly}} = \text{poly}(d_{\boldsymbol{\phi}}, d_{\boldsymbol{x}}, H, B_A, B_{\boldsymbol{\phi}}, L_{\boldsymbol{\phi}}, L_{\boldsymbol{\theta}}, L_{\text{cost}}, L_{\pi_\star}, \sigma_{\boldsymbol{w}}, \sigma_{\boldsymbol{w}}^{-1}, \tfrac{1}{\lambda_{\text{min}}^\star}, \log \tfrac{T}{\delta}).$$

Theorem 1 shows that Algorithm 1 is able to explore so as to optimally minimize the exploration loss $\text{tr}(\mathcal{H}(A_\star)\check{\boldsymbol{\Lambda}}_T^{-1})$, up to a lower-order term scaling as $T^{-3/2}$ and polynomially in system parameters. While Propositions 1 and 2 show that collecting data which minimizes $\text{tr}(\mathcal{H}(A_\star)\check{\boldsymbol{\Lambda}}_T^{-1})$ is in some sense fundamental, it is not clear it is necessary. We next show that it is indeed necessary.

### 4.2 Lower Bounds on Learning Controllers

Our goal is to show that, up to constants and lower-order terms, the bound given in Theorem 1 is not improvable, regardless of which controller estimate we use. To obtain such lower bounds, we need several additional assumptions. In particular, we require that the loss $\mathcal{J}(\pi^{\boldsymbol{\theta}}; A)$ grows quadratically in the distance $\boldsymbol{\theta}$ is from $\boldsymbol{\theta}_\star(A)$, and strengthen Assumption 3 to ensure (1.1) is sufficiently easy to excite. Formal statements of these conditions are given in Appendix F. Our lower bound is as follows.

**Theorem 2** (Informal). *Under Assumptions 1 to 6 and the additional regularity assumptions mentioned above, as long as $T \geq C_{\text{lb}}$, for any $\omega_{\text{exp}} \in \triangle_{\Pi_{\text{exp}}}$, we have*

$$\min_{\widehat{\pi}} \max_{A \in \mathcal{B}_T} \mathbb{E}_{\mathfrak{D}_T \sim A, \omega_{\text{exp}}}[\mathcal{J}(\widehat{\pi}(\mathfrak{D}_T); A) - \mathcal{J}(\pi_\star(A); A)] \geq \frac{\sigma_{\boldsymbol{w}}^2}{3T} \cdot \min_{\boldsymbol{\Lambda} \in \boldsymbol{\Omega}} \text{tr}(\mathcal{H}(A_\star)\check{\boldsymbol{\Lambda}}^{-1}) - \frac{C_{\text{lb}}}{T^{5/4}}$$

*for $\mathcal{B}_T := \mathcal{B}_{\text{F}}(A_\star; \mathcal{O}(T^{-5/6}))$, $\mathbb{E}_{\mathfrak{D}_T \sim A, \omega_{\text{exp}}}[\cdot] = \mathbb{E}_{\pi_{\text{exp}} \sim \omega_{\text{exp}}}[\mathbb{E}_{\mathfrak{D}_T \sim A, \pi_{\text{exp}}}[\cdot]]$ the expectation over trajectories generated by running policies $\pi \sim \omega_{\text{exp}}$ on system $A$ for $T$ episodes, $\widehat{\pi}$ any mapping from observations to policies in $\Pi^\star$, and $C_{\text{lb}}$ some value scaling polynomially in problem parameters.*

Note that this lower bound holds for *any* $A_\star$ and mapping $\boldsymbol{\phi}$, as long as our assumptions are met. Up to constants and lower-order terms, the scaling of Theorem 2 matches that of Theorem 1—both scale with $\min_{\boldsymbol{\Lambda} \in \boldsymbol{\Omega}} \text{tr}(\mathcal{H}(A_\star)\check{\boldsymbol{\Lambda}}^{-1})$—which implies that Algorithm 1 is indeed optimal (under certain additional regularity conditions). To the best of our knowledge, this is the first result characterizing the optimal statistical rates for learning in nonlinear dynamical systems. We emphasize that Theorem 2 holds for *any* decision rule $\widehat{\pi}$—it does not require that we use the certainty equivalence decision rule. As Algorithm 1 does rely on certainty equivalence, this result also implies that the certainty equivalence decision rule is optimal for (certain classes of) nonlinear dynamical systems.

The proof of Theorem 2 builds on the work [2], which shows a similar result for linear dynamical systems. It critically relies on our quadratic decomposition of the controller loss in Proposition 1, which reduces the problem of obtaining a lower bound on controller loss to a lower bound on estimating $A_\star$ in the $\mathcal{H}(A_\star)$ norm. Given this, the result can be obtained by applying lower bounds on regression in general norms.

## 5 Optimal Experiment Design in Arbitrary Dynamical Systems

We turn now to the DYNAMICOED routine, which is the key algorithmic tool we use to prove Theorem 1. DYNAMICOED is a general reduction from policy optimization to optimal experiment design in arbitrary dynamical systems, and is an extension of a recently proposed approach for experiment design in linear MDPs [56]. This section may be of independent interest.

To illustrate the generality of this reduction, in this section we consider the following system:

$$\boldsymbol{x}_{h+1} = f_h(\boldsymbol{x}_h, \boldsymbol{u}_h, \boldsymbol{w}_h), \quad h = 1, 2, \ldots, H, \tag{5.1}$$

where $\boldsymbol{x}_h \in \mathcal{X} \subseteq \mathbb{R}^{d_{\boldsymbol{x}}}$ denotes the state, $\boldsymbol{u}_h \in \mathcal{U} \subseteq \mathbb{R}^{d_{\boldsymbol{u}}}$ the input, and $\boldsymbol{w}_h \in \mathbb{R}^{d_{\boldsymbol{w}}}$ the noise. We take the dynamics $(f_h)_{h=1}^H$ to be unknown and arbitrary. We assume there is some known featurization of

our system that is of interest, $\phi(\boldsymbol{x}, \boldsymbol{u}) \to \mathbb{R}^{d_\phi}$, and an experiment design object on this featurization, $\Phi : \mathbb{R}^{d_\phi \times d_\phi} \to \mathbb{R}$. Our goal is to collect some set of trajectories $\{\boldsymbol{\tau}_t\}_{t=1}^T$ which minimizes $\Phi$:

$$\Phi\big(\tfrac{1}{TH} \cdot \sum_{t=1}^T \sum_{h=1}^H \phi(\boldsymbol{x}_h^t, \boldsymbol{u}_h^t)\phi(\boldsymbol{x}_h^t, \boldsymbol{u}_h^t)^\top\big).$$

As an example, if $\Phi(\boldsymbol{\Lambda}) = \log \det(\boldsymbol{\Lambda})$, this reduces to $D$-optimal design, and if $\Phi(\boldsymbol{\Lambda}) = \mathrm{tr}(\mathcal{H} \cdot \boldsymbol{\Lambda}^{-1})$, the setting considered in Section 4, this reduces to weighted $A$-optimal design. As before, we assume we have access to some set of exploration policies $\Pi_{\mathrm{exp}}$, and define $\boldsymbol{\Lambda}_\pi$ and $\boldsymbol{\Omega}$ as in Section 3, but with respect to this new feature map $\phi$ and system (5.1). We also define $\widehat{\boldsymbol{\Omega}}$ to be the space of all possible covariance matrices:

$$\widehat{\boldsymbol{\Omega}} := \big\{\sum_{h=1}^H \phi(\boldsymbol{x}_h, \boldsymbol{u}_h)\phi(\boldsymbol{x}_h, \boldsymbol{u}_h)^\top \ : \ \boldsymbol{x}_h \in \mathcal{X}, \boldsymbol{u}_h \in \mathcal{U}, \forall h \in [H]\big\}.$$

To facilitate efficient experiment design in this setting, we will make the following assumption on $\Phi$.

**Assumption 7** (Regularity of $\Phi$). *$\Phi$ is regular in the following sense:*

*1. $\Phi$ is convex, differentiable, and $\beta$-smooth in the norm $\|\cdot\|$ (with dual-norm $\|\cdot\|_*$):*

$$\|\nabla_{\boldsymbol{\Lambda}}\Phi(\boldsymbol{\Lambda}) - \nabla_{\boldsymbol{\Lambda}'}\Phi(\boldsymbol{\Lambda}')\|_* \le \beta \cdot \|\boldsymbol{\Lambda} - \boldsymbol{\Lambda}'\|, \quad \forall \boldsymbol{\Lambda}, \boldsymbol{\Lambda}' \in \widehat{\boldsymbol{\Omega}}.$$

*2. There exists some $M < \infty$ satisfying $\sup_{\boldsymbol{\Lambda} \in \widehat{\boldsymbol{\Omega}}} \sup_{\boldsymbol{x} \in \mathcal{X}, \boldsymbol{u} \in \mathcal{U}} |\phi(\boldsymbol{x}, \boldsymbol{u})^\top \nabla_{\boldsymbol{\Lambda}}\Phi(\boldsymbol{\Lambda})\phi(\boldsymbol{x}, \boldsymbol{u})| \le M$.*

The key algorithmic assumption we make is access to a regret minimization oracle on (5.1).

**Assumption 8** (Regret Minimization Oracle). *Let $\mathrm{cost}_h(\boldsymbol{x}, \boldsymbol{u}) = \phi(\boldsymbol{x}, \boldsymbol{u})^\top Q_h \phi(\boldsymbol{x}, \boldsymbol{u})$ for some $Q_h \in \mathbb{R}^{d_\phi \times d_\phi}$ such that $|\sum_h \mathrm{cost}_h(\boldsymbol{x}_h, \boldsymbol{u}_h)| \le 1$ for all $\boldsymbol{x}_h \in \mathcal{X}, \boldsymbol{u}_h \in \mathcal{U}$. We assume we have access to some learner $\mathbb{A}_\mathcal{R}$ which is able to achieve low regret on costs $\{\mathrm{cost}_h(\cdot, \cdot)\}_{h=1}^H$ with respect to policy class $\Pi_{\mathrm{exp}}$. That is, with probability at least $1 - \delta$:*

$$\sum_{t=1}^T \mathbb{E}_{f, \pi_t}[\sum_{h=1}^H \mathrm{cost}_h(\boldsymbol{x}_h^t, \boldsymbol{u}_h^t)] - T \cdot \min_{\pi \in \Pi_{\mathrm{exp}}} \mathbb{E}_{f, \pi}[\sum_{h=1}^H \mathrm{cost}_h(\boldsymbol{x}_h, \boldsymbol{u}_h)] \le C_\mathcal{R} \cdot \log^{p_\mathcal{R}} \tfrac{T}{\delta} \cdot T^\alpha$$

*for some $C_\mathcal{R} > 0$, $p_\mathcal{R} > 0$, and $\alpha \in (0, 1)$, and where $\pi_t$ is the policy $\mathbb{A}_\mathcal{R}$ plays at episode $t$.*

Note that the regret minimization algorithm satisfying Assumption 8 may be arbitrary. For example, for linear systems, we could apply provably efficient algorithms for the Linear Quadratic Regulator [25, 20]; for nonlinear systems of the form (1.1) we could apply the $\mathrm{LC}^3$ algorithm of [17]; for more general settings of reinforcement learning with function approximation, algorithms such as BiLin-UCB [30] or E2D [31] could be applied. In practice, though they may not formally satisfy the guarantee of Assumption 8, deep RL approaches could be used. We have the following result.

**Theorem 3.** *Fix $T > 0$ and denote $R := \sup_{\boldsymbol{\Lambda}, \boldsymbol{\Lambda}' \in \widehat{\boldsymbol{\Omega}}} \|\boldsymbol{\Lambda} - \boldsymbol{\Lambda}'\|$. Under Assumption 7, and assuming we have access to a learner $\mathbb{A}_\mathcal{R}$ satisfying Assumption 8 with $\alpha = 1/2$, DYNAMICOED runs for $T$ episodes on (5.1), and with probability at least $1 - \delta$ collects data $\{(\boldsymbol{x}_h^t, \boldsymbol{u}_h^t)\}_{h \in [H], t \in [T]}$ satisfying*

$$\Phi\bigg(\frac{1}{T} \cdot \sum_{t=1}^T \sum_{h=1}^H \phi_h^t(\phi_h^t)^\top\bigg) - \min_{\boldsymbol{\Lambda} \in \boldsymbol{\Omega}} \Phi(\boldsymbol{\Lambda}) \le \frac{\beta R^2 \log T + HM(C_\mathcal{R} \log^{p_\mathcal{R}} \frac{2T}{\delta} + 3\log^{1/2} \frac{4T}{\delta})}{T^{1/3}}$$

*where $\phi_h^t := \phi(\boldsymbol{x}_h^t, \boldsymbol{u}_h^t)$, and we recall $\boldsymbol{\Omega}$ is the set of possible expected covariates on (5.1).*

Theorem 3 shows that, given access only to a regret minimization oracle, it is possible to solve experiment design problems on arbitrary dynamical systems. The requirement that $\alpha = 1/2$ is for expositional purposes only—we generalize this result to arbitrary $\alpha$ (and more general feature maps) in Appendix C. Under certain conditions, it can be shown that, if the exploration policies DYNAMICOED runs to collect $\mathfrak{D}$ are *rerun*, the newly collected data satisfies a similar guarantee as Theorem 3. This lets us run DYNAMICOED to learn an approximate solution of $\min_{\boldsymbol{\Lambda} \in \boldsymbol{\Omega}} \Phi(\boldsymbol{\Lambda})$, and then rerun the learned policies as many times as desired to collect additional data approximately minimizing $\Phi$.

## 5.1 Overview of DYNAMICOED Algorithm

DYNAMICOED is inspired by recent work on experiment design in reinforcement learning [57, 58, 56, 59], and can be seen as an extension of the FWREGRET algorithm of [56] to arbitrary systems. We

refer the reader to [56] for a more in-depth discussion of the FWREGRET algorithm, and briefly sketch its extension to arbitrary systems here (see Appendix C and Algorithm 4 for precise definitions).

---

**Algorithm 2** Dynamic Optimal Experiment Design (DYNAMICOED, Informal)

1: **input**: objective $\Phi$, episodes $T$, confidence $\delta$, regret algorithm $\mathbb{A}_{\mathcal{R}}$, exploration policies $\Pi_{\mathrm{exp}}$
2: Set $K \leftarrow \mathcal{O}(T^{2/3})$, $N \leftarrow \mathcal{O}(T^{1/3})$, $\gamma_n \leftarrow \frac{1}{n+1}$
    $// \ \boldsymbol{\phi}_h^{k,n} := \boldsymbol{\phi}(\boldsymbol{x}_h^{k,n}, \boldsymbol{u}_h^{k,n})$ for $(\boldsymbol{x}_h^{k,n}, \boldsymbol{u}_h^{k,n})$ the state-input at step $h$ of episode $k$ of iteration $n$
3: Play any $\pi_{\mathrm{exp}} \in \Pi_{\mathrm{exp}}$ for $K$ episodes, set $\boldsymbol{\Lambda}_0 \leftarrow \frac{1}{K} \sum_{k=1}^{K} \sum_{h=1}^{H} \boldsymbol{\phi}_h^{k,0} (\boldsymbol{\phi}_h^{k,0})^\top$
4: **for** $n = 1, 2, \ldots, N$ **do**
5:     Compute derivative of $\Phi(\boldsymbol{\Lambda}_{n-1})$, $\Xi_n \leftarrow \nabla_{\boldsymbol{\Lambda}} \Phi(\boldsymbol{\Lambda})|_{\boldsymbol{\Lambda}=\boldsymbol{\Lambda}_{n-1}}$
6:     Run $\mathbb{A}_{\mathcal{R}}$ on cost $\mathrm{cost}_h^n(\boldsymbol{x}, \boldsymbol{u}) \leftarrow \frac{1}{M} \cdot \boldsymbol{\phi}(\boldsymbol{x}, \boldsymbol{u})^\top (\Xi_n) \boldsymbol{\phi}(\boldsymbol{x}, \boldsymbol{u})$ for $K$ episodes
7:     $\boldsymbol{\Lambda}_n \leftarrow (1 - \gamma_n) \boldsymbol{\Lambda}_{n-1} + \frac{\gamma_n}{K} \cdot \sum_{k=1}^{K} \sum_{h=1}^{H} \boldsymbol{\phi}_h^{k,n} (\boldsymbol{\phi}_h^{k,n})^\top$
8: **return** $\frac{1}{T} \sum_{n=0}^{N} \sum_{k=1}^{K} \sum_{h=1}^{H} \boldsymbol{\phi}_h^{k,n} (\boldsymbol{\phi}_h^{k,n})^\top$

---

Conceptually, DYNAMICOED runs a variant of conditional gradient descent on the objective $\Phi(\boldsymbol{\Lambda})$. At each iteration, $n$, it computes the gradient of the loss at the current iterate, $\Xi_n \leftarrow \nabla_{\boldsymbol{\Lambda}} \Phi(\boldsymbol{\Lambda})|_{\boldsymbol{\Lambda}=\boldsymbol{\Lambda}_{n-1}}$. To run a standard gradient descent algorithm on this objective, we would simply update $\boldsymbol{\Lambda}_{n-1}$ by taking a step in the direction $\Xi_n$. However, our objective is to minimize $\Phi$ over the constraint set, $\boldsymbol{\Omega}$. Thus, rather than taking a step in the direction $\Xi_n$, we wish to take a step in the direction of steepest descent *within the constraint set*.

The challenge is that the constraint set in our setting, $\boldsymbol{\Omega}$, is *unknown*, as it depends on the expectation over trajectories induced on the unknown dynamics $(f_h)_{h=1}^{H}$, and therefore we cannot directly compute this steepest descent direction. The key observation is that the computation of this steepest descent direction is equivalent to solving:

$$\arg \min_{\pi_{\mathrm{exp}} \in \Pi_{\mathrm{exp}}} \mathbb{E}_{f, \pi_{\mathrm{exp}}} \left[ \sum_{h=1}^{H} \boldsymbol{\phi}(\boldsymbol{x}_h, \boldsymbol{u}_h)^\top (\Xi_n) \boldsymbol{\phi}(\boldsymbol{x}_h, \boldsymbol{u}_h) \right].$$

This is simply a policy optimization problem, however, and can be solved approximately by $\mathbb{A}_{\mathcal{R}}$ under Assumption 8. Thus, in the call to $\mathbb{A}_{\mathcal{R}}$ on Line 6, we approximate the steepest descent direction, and on Line 7 update $\boldsymbol{\Lambda}_{n-1}$ in this direction. Convergence of this procedure to the optimal value, $\min_{\boldsymbol{\Lambda} \in \boldsymbol{\Omega}} \Phi(\boldsymbol{\Lambda})$, can then be shown by the standard analysis of conditional gradient descent. We remark that, under Assumption 7 and Assumption 8, this argument is completely generic and does not require that our system, (5.1), exhibit any additional properties.

## 5.2 From Theorem 3 to Theorem 1

In Algorithm 1, our goal is to collect covariates, $\check{\boldsymbol{\Lambda}}_{T_\ell}^{-1}$, on (1.1) such that $\mathrm{tr}(\mathcal{H}(\widehat{A}^\ell) \check{\boldsymbol{\Lambda}}_{T_\ell}^{-1})$ is as small as possible. To achieve this, we apply DYNAMICOED to the dynamics (1.1) and objective $\Phi_\ell(\boldsymbol{\Lambda}) = \mathrm{tr}(\mathcal{H}(\widehat{A}^\ell) \check{\boldsymbol{\Lambda}}^{-1})$, with Assumption 8 instantiated by the LC$^3$ algorithm of [17]. By the guarantee given in Theorem 3, after running for a number of episodes $N$ which scales polynomially in problem parameters, DYNAMICOED will collect covariates $\boldsymbol{\Lambda}_N$ such that $\Phi_\ell(\frac{1}{N} \boldsymbol{\Lambda}_N) \leq 2 \cdot \min_{\boldsymbol{\Lambda} \in \boldsymbol{\Omega}} \Phi_\ell(\boldsymbol{\Lambda})$, which implies $\mathrm{tr}(\mathcal{H}(\widehat{A}^\ell) \check{\boldsymbol{\Lambda}}_N^{-1}) \leq \frac{2}{N} \cdot \min_{\boldsymbol{\Lambda} \in \boldsymbol{\Omega}} \mathrm{tr}(\mathcal{H}(\widehat{A}^\ell) \check{\boldsymbol{\Lambda}}^{-1})$. By rerunning the policies DYNAMICOED used to collect this $\boldsymbol{\Lambda}_N$ for $T_\ell/N$ additional times, we can ensure $\mathrm{tr}(\mathcal{H}(\widehat{A}^\ell) \check{\boldsymbol{\Lambda}}_{T_\ell}^{-1}) \lesssim \frac{1}{T_\ell} \cdot \min_{\boldsymbol{\Lambda} \in \boldsymbol{\Omega}} \mathrm{tr}(\mathcal{H}(\widehat{A}^\ell) \check{\boldsymbol{\Lambda}}^{-1})$ as desired.

We remark that the LC$^3$ algorithm requires access to a computation oracle. As the focus of this work is primarily statistical, we leave addressing this computational challenge for future work. Furthermore, as we show in the following section, computationally efficient, sampling-based implementations of our approach are very effective in practice. We remark as well that the objective we ultimately care about minimizing is $\mathrm{tr}(\mathcal{H}(A_\star) \check{\boldsymbol{\Lambda}}^{-1})$. As we show, by including a small amount of uniform exploration, we can ensure that the suboptimality incurred optimizing $\mathrm{tr}(\mathcal{H}(\widehat{A}^\ell) \check{\boldsymbol{\Lambda}}^{-1})$ instead of $\mathrm{tr}(\mathcal{H}(A_\star) \check{\boldsymbol{\Lambda}}^{-1})$ only contributes to the lower-order terms of the final guarantee in Theorem 1.

## 6 Experimental Results

Finally, we demonstrate the effectiveness of our proposed approach (Algorithm 1, the `Task-Driven Exploration` method in Figures 1 to 3) on several systems motivated by robotic applications. We

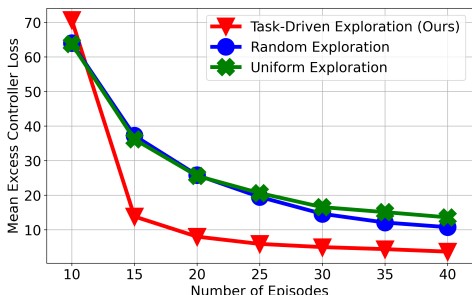

Figure 2: Performance on Drone

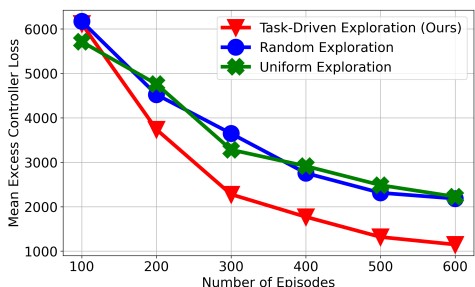

Figure 3: Performance on Car

compare Algorithm 1 with an approach that plays $\boldsymbol{u}_h \sim \mathcal{N}(0, \sigma_{\boldsymbol{u}}^2 \cdot I)$ (Gaussian Exploration), and an approach inspired by [1] (Uniform Exploration), which seeks to estimate $A_\star$ uniformly well, playing inputs that reduce $\|\widehat{A} - A_\star\|_{\mathrm{op}}$.

To benchmark the performance of these approaches, we consider an affine system with dynamics corresponding to that of a simplified 3-D drone (i.e., 3-D double integrator with a gravity term), and a nonlinear system with dynamics corresponding to that of a 2-D car. For both systems, we choose $H = 50$, and plot the value of $\mathcal{J}(\widehat{\pi}_t; A_\star) - \mathcal{J}(\pi_\star(A_\star); A_\star)$ for $\widehat{\pi}_t$ the certainty-equivalence controller computed on the estimate of the system obtained at time $t$. For the drone, we let $\Pi^\star$ be the class of linear-affine feedback controllers, and for the car, $\Pi^\star$ is a set of nonlinear controllers with dimension 4. While the optimal controller for the drone can be computed in closed-form, for the car we rely on a sampling-based routine to find an approximately optimal controller. The model-task hessian $\mathcal{H}(\widehat{A}^\ell)$ is computed via automatic differentiation. For the exploration policies of Task-Driven Exploration and Uniform Exploration, $\Pi_{\mathrm{exp}}$, we rely on MPC-style sampling based methods. For all approaches, we require that $\mathbb{E}_{A, \pi_{\mathrm{exp}}}[\sum_{h=1}^{H} \|\boldsymbol{u}_h\|_2^2] \leq \gamma^2$ for some $\gamma^2 > 0$ and all $\pi_{\mathrm{exp}} \in \Pi_{\mathrm{exp}}$. On all examples, we implement DYNAMICOED with $\mathbb{A}_\mathcal{R}$ a posterior sampling-inspired version of the LC$^3$ of [17]. Figures 1 and 3 shows performance averaged over 100 trials, and Figure 2 over 200 trials. Additional experimental details can be found in Appendix G.

As illustrated in Figures 1 to 3, our approach yields a non-trivial gain over existing approaches on all systems. In particular, in Figures 2 and 3 it improves on the sample complexity of existing approaches by roughly a factor of 2—for example, in the drone system, reaching excess controller cost of 10 after less than 20 episodes, as compared to over 40 episodes for existing approaches.

Our implementation is very modular, and any piece (for example, the parameterization of $\Pi_{\mathrm{exp}}$ and $\Pi^\star$, the policy optimizer, or the exploration routine) can be easily replaced with other procedures. Our results highlight that, even when using, for example, a possibly suboptimal policy optimizer, exploring so as to minimize uncertainty in the model-task hessian yields a non-trivial gain. We expect that this would hold true regardless of the policy optimizer used—the model-task hessian will adapt to the structure of the policy optimizer. Integration of our approach with deep model-based RL approaches is an interesting direction for future work, but we believe the approach will scale to these settings as well.

## 7 Conclusion

In this work, we have characterized the instance-optimal rate of learning controllers in nonlinear dynamical systems. To the best of our knowledge, this is the first work to obtain an optimal sample complexity for learning in nonlinear dynamical systems. Furthermore, our experimental results demonstrate the effectiveness of our proposed algorithm in realistic nonlinear systems. This work opens the door for several interesting directions for future work. First, it is not clear that the assumption on uniform feature excitation, Assumption 3, is necessary. Can this be removed with a more refined analysis (or shown to be necessary)? Second, while in many settings $\phi$ is known or can be effectively represented by random features, in some settings it is helpful to learn it [12]. Can we obtain end-to-end guarantees on learning both $\phi$ and $A_\star$? Finally, it is of much interest to extend our experimental results to larger-scale systems for real-world deployment.

**Acknowledgements**

AW would like to thank Kevin Tully for helpful discussions. The work of AW is supported by NSF HDR 62-0221. The work of KJ is supported in part by NSF TRIPODS 2023166 and CIF 2007036.

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

# A Technical Tools

**Additional Notation.** We let $\mathcal{S}^{d-1}$ refer to the unit ball in $d$ dimensions and $\mathbb{S}_+^d$ (resp. $\mathbb{S}_{++}^d$) the set of positive semi-definite matrices (resp. positive definite matrices) in $\mathbb{R}^{d\times d}$. Throughout, $\mathcal{O}(\cdot)$ denotes standard big-O notation, $\widetilde{\mathcal{O}}(\cdot)$ hides additional logarithmic factors. $\mathrm{poly}(\cdot)$ denotes some term that is polynomial in its arguments, with exponents absolute constants. $\triangle_{\mathcal{X}}$ denotes the set of distributions over set $\mathcal{X}$.

**Lemma A.1.** *Let $\boldsymbol{w}_i \sim \mathcal{N}(0, I_{d_{\boldsymbol{x}}})$ for all $i \in \{1, 2, \ldots, n\}$. Then,*

$$\mathbb{E}\left[\left(\sum_{i=1}^n \|\boldsymbol{w}_i\|_2\right)^c\right] \leq n^2 \cdot \mathrm{poly}(d_{\boldsymbol{x}}).$$

*for $c$ an absolute constant.*

*Proof.* We first bound

$$(\sum_{i=1}^n \|\boldsymbol{w}_i\|_2)^c \leq n \cdot \max_i \|\boldsymbol{w}_i\|_2^c \leq n \cdot \sum_i \|\boldsymbol{w}_i\|_2^c.$$

The result then follows since we can bound the $\mathbb{E}[\|\boldsymbol{w}_i\|_2^c] \leq \mathrm{poly}(d)$ for $\boldsymbol{w}_i \sim \mathcal{N}(0, I)$ and $c$ an absolute constant. $\qquad\square$

**Lemma A.2** (Lemma I.4 of [2])**.** *Assume $A, B \in \mathbb{S}_{++}^d$, $\|A - B\|_{\mathrm{op}} \leq \epsilon$, and $\epsilon < \lambda_{\min}(B)$. Then*

$$\|A^{-1} - B^{-1}\|_{\mathrm{op}} \leq \frac{\epsilon}{\lambda_{\min}(B)(\lambda_{\min}(B) - \epsilon)}.$$

## A.1 Martingale Regression in General Norms

For the following two results, we consider the martingale regression setting of [2] (referred to as the MDM setting). In particular, we consider observations of the form

$$y_{\mathsf{t}} = \langle \boldsymbol{\mu}^{\star}, \boldsymbol{z}_{\mathsf{t}} \rangle + w_{\mathsf{t}}, \tag{A.1}$$

for $y_{\mathsf{t}} \in \mathbb{R}$, unknown parameter $\boldsymbol{\mu}^{\star} \in \mathbb{R}^{d_{\boldsymbol{\mu}}}$, $w_{\mathsf{t}} \mid \mathcal{F}_{\mathsf{t}-1} \sim \mathcal{N}(0, \sigma_{\boldsymbol{w}}^2)$, and $\boldsymbol{z}_{\mathsf{t}}$ $\mathcal{F}_{\mathsf{t}-1}$-measurable, for a filtration $(\mathcal{F}_{\mathsf{t}})_{\mathsf{t}\geq 1}$. This setting therefore encompasses general stochastic processes where the observations are linear—the evolution of $\boldsymbol{z}_{\mathsf{t}}$ could be arbitrary.

We consider the setting where we interact with (A.1) for $\mathsf{T}$ steps, collecting observations $\{(\boldsymbol{z}_{\mathsf{t}}, y_{\mathsf{t}})\}_{\mathsf{t}=1}^{\mathsf{T}}$, and then form the least-squares estimate of $\boldsymbol{\mu}^{\star}$:

$$\widehat{\boldsymbol{\mu}} = \left(\sum_{\mathsf{t}=1}^{\mathsf{T}} \boldsymbol{z}_{\mathsf{t}} \boldsymbol{z}_{\mathsf{t}}^{\top}\right)^{-1} \sum_{\mathsf{t}=1}^{\mathsf{T}} \boldsymbol{z}_{\mathsf{t}} y_{\mathsf{t}}.$$

We also denote $\boldsymbol{\Sigma}_{\mathsf{T}} := \sum_{\mathsf{t}=1}^{\mathsf{T}} \boldsymbol{z}_{\mathsf{t}} \boldsymbol{z}_{\mathsf{t}}^{\top}$. The following results characterize the estimation error of $\widehat{\boldsymbol{\mu}}$ in the $M$-norm and 2-norm.

**Proposition 3** (Theorem 7.2 of [2])**.** *Fix any matrices $\boldsymbol{\Gamma} \in \mathbb{S}_{++}^{d_{\boldsymbol{\mu}}}$, $M \in \mathbb{S}_+^{d_{\boldsymbol{\mu}}}$, with $M \neq 0$. Given a parameter $\beta \in (0, 1/4)$, define the event*

$$\mathcal{E} := \{\|\boldsymbol{\Sigma}_{\mathsf{T}} - \boldsymbol{\Gamma}\|_{\mathrm{op}} \leq \beta\lambda_{\min}(\boldsymbol{\Gamma})\}.$$

*Then, if $\mathcal{E}$ holds, the following holds with probability at least $1 - \delta$:*

$$\|\widehat{\boldsymbol{\mu}} - \boldsymbol{\mu}^{\star}\|_M^2 \leq 5(1 + \zeta) \cdot \sigma_{\boldsymbol{w}}^2 \log \frac{6d_{\boldsymbol{\mu}}}{\delta} \cdot \mathrm{tr}(M\boldsymbol{\Gamma}^{-1})$$

*where $\zeta = 26\beta^2 \lambda_{\max}(\boldsymbol{\Gamma})\mathrm{tr}(\boldsymbol{\Gamma}^{-1})$.*

**Proposition 4** (Lemma E.1 of [2])**.** *On the event*

$$\mathcal{E}_{\mathrm{op}} := \{\lambda_{\min}(\boldsymbol{\Sigma}_{\mathsf{T}}) \geq \underline{\lambda}\mathsf{T}, \boldsymbol{\Sigma}_{\mathsf{T}} \preceq \mathsf{T}\bar{\boldsymbol{\Gamma}}_{\mathsf{T}}\},$$

*then we have that with probability at least $1 - \delta$:*

$$\|\widehat{\boldsymbol{\mu}} - \boldsymbol{\mu}^{\star}\|_2 \leq C \cdot \sigma_{\boldsymbol{w}} \sqrt{\frac{\log 1/\delta + d_{\boldsymbol{\mu}} + \log\det(\bar{\boldsymbol{\Gamma}}_{\mathsf{T}}/\underline{\lambda} + I)}{\underline{\lambda}\mathsf{T}}}.$$

### A.1.1 Connection Between (1.1) and (A.1)

We will apply the results Proposition 3 and Proposition 4 in the setting of (1.1) in order to obtain estimation bounds on $A_\star$. As the setting of (1.1) has vector observations, we briefly describe here how it can be mapped into the setting described above.

Recall that (1.1) evolves as

$$\boldsymbol{x}_{h+1} = A_\star \boldsymbol{\phi}(\boldsymbol{x}_h, \boldsymbol{u}_h) + \boldsymbol{w}_h, \quad h = 1, \ldots, H,$$

for $\boldsymbol{x}_h \in \mathbb{R}^{d_{\boldsymbol{x}}}$, $\boldsymbol{\phi}(\boldsymbol{x}, \boldsymbol{u}) \in \mathbb{R}^{d_{\boldsymbol{\phi}}}$, and $A_\star \in \mathbb{R}^{d_{\boldsymbol{x}} \times d_{\boldsymbol{\phi}}}$. We assume that $\boldsymbol{x}_1$ is some fixed starting state. Assume that we have run for $T$ episodes, and collected observations $\{(\boldsymbol{x}_1^t, \boldsymbol{u}_1^t, \boldsymbol{x}_2^t, \ldots, \boldsymbol{x}_h^t, \boldsymbol{u}_h^t, \boldsymbol{x}_{h+1}^t)\}_{t=1}^T$. Now let $\boldsymbol{\mu}^\star := \mathrm{vec}(A_\star)$. Furthermore, for any $t, h$, and $i \in [d_{\boldsymbol{x}}]$, let $\mathtt{t} = (t, h, i)$ and $\boldsymbol{z}_{\mathtt{t}} = [\boldsymbol{0}_{d_{\boldsymbol{\phi}}(i-1)}, \boldsymbol{\phi}(\boldsymbol{x}_h^t, \boldsymbol{u}_h^t), \boldsymbol{0}_{d_{\boldsymbol{\phi}}(d_{\boldsymbol{x}}-i)}] \in \mathbb{R}^{d_{\boldsymbol{x}} d_{\boldsymbol{\phi}}}$ where $\boldsymbol{0}_d$ denotes the zero vector of length $d$. Then we see that

$$[\boldsymbol{x}_{h+1}^t]_i = \langle \boldsymbol{\mu}^\star, \boldsymbol{z}_{\mathtt{t}} \rangle + [\boldsymbol{w}_h^t]_i.$$

Setting $y_{\mathtt{t}} = [\boldsymbol{x}_{h+1}^t]_i$ and $w_{\mathtt{t}} = [\boldsymbol{w}_h^t]_i$, it is clear that this follows the observation model of (A.1) with $d_{\boldsymbol{\mu}} = d_{\boldsymbol{\phi}} d_{\boldsymbol{x}}$ and $\mathtt{T} = d_{\boldsymbol{x}} T H$. It is also straightforward to see that the measurability assumptions of the setting of (A.1) are satisfied by this.

## B  Proof of Main Result

---

**Algorithm 3** Optimal Exploration in Nonlinear Systems (Full Version of Algorithm 1)

---

1: **inputs:** number of episodes to run $T$, cost function $(\mathrm{cost}_h)_{h=1}^H$, confidence $\delta$, control policies $\Pi^\star$, exploration policies $\Pi_{\mathrm{exp}}$
2: $\widehat{A}^1 \leftarrow$ anything, $\ell_T \leftarrow \lceil \log_2 T/8 \rceil$
3: **for** $\ell = 1, 2, 3, \ldots, \ell_T$ **do**
4: $\quad T_\ell \leftarrow 2^\ell, \delta_\ell \leftarrow \delta/12\ell^2$
5: $\quad$ Compute estimate of cost matrix: $\mathcal{H}_\ell \leftarrow \mathcal{H}(\widehat{A}^\ell)$
6: $\quad \Pi_\ell \leftarrow \mathrm{LEARNEXP}\Pi(\mathcal{H}_\ell, T_\ell, \delta_\ell, \mathbb{A}_{\mathcal{R}}, \Pi_{\mathrm{exp}})$ (Algorithm 7), with $\mathbb{A}_{\mathcal{R}}$ the LC$^3$ algorithm [17]
7: $\quad$ Rerun each policy in $\Pi_\ell$ $N_\ell = \lceil T_\ell/|\Pi_\ell| \rceil$ times, denote collected data $\mathfrak{D}_\ell$
8: $\quad$ Estimate system parameters
9:

$$\widehat{A}^{\ell+1} = \arg\min_A \sum_{h=1}^H \sum_{(\boldsymbol{x}_{h+1}, \boldsymbol{u}_h, \boldsymbol{x}_h) \in \mathfrak{D}_\ell} \|\boldsymbol{x}_{h+1} - A\boldsymbol{\phi}(\boldsymbol{x}_h, \boldsymbol{u}_h)\|_2^2$$

10: **return** $\widehat{\pi}_T \leftarrow \pi_\star(\widehat{A}^{\ell_T+1})$

---

**Theorem 4** (Full Version of Theorem 1). *Assume Assumptions 1 to 5 and 13 hold. Then if*

$$T \geq \mathrm{poly}\left(d_{\boldsymbol{\phi}}, d_{\boldsymbol{x}}, H, B_A, B_{\boldsymbol{\phi}}, L_{\boldsymbol{\phi}}, L_{\boldsymbol{\theta}}, L_{\mathrm{cost}}, L_{\pi_\star}, \sigma_{\boldsymbol{w}}, \sigma_{\boldsymbol{w}}^{-1}, \frac{1}{\lambda_{\min}^\star}, \log\frac{T}{\delta}\right) \cdot \max\left\{1, \frac{1}{r_{\mathrm{cost}}(A_\star)^2}, \frac{1}{r_{\boldsymbol{\theta}}(A_\star)^2}\right\},$$
$\tag{B.1}$

*with probability at least $1 - \delta$, Algorithm 3 plays exploration policies $\pi_{\mathrm{exp}} \in \Pi_{\mathrm{exp}}$ at every episode, runs for at most $T$ episodes, and the controller $\widehat{\pi}_T$ returned Algorithm 3 satisfies, with probability at least $1 - \delta$:*

$$\mathcal{J}(\widehat{\pi}_T; A_\star) - \mathcal{J}(\pi_\star(A_\star); A_\star) \leq \frac{\sigma_{\boldsymbol{w}}^2}{T} \cdot \min_{\boldsymbol{\Lambda} \in \boldsymbol{\Omega}} \mathrm{tr}\left(\mathcal{H}(A_\star)\check{\boldsymbol{\Lambda}}^{-1}\right) \cdot C \log\frac{6d_{\boldsymbol{x}}d_{\boldsymbol{\phi}}}{\delta} + \frac{C_{\mathrm{lot}}}{T^{3/2}}$$

*for $C$ a universal constant and*

$$C_{\mathrm{poly}} := \mathrm{poly}\left(d_{\boldsymbol{\phi}}, d_{\boldsymbol{x}}, H, B_A, B_{\boldsymbol{\phi}}, L_{\boldsymbol{\phi}}, L_{\boldsymbol{\theta}}, L_{\mathrm{cost}}, L_{\pi_\star}, \sigma_{\boldsymbol{w}}, \sigma_{\boldsymbol{w}}^{-1}, \frac{1}{\lambda_{\min}^\star}, \log\frac{T}{\delta}\right).$$

*Proof.* Let $\mathcal{E}_\ell$ denote that the good event of Lemma B.3 holds at round $\ell$ which, by Lemma B.3, occurs with probability at least $1 - 6\delta_\ell$. By our setting of $\delta_\ell = \delta/12\ell^2$, we have that the total failure

probability of $\mathcal{E}_\ell$ for all $\ell$ is bounded as

$$\sum_{\ell=1}^{\infty} 6 \cdot \frac{\delta}{12\ell^2} \leq \delta.$$

Henceforth we assume that $\mathcal{E} := \cap_\ell \mathcal{E}_\ell$ holds. Let $\widehat{A} := \widehat{A}^{\ell_T+1}$, and $\widehat{A}^- := \widehat{A}^{\ell_T}$.

Before proceeding to the main proof, we note that the conclusion that Algorithm 3 only explores with policies in $\Pi_{\mathrm{exp}}$ follows from the definition of LEARNEXPΠ and LC$^3$. Note that LEARNEXPΠ only interacts with (1.1) through calls to DYNAMICOED, which itself only interacts with (1.1) through calls to $\mathbb{A}_\mathcal{R}$, instantiated in Algorithm 3 by LC$^3$. Inspection of the LC$^3$ algorithm in [17] reveals that LC$^3$ only interacts with (1.1) by playing policies in $\Pi_{\mathrm{exp}}$, from which the conclusion follows.

**Bounding the Number of Episodes.** Denote $T_\ell^{\mathrm{oed}} = |\Pi_\ell|$. Note that by construction we always have $N_\ell T_\ell^{\mathrm{oed}} \geq T_\ell$. By Lemma B.2, as long as (B.1) is met, we can bound the total number of episodes collected up to and including round $\ell$ by $4T_\ell$ for $\ell \in \{\ell_T, \ell_T - 1\}$. We therefore, in the following, will make use of the fact that

$$c \cdot T \leq N_\ell K_\ell \leq c' \cdot T$$

for $\ell \in \{\ell_T, \ell_T - 1\}$ and absolute constants $c, c'$. Furthermore, it also follows from this that the total number of episodes run by Algorithm 3 is bounded by

$$4T_{\ell_T} = 4 \cdot 2^{\lceil \log T/8 \rceil} \leq 8 \cdot \frac{T}{8} = T,$$

so Algorithm 3 runs for at most $T$ episodes.

**Approximating the Controller Loss.** Let $r_{\mathrm{est}}(A_\star) := \min\{1, r_{\mathrm{cost}}(A_\star), r_{\boldsymbol{\theta}}(A_\star)\}$, for $r_{\mathrm{cost}}(A_\star)$ as in Assumption 2 and $r_{\boldsymbol{\theta}}(A_\star)$ as in Assumption 13. By Lemma D.2, under Assumptions 1, 2, 4, 5 and 13, as long as $\widehat{A} \in \mathcal{B}_{\mathrm{F}}(A_\star; r_{\mathrm{est}}(A_\star))$, we have

$$\mathcal{J}(\widehat{\pi}_T; A_\star) - \mathcal{J}(\pi_\star(A_\star); A_\star) \leq \|\mathrm{vec}(\widehat{A} - A_\star)\|^2_{\mathcal{H}(A_\star)}$$
$$+ \mathrm{poly}(L_{\pi_\star}, \|A_\star\|_{\mathrm{op}}, L_{\boldsymbol{\phi}}, L_{\boldsymbol{\theta}}, L_{\mathrm{cost}}, \sigma_{\boldsymbol{w}}^{-1}, H, d_{\boldsymbol{x}}) \cdot \|\widehat{A} - A_\star\|^3_{\mathrm{op}}.$$

Furthermore, we can bound

$$\|\mathrm{vec}(\widehat{A} - A_\star)\|^2_{\mathcal{H}(A_\star)} = \|\mathrm{vec}(\widehat{A} - A_\star)\|^2_{\mathcal{H}(\widehat{A}^-)} + \mathrm{vec}(\widehat{A} - A_\star)^\top (\mathcal{H}(A_\star) - \mathcal{H}(\widehat{A}^-))\mathrm{vec}(\widehat{A} - A_\star)$$
$$\leq (\widehat{A} - A_\star)^\top \mathcal{H}(\widehat{A}^-)(\widehat{A} - A_\star) + \|\widehat{A}^- - A_\star\|^2_{\mathrm{op}}\|\mathcal{H}(A_\star) - \mathcal{H}(\widehat{A}^-)\|_{\mathrm{op}}.$$

**Bounding the Hessian Estimation Error.** On $\mathcal{E}$, by Lemma B.3 and as long as (B.1) is met, we have (note that at the final epoch, the plug-in estimator $\mathcal{H}(\widehat{A}^-)$ is given as input to DYNAMICOED):

$$\|\mathrm{vec}(\widehat{A} - A_\star)\|^2_{\mathcal{H}(\widehat{A}^-)} \leq \frac{60}{N_{\ell_T} T_{\ell_T}^{\mathrm{oed}}} \cdot \min_{\boldsymbol{\Lambda} \in \boldsymbol{\Omega}} \mathrm{tr}\left(\mathcal{H}(\widehat{A}^-)\check{\boldsymbol{\Lambda}}^{-1}\right) \cdot \sigma_{\boldsymbol{w}}^2 \log \frac{6d_{\boldsymbol{x}}d_{\boldsymbol{\phi}}}{\delta}$$
$$+ \mathrm{poly}\left(d_{\boldsymbol{\phi}}, d_{\boldsymbol{x}}, \frac{1}{\lambda_{\min}^\star}, B_A, B_{\boldsymbol{\phi}}, \log \frac{1}{\sigma_{\boldsymbol{w}}}, H, \|\mathcal{H}(\widehat{A}^-)\|_{\mathrm{op}}, \log \frac{T_{\ell_T}}{\delta}\right) \cdot \frac{1}{N_{\ell_T}^2}$$
$$\leq \frac{C}{T} \cdot \min_{\boldsymbol{\Lambda} \in \boldsymbol{\Omega}} \mathrm{tr}\left(\mathcal{H}(\widehat{A}^-)\check{\boldsymbol{\Lambda}}^{-1}\right) \cdot \sigma_{\boldsymbol{w}}^2 \log \frac{6d_{\boldsymbol{x}}d_{\boldsymbol{\phi}}}{\delta}$$
$$+ \mathrm{poly}\left(d_{\boldsymbol{\phi}}, d_{\boldsymbol{x}}, \frac{1}{\lambda_{\min}^\star}, B_A, B_{\boldsymbol{\phi}}, \log \frac{1}{\sigma_{\boldsymbol{w}}}, H, \|\mathcal{H}(\widehat{A}^-)\|_{\mathrm{op}}, \log \frac{T}{\delta}\right) \cdot \frac{1}{T^2}$$

where the last line uses that $N_{\ell_T} T_{\ell_T}^{\mathrm{oed}}$ is within a constant of $T$, and that $T_{\ell_T}^{\mathrm{oed}}$ can be bounded by

$$\mathrm{poly}\left(d_{\boldsymbol{\phi}}, d_{\boldsymbol{x}}, \frac{1}{\lambda_{\min}^\star}, B_A, B_{\boldsymbol{\phi}}, \log \frac{1}{\sigma_{\boldsymbol{w}}}, H, \log \frac{T}{\delta}\right)$$

by Lemma B.3. By Lemma B.4 we can bound

$$\min_{\boldsymbol{\Lambda} \in \boldsymbol{\Omega}} \mathrm{tr}\left(\mathcal{H}(\widehat{A}^-)\check{\boldsymbol{\Lambda}}^{-1}\right) \leq \min_{\boldsymbol{\Lambda} \in \boldsymbol{\Omega}} 2\mathrm{tr}\left(\mathcal{H}(A_\star)\check{\boldsymbol{\Lambda}}^{-1}\right) + \frac{2d_{\boldsymbol{x}}d_{\boldsymbol{\phi}}}{\lambda_{\min}^\star} \cdot \|\mathcal{H}(A_\star) - \mathcal{H}(\widehat{A}^-)\|_{\mathrm{op}}$$

and by Lemma D.3, under Assumptions 1, 2, 4, 5 and 13, and as long as $\widehat{A}^- \in \mathcal{B}_F(A_\star; r_{est}(A_\star))$, we can bound

$$\|\mathcal{H}(A_\star) - \mathcal{H}(\widehat{A}^-)\|_{op} \le poly(L_{\pi_\star}, \|A_\star\|_{op}, L_\phi, L_\theta, L_{cost}, \sigma_w^{-1}, H, d_x) \cdot \|\widehat{A}^- - A_\star\|_{op}.$$

Let

$$C_{lot} := poly\left(L_{\pi_\star}, B_A, B_\phi, L_\phi, L_\theta, L_{cost}, d_x, d_\phi, \frac{1}{\lambda_{min}^\star}, \sigma_w, \sigma_w^{-1}, H, \|\mathcal{H}(A_\star)\|_{op}, \log\frac{T}{\delta}\right)$$

denote some lower-order constant, whose precise polynomial dependence may change from line to line. On $\mathcal{E}$, by Lemma B.3 we can bound

$$\|\widehat{A} - A_\star\|_{op}, \|\widehat{A}^- - A_\star\|_{op} \le C_{lot} \cdot \frac{1}{\sqrt{T}}, \tag{B.2}$$

and so assuming the burn-in (B.1) is met, we can bound $\|\widehat{A}^- - A_\star\|_{op} \le \frac{1}{2}r_{est}(A_\star)$ and $\|\widehat{A} - A_\star\|_{op} \le \frac{1}{2}r_{est}(A_\star)$. This then implies that

$$\|\mathcal{H}(A_\star) - \mathcal{H}(\widehat{A}^-)\|_{op} \le C_{lot} \cdot \frac{1}{\sqrt{T}},$$

so in particular we can bound

$$poly\left(d_\phi, d_x, \frac{1}{\lambda_{min}^\star}, B_A, B_\phi, H, \log\frac{1}{\sigma_w}, \|\mathcal{H}(\widehat{A}^-)\|_{op}, \log\frac{T}{\delta}\right)$$
$$\le poly\left(d_\phi, d_x, \frac{1}{\lambda_{min}^\star}, B_A, B_\phi, H, \log\frac{1}{\sigma_w}, \|\mathcal{H}(A_\star)\|_{op}, \log\frac{T}{\delta}\right).$$

We have therefore shown that

$$\mathcal{J}(\widehat{\pi}_T; A_\star) - \mathcal{J}(\pi_\star(A_\star); A_\star) \le \frac{C}{T} \cdot \min_{\boldsymbol{\Lambda}\in\boldsymbol{\Omega}} tr\left(\mathcal{H}(A_\star)\check{\boldsymbol{\Lambda}}^{-1}\right) \cdot \sigma_w^2 \log\frac{6d_x d_\phi}{\delta}$$
$$+ C_{lot} \cdot \left(\|\widehat{A} - A_\star\|_{op}^3 + \|\widehat{A}^- - A_\star\|_{op}^3 + \frac{1}{T^2} + \frac{1}{T} \cdot \|\widehat{A}^- - A_\star\|_{op}\right)$$
$$\le \frac{C}{T} \cdot \min_{\boldsymbol{\Lambda}\in\boldsymbol{\Omega}} tr\left(\mathcal{H}(A_\star)\check{\boldsymbol{\Lambda}}^{-1}\right) \cdot \sigma_w^2 \log\frac{6d_x d_\phi}{\delta} + \frac{C_{lot}}{T^{3/2}},$$

The final result follows from using Lemma D.4 to bound

$$\|\mathcal{H}(A_\star)\|_{op} \le poly(\|A_\star\|_{op}, B_\phi, L_\phi, L_\theta, L_{cost}, L_{\pi_\star}, \sigma_w^{-1}, H, d_x).$$

$\square$

*Proof of Theorem 1.* The proof of Theorem 1 is identical to that of Theorem 4, the only difference being that we replace Assumption 13 with Assumption 6. However, by Proposition 6, the conditions of Assumption 13 are met when Assumption 6 holds. $\square$

## B.1 Supporting Lemmas

**Lemma B.1.** *Under Assumptions 1 and 3, the system* (1.1) *satisfies Assumptions 11 and 12 with*

$$\boldsymbol{\psi}(\boldsymbol{\tau}) \leftarrow I_{d_x} \otimes \sum_{h=1}^{H} \boldsymbol{\phi}(\boldsymbol{x}_h^{\boldsymbol{\tau}}, \boldsymbol{u}_h^{\boldsymbol{\tau}})\boldsymbol{\phi}(\boldsymbol{x}_h^{\boldsymbol{\tau}}, \boldsymbol{u}_h^{\boldsymbol{\tau}})^\top, \quad D = d_x H B_\phi^2,$$

$d_\psi \leftarrow d_\phi d_x$, *and where $\boldsymbol{x}_h^{\boldsymbol{\tau}}$ (resp. $\boldsymbol{u}_h^{\boldsymbol{\tau}}$) denotes the state (resp. input) at step h of trajectory $\boldsymbol{\tau}$. Furthermore, it satisfies Assumption 10 with $\mathbb{A}_\mathcal{R}$ instantiated with the $LC^3$ algorithm of [17] and*

$$C_\mathcal{R} = C \cdot H\sqrt{d_\phi(d_\phi + d_x + B_A)} \cdot \log(1 + B_\phi H/\sigma_w), \quad p_\mathcal{R} = 3/2, \quad \alpha = 1/2$$

*for a universal constant C.*

*Proof.* That Assumption 12 is satisfied is immediate under Assumption 3. It is clear that $\psi(\boldsymbol{\tau}) \in \mathcal{S}_+^{d_\phi d_x}$. To obtain a bound on $D$, we only need to bound the trace of $\psi(\boldsymbol{\tau})$:

$$\mathrm{tr}(\psi(\boldsymbol{\tau})) = \sum_{h=1}^{H} \mathrm{tr}(I_{d_x}) \cdot \mathrm{tr}(\phi(\boldsymbol{x}_h^{\boldsymbol{\tau}}, \boldsymbol{u}_h^{\boldsymbol{\tau}})\phi(\boldsymbol{x}_h^{\boldsymbol{\tau}}, \boldsymbol{u}_h^{\boldsymbol{\tau}})^\top)$$

$$= d_x \sum_{h=1}^{H} \|\phi(\boldsymbol{x}_h^{\boldsymbol{\tau}}, \boldsymbol{u}_h^{\boldsymbol{\tau}})\|_2^2$$

$$\leq d_x H B_\phi^2$$

where the inequality holds under Assumption 1.

To show that Assumption 10 is satisfied in this setting, we have by Theorem 6 that with probability at least $1 - \delta$, $\mathrm{LC}^3$ has regret bounded as (using that $c_{\max} \leq 1$ in the setting of Assumption 10):

$$\mathcal{R}_T \leq C \cdot H \sqrt{d_\phi \cdot (d_\phi + d_x + B_A + \log\frac{1}{\delta}) \cdot T} \cdot \log(1 + B_\phi H T/\sigma_{\boldsymbol{w}})$$

for $C$ a universal constant. We can therefore take $\alpha = 1/2$, $p_\mathcal{R} = 3/2$, and

$$C_\mathcal{R} = C' \cdot H \sqrt{d_\phi(d_\phi + d_x + B_A)} \cdot \log(1 + B_\phi H/\sigma_{\boldsymbol{w}}).$$

$\square$

**Lemma B.2.** *Let $\bar{T}_\ell$ denote the total number of episodes collected by Algorithm 3 at round $\ell$. For*

$$T_\ell \geq \mathrm{poly}\left(d_\phi, d_x, \frac{1}{\lambda_{\min}^\star}, B_A, B_\phi, \log\frac{1}{\sigma_{\boldsymbol{w}}}, H, \log\frac{T_\ell}{\delta}\right), \tag{B.3}$$

*on the success event of Lemma B.3, we have $2T_\ell \geq \bar{T}_\ell$ and*

$$2T_\ell \geq \sum_{i=1}^{\ell-1} \bar{T}_i.$$

*Proof.* Recall that $T_\ell = 2^\ell$ and $N_\ell = \lceil T_\ell/T_\ell^{\mathrm{oed}} \rceil$ for $T_\ell^{\mathrm{oed}} = |\Pi_\ell|$. By Lemma B.3, $\bar{T}_\ell$ can be bounded as

$$\bar{T}_\ell \leq N_\ell T_\ell^{\mathrm{oed}} + (16 + 2\log T_\ell^{\mathrm{oed}})T_\ell^{\mathrm{oed}}$$

and $T_\ell^{\mathrm{oed}}$ can be bounded as

$$T_\ell^{\mathrm{oed}} \leq \mathrm{poly}\left(d_\phi, d_x, \frac{1}{\lambda_{\min}^\star}, B_A, B_\phi, \log\frac{1}{\sigma_{\boldsymbol{w}}}, H, \log\frac{T_\ell}{\delta}\right). \tag{B.4}$$

Note that we can bound

$$\bar{T}_i \leq N_i T_i^{\mathrm{oed}} + (16 + 2\log T_i^{\mathrm{oed}})T_i^{\mathrm{oed}}$$

$$\leq T_i + (17 + 2\log T_i^{\mathrm{oed}})T_i^{\mathrm{oed}}$$

$$\leq T_i + \mathrm{poly}\left(d_\phi, d_x, \frac{1}{\lambda_{\min}^\star}, B_A, B_\phi, \log\frac{1}{\sigma_{\boldsymbol{w}}}, H, \log\frac{T_\ell}{\delta}\right).$$

From this it is immediately obvious that $2T_\ell \geq \bar{T}_\ell$ as long as (B.3) is satisfied.

To show the second conclusion note that, by our choice of $T_i = 2^i$, we have that $T_\ell \geq \sum_{i=1}^{\ell-1} T_i$, so it therefore remains to show that

$$T_\ell \geq \sum_{i=1}^{\ell-1} \mathrm{poly}\left(d_\phi, d_x, \frac{1}{\lambda_{\min}^\star}, B_A, B_\phi, \log\frac{1}{\sigma_{\boldsymbol{w}}}, H, \log\frac{T_i}{\delta}\right).$$

However, we can bound

$$\sum_{i=1}^{\ell-1} \mathrm{poly}\left(d_{\boldsymbol{\phi}}, d_{\boldsymbol{x}}, \frac{1}{\lambda_{\min}^{\star}}, B_A, B_{\boldsymbol{\phi}}, \log\frac{1}{\sigma_{\boldsymbol{w}}}, H, \log\frac{T_i}{\delta}\right) \le \mathrm{poly}\left(d_{\boldsymbol{\phi}}, d_{\boldsymbol{x}}, \frac{1}{\lambda_{\min}^{\star}}, B_A, B_{\boldsymbol{\phi}}, \log\frac{1}{\sigma_{\boldsymbol{w}}}, H, \log\frac{T_\ell}{\delta}\right) \cdot \log T_\ell,$$

so a sufficient condition is

$$T_\ell \ge \mathrm{poly}\left(d_{\boldsymbol{\phi}}, d_{\boldsymbol{x}}, \frac{1}{\lambda_{\min}^{\star}}, B_A, B_{\boldsymbol{\phi}}, \log\frac{1}{\sigma_{\boldsymbol{w}}}, H, \log\frac{T_\ell}{\delta}\right) \cdot \log T_\ell$$

which we see is met when (B.3) holds. $\qquad\square$

**Lemma B.3.** *Consider running Algorithm 7 with weight matrix $\mathcal{H}$, parameter $\widetilde{N}$, and confidence $\delta$, and rerunning each policy in $\Pi_{\mathrm{out}}$ $N \le \widetilde{N}$ times. Then, under Assumptions 1 and 3, with probability at least $1 - 6\delta$:*

$$\|\mathrm{vec}(\widehat{A} - A_\star)\|_{\mathcal{H}}^2 \le \frac{60}{NT_{\mathrm{out}}} \cdot \min_{\boldsymbol{\Lambda}\in\boldsymbol{\Omega}} \mathrm{tr}\left(\mathcal{H}\check{\boldsymbol{\Lambda}}^{-1}\right) \cdot \sigma_{\boldsymbol{w}}^2 \log\frac{6d_{\boldsymbol{x}}d_{\boldsymbol{\phi}}}{\delta}$$

$$+ \mathrm{poly}\left(d_{\boldsymbol{\phi}}, d_{\boldsymbol{x}}, \frac{1}{\lambda_{\min}^{\star}}, B_A, B_{\boldsymbol{\phi}}, \log\frac{1}{\sigma_{\boldsymbol{w}}}, H, \|\mathcal{H}\|_{\mathrm{op}}, \log\frac{\widetilde{N}}{\delta}\right) \cdot \frac{1}{N^2}$$

*where $\widehat{A}$ denotes the least-squares estimate of $A_\star$ obtained on the data generated by rerunning $\Pi_{\mathrm{out}}$. In addition, we have*

$$\|\widehat{A} - A_\star\|_{\mathrm{F}} \le \mathrm{poly}\left(d_{\boldsymbol{\phi}}, d_{\boldsymbol{x}}, \frac{1}{\lambda_{\min}^{\star}}, B_A, B_{\boldsymbol{\phi}}, \sigma_{\boldsymbol{w}}, H, \log\frac{\widetilde{N}}{\delta}\right) \cdot \frac{1}{\sqrt{N}}.$$

*Furthermore, we have*

$$T_{\mathrm{out}} \le \mathrm{poly}\left(d_{\boldsymbol{\phi}}, d_{\boldsymbol{x}}, \frac{1}{\lambda_{\min}^{\star}}, B_A, B_{\boldsymbol{\phi}}, \log\frac{1}{\sigma_{\boldsymbol{w}}}, H, \log\frac{\widetilde{N}}{\delta}\right)$$

*and the total number of episodes collected by this procedure is bounded by $NT_{\mathrm{out}} + (16 + 2\log(T_{\mathrm{out}})) \cdot T_{\mathrm{out}}$, for $T_{\mathrm{out}} = |\Pi_{\mathrm{out}}|$.*

*Proof.* By Lemma B.1, the assumptions of Lemma C.6 and Lemma C.7 are met, so we can therefore apply these results in our setting. By Lemma C.6, the event $\mathcal{E}_{\mathrm{exp}}$ occurs with probability at least $1 - \delta$. Throughout the remainder of the proof we union bound over the success event of Lemma C.7 and $\mathcal{E}_{\mathrm{exp}}$, which together occur with probability at least $1 - 4\delta$.

Let

$$\widetilde{\boldsymbol{\Lambda}} := \sum_{t=1}^{NT_{\mathrm{out}}} \boldsymbol{\psi}(\boldsymbol{\tau}^t) = I_{d_{\boldsymbol{x}}} \otimes \sum_{t=1}^{NT_{\mathrm{out}}} \sum_{h=1}^{H} \boldsymbol{\phi}(\boldsymbol{x}_h^t, \boldsymbol{u}_h^t)\boldsymbol{\phi}(\boldsymbol{x}_h^t, \boldsymbol{u}_h^t)^\top$$

denote the features returned by rerunning every policy in $\Pi_{\mathrm{out}}$ $N$ times. By Lemma C.7, we then have that:

$$\left\|\widetilde{\boldsymbol{\Lambda}} - N\cdot\textstyle\sum_{\pi\in\Pi_{\mathrm{out}}}\check{\boldsymbol{\Lambda}}_\pi\right\|_{\mathrm{op}} \le \underbrace{\frac{\sqrt{T_{\mathrm{out}}}\cdot\sqrt{8d_{\boldsymbol{\phi}}d_{\boldsymbol{x}}\log(1 + 8\sqrt{NT_{\mathrm{out}}}) + 8\log 1/\delta}}{\sqrt{N}\cdot 6272d_{\boldsymbol{\phi}}d_{\boldsymbol{x}}\log\frac{68\widetilde{N}}{\delta}}}_{=:\beta}\cdot\lambda_{\min}\left(N\cdot\textstyle\sum_{\pi\in\Pi_{\mathrm{out}}}\check{\boldsymbol{\Lambda}}_\pi\right)\cdot$$

Applying Proposition 3 with $\mathcal{E}$ the event that the above conclusion holds and $\boldsymbol{\Gamma} := N\cdot\sum_{\pi\in\Pi_{\mathrm{out}}}\check{\boldsymbol{\Lambda}}_\pi$, we obtain that, with probability at least $1 - \delta$ (using the mapping to the martingale regression setting described in Appendix A.1.1):

$$\|\mathrm{vec}(\widehat{A} - A_\star)\|_{\mathcal{H}}^2 \le 5(1 + \zeta)\cdot\sigma_{\boldsymbol{w}}^2\log\frac{6d_{\boldsymbol{x}}d_{\boldsymbol{\phi}}}{\delta}\cdot\mathrm{tr}(\mathcal{H}\boldsymbol{\Gamma}^{-1})$$

for $\zeta = 26\beta^2 \lambda_{\max}(\mathbf{\Gamma})\mathrm{tr}(\mathbf{\Gamma}^{-1})$ and $\beta$ as defined above. Since $\|\phi(x, u)\|_2 \le B_\phi$ under Assumption 1, we have $\|\check{\mathbf{\Lambda}}_\pi\|_2 \le B_\phi^2$, so we can upper bound $\lambda_{\max}(\mathbf{\Gamma}) \le NT_{\mathrm{out}}B_\phi^2$. By Lemma C.7, we can also bound (using that $D = d_x H B_\phi^2$ by Lemma B.1):

$$\mathrm{tr}(\mathbf{\Gamma}^{-1}) \le \frac{1}{N \cdot 6272 d_x H B_\phi^2 \log \frac{68\widetilde{N}}{\delta}}.$$

Combining these and using that

$$T_{\mathrm{out}} \le \mathrm{poly}\left( d_\phi, d_x, \frac{1}{\lambda_{\min}^\star}, B_A, B_\phi, \log \frac{1}{\sigma_w}, H, \log \frac{\widetilde{N}}{\delta} \right) \tag{B.5}$$

as shown in Lemma C.8 (and using our bounds on $C_{\mathcal{R}}$ and $p_{\mathcal{R}}$ in Lemma B.1), we can therefore bound

$$\zeta \le \mathrm{poly}\left( d_\phi, d_x, \frac{1}{\lambda_{\min}^\star}, B_A, B_\phi, \log \frac{1}{\sigma_w}, H, \log \frac{\widetilde{N}}{\delta} \right) \cdot \frac{1}{N}.$$

Using that $\mathrm{tr}(\mathcal{H}\mathbf{\Gamma}^{-1}) \le \|\mathcal{H}\|_{\mathrm{op}} \cdot \mathrm{tr}(\mathbf{\Gamma}^{-1})$, and the bound on $\mathrm{tr}(\mathbf{\Gamma}^{-1})$ given above, it follows that

$$\|\mathrm{vec}(\widehat{A} - A_\star)\|_{\mathcal{H}}^2 \le 5\sigma_w^2 \log \frac{6 d_x d_\phi}{\delta} \cdot \mathrm{tr}(\mathcal{H}\mathbf{\Gamma}^{-1}) + \mathrm{poly}\left( d_\phi, d_x, \frac{1}{\lambda_{\min}^\star}, B_A, B_\phi, \log \frac{1}{\sigma_w}, H, \|\mathcal{H}\|_{\mathrm{op}}, \log \frac{\widetilde{N}}{\delta} \right) \cdot \frac{1}{N^2}.$$

Finally, by Lemma C.7, we can bound

$$\mathrm{tr}(\mathcal{H}\mathbf{\Gamma}^{-1}) \le \frac{12}{NT_{\mathrm{out}}} \cdot \min_{\check{\mathbf{\Lambda}}\in\mathbf{\Omega}} \mathrm{tr}(\mathcal{H}\check{\mathbf{\Lambda}}^{-1}).$$

By Lemma C.8, we can bound the total number of episodes collected by Algorithm 7 by $(16 + 2\log(T_{\mathrm{out}})) \cdot T_{\mathrm{out}}$.

**Bound on Frobenius Norm Error.** By Lemma C.7, we can lower bound

$$\lambda_{\min}(\widetilde{\mathbf{\Lambda}}) \ge N \cdot 6272 d_x d_\phi H B_\phi^2 \log \frac{68\widetilde{N}}{\delta}.$$

Furthermore, since $\|\phi(x, u)\|_2 \le B_\phi$, we always have $\|\widetilde{\mathbf{\Lambda}}\|_{\mathrm{op}} \le NT_{\mathrm{out}}B_\phi^2$, which implies $\widetilde{\mathbf{\Lambda}} \preceq NT_{\mathrm{out}}B_\phi^2 \cdot I$. By Proposition 4, we then have that with probability at least $1 - \delta$ (again using the mapping to the martingale regression setting described in Appendix A.1.1):

$$\|\widehat{A} - A_\star\|_{\mathrm{F}} \le C \cdot \sigma_w \sqrt{\frac{\log 1/\delta + d_x d_\phi + \log\det(\frac{T_{\mathrm{out}}}{6272 d_x d_\phi H \log \frac{68\widetilde{N}}{\delta}} \cdot I + I)}{N \cdot 6272 d_x d_\phi H B_\phi^2 \log \frac{68\widetilde{N}}{\delta}}}$$

$$\le \mathrm{poly}\left( d_\phi, d_x, \frac{1}{\lambda_{\min}^\star}, B_A, B_\phi, \sigma_w, H, \log \frac{\widetilde{N}}{\delta} \right) \cdot \frac{1}{\sqrt{N}}.$$

$\square$

**Lemma B.4.** *Under Assumption 3, for any $\mathcal{H}, \mathcal{H}'$, we can bound*

$$\min_{\check{\mathbf{\Lambda}}\in\mathbf{\Omega}} \mathrm{tr}(\mathcal{H}\check{\mathbf{\Lambda}}^{-1}) \le \min_{\check{\mathbf{\Lambda}}\in\mathbf{\Omega}} 2\mathrm{tr}(\mathcal{H}'\check{\mathbf{\Lambda}}^{-1}) + \frac{2 d_x d_\phi}{\lambda_{\min}^\star} \cdot \|\mathcal{H} - \mathcal{H}'\|_{\mathrm{op}}.$$

*Proof.* We have

$$\min_{\check{\mathbf{\Lambda}}\in\mathbf{\Omega}} \mathrm{tr}(\mathcal{H}\check{\mathbf{\Lambda}}^{-1}) = \min_{\check{\mathbf{\Lambda}}\in\mathbf{\Omega}} \mathrm{tr}(\mathcal{H}'\check{\mathbf{\Lambda}}^{-1}) + \mathrm{tr}((\mathcal{H} - \mathcal{H}')\check{\mathbf{\Lambda}}^{-1})$$

$$\le \min_{\check{\mathbf{\Lambda}}\in\mathbf{\Omega}} \mathrm{tr}(\mathcal{H}'\check{\mathbf{\Lambda}}^{-1}) + \|\mathcal{H} - \mathcal{H}'\|_{\mathrm{op}} \cdot \mathrm{tr}(\check{\mathbf{\Lambda}}^{-1})$$

Under Assumption 3, we know that there exists some $\check{\Lambda}' \in \Omega$ such that $\lambda_{\min}(\check{\Lambda}') \geq \lambda_{\min}^\star$. We can then bound

$$\min_{\check{\Lambda} \in \Omega} \text{tr}(\mathcal{H}'\check{\Lambda}^{-1}) + \|\mathcal{H} - \mathcal{H}'\|_{\text{op}} \cdot \text{tr}(\check{\Lambda}^{-1})$$

$$\leq \min_{\check{\Lambda} \in \Omega} \text{tr}(\mathcal{H}'(\frac{1}{2}\check{\Lambda} + \frac{1}{2}\check{\Lambda}')^{-1}) + \|\mathcal{H} - \mathcal{H}'\|_{\text{op}} \cdot \text{tr}((\frac{1}{2}\check{\Lambda} + \frac{1}{2}\check{\Lambda}')^{-1})$$

$$\leq \min_{\check{\Lambda} \in \Omega} 2\text{tr}(\mathcal{H}'\check{\Lambda}^{-1}) + 2\|\mathcal{H} - \mathcal{H}'\|_{\text{op}} \cdot \frac{d_{\boldsymbol{x}}d_{\boldsymbol{\phi}}}{\lambda_{\min}^\star}$$

which proves the result. $\qquad\square$

## C   Experiment Design in Arbitrary Dynamical Systems

In this section we generalize somewhat the setting of Section 5. In particular, our goal will now be to collect some set of trajectories $\mathfrak{D} = \{\boldsymbol{\tau}_t\}_{t=1}^T$, which minimize

$$\Phi\left(\frac{1}{T}\sum_{t=1}^T \boldsymbol{\psi}(\boldsymbol{\tau}_t)\right)$$

for some general feature mapping $\boldsymbol{\psi} : \mathcal{T} \to \mathbb{R}^d$, $\Phi : \mathbb{R}^d \to \mathbb{R}$, and $\mathcal{T} = (\mathcal{X} \times \mathcal{U})^H \times \mathcal{X}$ the space of possible state-input trajectories, $\boldsymbol{\tau} = (\boldsymbol{x}_0, \boldsymbol{u}_0, \boldsymbol{x}_1, \ldots, \boldsymbol{x}_{H-1}, \boldsymbol{u}_{H-1}, \boldsymbol{x}_H) \in \mathcal{T}$. We will assume that $\boldsymbol{\psi}$ can be decomposed additively as

$$\boldsymbol{\psi}(\boldsymbol{\tau}) = \sum_{h=1}^H \boldsymbol{\psi}_h(\boldsymbol{x}_h, \boldsymbol{u}_h).$$

In Section 5 we considered the special case where $\boldsymbol{\psi}_h(\boldsymbol{x}, \boldsymbol{u}) = \boldsymbol{\phi}(\boldsymbol{x}, \boldsymbol{u})\boldsymbol{\phi}(\boldsymbol{x}, \boldsymbol{u})^\top$; in this section $\boldsymbol{\psi}$ could instead be any arbitrary mapping.

As before, we will be interested in defining optimal exploration with respect to some set of exploration policies, $\Pi_{\text{exp}}$. Let

$$\boldsymbol{\Omega}_{\boldsymbol{\psi}} := \{\mathbb{E}_{\pi \sim \omega}[\mathbb{E}_\pi[\boldsymbol{\psi}(\boldsymbol{\tau})]] \; : \; \omega \in \triangle_\Pi\}$$

denote the space of expected value of $\boldsymbol{\psi}(\boldsymbol{\tau})$ for mixtures of policies in $\Pi_{\text{exp}}$. To distinguish elements $\boldsymbol{\Lambda} \in \boldsymbol{\Omega}$ from elements in $\boldsymbol{\Omega}_{\boldsymbol{\psi}}$, we will let $\boldsymbol{\Gamma} = \mathbb{E}_{\pi \sim \omega}[\mathbb{E}_\pi[\boldsymbol{\psi}(\boldsymbol{\tau})]]$ refer to elements of $\boldsymbol{\Omega}_{\boldsymbol{\psi}}$, and in particular define $\boldsymbol{\Gamma}_\pi := \mathbb{E}_\pi[\boldsymbol{\psi}(\boldsymbol{\tau})]$ (where it is assumed that the expectation is collected over trajectories on (5.1)). We will usually denote unnormalized sums of features, e.g. $\sum_{t=1}^T \boldsymbol{\psi}(\boldsymbol{\tau}_t)$, with $\boldsymbol{\Sigma}$. We also define $\widehat{\boldsymbol{\Omega}}_{\boldsymbol{\psi}}$ to be the space of all possible combinations of $\boldsymbol{\psi}(\boldsymbol{\tau})$:

$$\widehat{\boldsymbol{\Omega}}_{\boldsymbol{\psi}} := \{\mathbb{E}_{\boldsymbol{\tau} \sim \omega}[\boldsymbol{\psi}(\boldsymbol{\tau})] \; : \; \omega \in \triangle_\mathcal{T}\}.$$

We generalize Assumption 7 and Assumption 8 as follows.

**Assumption 9** (Regularity of $\Phi$). *We make the following assumptions:*

*1. $\Phi$ is convex, differentiable, and $\beta$-smooth in the norm $\|\cdot\|$:*

$$\|\nabla_{\boldsymbol{\Gamma}}\Phi(\boldsymbol{\Gamma}) - \nabla_{\boldsymbol{\Gamma}'}\Phi(\boldsymbol{\Gamma}')\|_* \leq \beta \cdot \|\boldsymbol{\Gamma} - \boldsymbol{\Gamma}'\|, \quad \forall \boldsymbol{\Gamma}, \boldsymbol{\Gamma}' \in \widehat{\boldsymbol{\Omega}}_{\boldsymbol{\psi}}$$

*for $\|\cdot\|_*$ the dual norm of $\|\cdot\|$.*

*2. There exists some $M < \infty$ satisfying*

$$\sup_{\boldsymbol{\Gamma} \in \widehat{\boldsymbol{\Omega}}_{\boldsymbol{\psi}}} \sup_{\boldsymbol{\tau} \in \mathcal{T}} |\langle \nabla_{\boldsymbol{\Gamma}}\Phi(\boldsymbol{\Gamma}), \boldsymbol{\psi}(\boldsymbol{\tau})\rangle| \leq M.$$

**Assumption 10** (Regret Minimization Oracle). *Let $\text{cost}_h(\boldsymbol{\tau}) = \langle Q_h, \boldsymbol{\psi}_h(\boldsymbol{\tau})\rangle$ for some $Q_h \in \mathbb{R}^d$, and $\text{cost}(\boldsymbol{\tau}) = \sum_{h=1}^H \text{cost}_h(\boldsymbol{\tau})$ the total cost of trajectory $\boldsymbol{\tau}$. We assume we have access to some learner $\mathbb{A}_\mathcal{R}$ which, in the setting when $|\text{cost}(\boldsymbol{\tau})| \leq 1$ for all $\boldsymbol{\tau} \in \mathcal{T}$, is able to achieve low regret on $\{\text{cost}_h(\cdot, \cdot)\}_{h=1}^H$ with respect to policy class $\Pi_{\text{exp}}$. That is, with probability at least $1 - \delta$:*

$$\sum_{t=1}^T \mathbb{E}_{f,\pi_t}[\text{cost}(\boldsymbol{\tau}_t)] - T \cdot \inf_{\pi \in \Pi_{\text{exp}}} \mathbb{E}_{f,\pi}[\text{cost}(\boldsymbol{\tau})] \leq C_\mathcal{R} \cdot \log^{p_\mathcal{R}}\frac{T}{\delta} \cdot T^\alpha$$

*for some $C_\mathcal{R} > 0$, $p_\mathcal{R} > 0$, and $\alpha \in (0, 1)$, and where $\pi_t$ is the policy $\mathbb{A}_\mathcal{R}$ plays at episode $t$.*

---

**Algorithm 4** Dynamic Optimal Experiment Design (DYNAMICOED)

---

1: **input**: objective $\Phi$, number of episodes $T$ (OR number of iterates $N$, episodes per iterate $K$), confidence $\delta$, regret minimization algorithm $\mathbb{A}_\mathcal{R}$, exploration policies $\Pi_{\exp}$
2: Play any policy $\pi_{\exp} \in \Pi_{\exp}$ for $K$ episodes, collect trajectories $\mathfrak{D}_0 = \{\boldsymbol{\tau}_k^0\}_{k=1}^K$, set $\boldsymbol{\Gamma}_0 \leftarrow K^{-1} \sum_{k=1}^K \boldsymbol{\psi}(\boldsymbol{\tau}_k^0)$
3: **for** $n = 1, 2, \ldots, N$ **do**
4:     Set $\gamma_n \leftarrow \frac{1}{n+1}$
5:     Run $\mathbb{A}_\mathcal{R}$ on cost

$$\mathrm{cost}_h^n(\boldsymbol{\tau}) \leftarrow \frac{1}{M} \langle \Xi_n, \boldsymbol{\psi}_h(\boldsymbol{x}_h, \boldsymbol{u}_h) \rangle \quad \text{for} \quad \Xi_n \leftarrow \nabla_{\boldsymbol{\Gamma}} \Phi(\boldsymbol{\Gamma})|_{\boldsymbol{\Gamma}=\boldsymbol{\Gamma}_{n-1}}$$

    for $K$ episodes, collect trajectories $\mathfrak{D}_n = \{\boldsymbol{\tau}_k^n\}_{k=1}^K$, denote policies run as $\Pi_n$
6:     $\boldsymbol{\Gamma}_n \leftarrow (1 - \gamma_n)\boldsymbol{\Gamma}_{n-1} + \gamma_n K^{-1} \sum_{k=1}^K \boldsymbol{\psi}(\boldsymbol{\tau}_k^n)$
7: **return** $(N+1)K\boldsymbol{\Gamma}_N, \cup_{n=0}^N \Pi_n, \cup_{n=0}^N \mathfrak{D}_n$

---

We define DYNAMICOED as in Algorithm 4. We then have the following generalization of Theorem 3.

**Theorem 5** (Full Version of Theorem 3). *Let Assumption 7 hold, and assume that we have access to a learner $\mathbb{A}_\mathcal{R}$ satisfying Assumption 8. Fix $N, K > 0$. Then, with probability at least $1 - \delta$, DYNAMICOED runs for at most $(N+1)K$ episodes, and collects a dataset satisfying $\mathfrak{D} = \{\{\boldsymbol{\tau}_k^n\}_{k=1}^K\}_{n=0}^N$ satisfying*

$$\Phi\left(\frac{1}{K(N+1)} \sum_{n=0}^N \sum_{k=1}^K \boldsymbol{\psi}(\boldsymbol{\tau}_k^n)\right) - \min_{\boldsymbol{\Gamma} \in \boldsymbol{\Omega}_\psi} \Phi(\boldsymbol{\Gamma}) \leq \frac{\beta R^2(\log N + 1)}{2(N+1)} + M \cdot \left(C_\mathcal{R} \log^{p_\mathcal{R}} \frac{2NK}{\delta} \cdot K^{\alpha-1}\right.$$
$$\left. + \sqrt{\frac{8\log(4N/\delta)}{K}}\right)$$

*where $R = \sup_{\boldsymbol{\Gamma}, \boldsymbol{\Gamma}' \in \widehat{\boldsymbol{\Omega}}_\psi} \|\boldsymbol{\Gamma} - \boldsymbol{\Gamma}'\|$.*

In this work we are particularly interested in the case where $\boldsymbol{\psi}(\boldsymbol{\tau}) \in \mathcal{S}_+^{d_\psi}$. We encapsulate this in the following assumption.

**Assumption 11** (Matrix Experiment Design). *We assume that $\boldsymbol{\psi}(\boldsymbol{\tau}) \in \mathcal{S}_+^{d_\psi}$ and that, for all $\boldsymbol{\tau} \in \mathcal{T}$, $\mathrm{tr}(\boldsymbol{\psi}(\boldsymbol{\tau})) \leq D$ for some $D > 0$.*

The following corollary instantiates Theorem 3 under Assumption 11 with objective $\Phi(\boldsymbol{\Gamma}) = \mathrm{tr}\left(\mathcal{H}(\boldsymbol{\Gamma} + \boldsymbol{\Gamma}_0)^{-1}\right)$, the objective considered in Algorithm 1.

**Corollary 1.** *Consider the objective*

$$\Phi(\boldsymbol{\Gamma}) = \mathrm{tr}\left(\mathcal{H} \cdot (\boldsymbol{\Gamma} + \boldsymbol{\Gamma}_0)^{-1}\right)$$

*and assume that $\mathcal{H} \succeq 0$ and Assumption 10 holds with $\alpha = 1/2$ and Assumption 11 holds. Fix $N, K$, let $T := (N+1)K$, and consider running Algorithm 4 on this objective and with these choices of $N$ and $K$. Then Algorithm 4 will run for at most $T$ episodes, and, with probability at least $1 - \delta$, will return data satisfying*

$$\mathrm{tr}\left(\mathcal{H}\left(\sum_{t=1}^T \boldsymbol{\psi}(\boldsymbol{\tau}_t) + T\boldsymbol{\Gamma}_0\right)^{-1}\right) \leq \frac{1}{T} \cdot \min_{\boldsymbol{\Gamma} \in \boldsymbol{\Omega}_\psi} \mathrm{tr}\left(\mathcal{H}(\boldsymbol{\Gamma} + \boldsymbol{\Gamma}_0)^{-1}\right) + \frac{8D^4 \|\mathcal{H}\|_{\mathrm{op}} \|\boldsymbol{\Gamma}_0^{-1}\|_{\mathrm{op}}^3}{T(N+1)}$$
$$+ \frac{8D\|\mathcal{H}\|_{\mathrm{op}} \|\boldsymbol{\Gamma}_0^{-1}\|_{\mathrm{op}}^2 (\log^{1/2} \frac{4T}{\delta} + C_\mathcal{R} \log^{p_\mathcal{R}} \frac{2T}{\delta})}{T\sqrt{K}}$$

## C.1   Proof of Theorem 3 and Theorem 5

**Lemma C.1** (Lemma C.1 of [56]). *Consider running Algorithm 5 with some convex function $f$ that is $\beta$-smooth with respect to some norm $\|\cdot\|$, assume that $\boldsymbol{y}_n \in \mathcal{Y}$ for some $\mathcal{Y}$ and all $n$, and let*

---

**Algorithm 5** Approximate Frank-Wolfe

---

1: **input**: function to optimize $f$, number of iterations to run $N$, starting iterate $\boldsymbol{x}_1$
2: **for** $t = 1, 2, \ldots, N$ **do**
3:     Set $\gamma_n \leftarrow \frac{1}{n+1}$
4:     Choose $\boldsymbol{y}_n$ to be any point such that

$$\nabla f(\boldsymbol{z}_n)^\top \boldsymbol{y}_n \leq \min_{\boldsymbol{y} \in \mathcal{Z}} \nabla f(\boldsymbol{z}_n)^\top \boldsymbol{y} + \epsilon_n$$

5:     $\boldsymbol{z}_{n+1} \leftarrow (1 - \gamma_n)\boldsymbol{z}_n + \gamma_n \boldsymbol{y}_n$
6: **return** $\boldsymbol{x}_{N+1}$

---

$R := \sup_{\boldsymbol{z}, \boldsymbol{y} \in \mathcal{Z} \cup \mathcal{Y}} \|\boldsymbol{z} - \boldsymbol{y}\|$. *Then for $N \geq 2$, we have*

$$f(\boldsymbol{z}_{N+1}) - \min_{\boldsymbol{z} \in \mathcal{Z}} f(\boldsymbol{z}) \leq \frac{\beta R^2 (\log N + 1)}{2(N+1)} + \frac{1}{N+1} \sum_{n=1}^{N} \epsilon_n.$$

**Lemma C.2** (Lemma C.2 of [56])**.** *When running Algorithm 5, we have*

$$\boldsymbol{z}_{N+1} = \frac{1}{N+1} \left( \sum_{n=1}^{N} \boldsymbol{y}_n + \boldsymbol{z}_1 \right).$$

*Proof of Theorem 3.* By our assumption on $\mathbb{A}_\mathcal{R}$, Assumption 8, we have that, at round $n$, with probability at least $1 - \delta/2N$,

$$\sum_{k=1}^{K} \mathbb{E}_{\pi_k}[\text{cost}^n(\boldsymbol{\tau}_k)] - K \cdot \inf_{\pi \in \Pi_{\exp}} \mathbb{E}_\pi[\text{cost}^n(\boldsymbol{\tau})] \leq C_\mathcal{R} \log^{p_\mathcal{R}} \frac{2NK}{\delta} \cdot K^\alpha$$

where we have used that, under Assumption 9 and by the definition of $\text{cost}_h^n(\boldsymbol{\tau})$, $|\text{cost}^n(\boldsymbol{\tau})| \leq 1$ for all $\boldsymbol{\tau} \in \mathcal{T}$. This implies that

$$\frac{1}{K} \sum_{k=1}^{K} \mathbb{E}_{\pi_k}[\langle \Xi_n, \boldsymbol{\psi}(\boldsymbol{\tau}_k) \rangle] \leq M \cdot \inf_{\pi \in \Pi_{\exp}} \mathbb{E}_\pi[\text{cost}^n(\boldsymbol{\tau})] + M C_\mathcal{R} \log^{p_\mathcal{R}} \frac{2NK}{\delta} \cdot K^{\alpha-1}.$$

Furthermore, by Azuma-Hoeffding and under Assumption 9, we have that, with probability at least $1 - \delta/2N$,

$$\left| \frac{1}{K} \sum_{k=1}^{K} \langle \Xi_n, \boldsymbol{\psi}(\boldsymbol{\tau}_k) \rangle - \frac{1}{K} \sum_{k=1}^{K} \mathbb{E}_{\pi_k}[\langle \Xi_n, \boldsymbol{\psi}(\boldsymbol{\tau}_k) \rangle] \right| \leq \sqrt{\frac{8M^2 \log(4N/\delta)}{K}}.$$

This implies that

$$\frac{1}{K} \sum_{k=1}^{K} \langle \Xi_n, \boldsymbol{\psi}(\boldsymbol{\tau}_k) \rangle \leq M \cdot \inf_{\pi \in \Pi_{\exp}} \mathbb{E}_\pi[\text{cost}^n(\boldsymbol{\tau})] + M \cdot \left( C_\mathcal{R} \log^{p_\mathcal{R}} \frac{K}{\delta} \cdot K^{\alpha-1} + \sqrt{\frac{8 \log(4N/\delta)}{K}} \right).$$

$$\text{(C.1)}$$

Note that

$$\langle \Xi_n, \boldsymbol{\psi}(\boldsymbol{\tau}_k) \rangle = \langle \nabla_{\boldsymbol{\Gamma}} \Phi(\boldsymbol{\Gamma})|_{\boldsymbol{\Gamma} = \boldsymbol{\Gamma}_n}, \boldsymbol{\psi}(\boldsymbol{\tau}) \rangle,$$

and that for any $\boldsymbol{\Gamma} \in \boldsymbol{\Omega}_{\boldsymbol{\psi}}$, we have

$$\langle \nabla_{\boldsymbol{\Gamma}} \Phi(\boldsymbol{\Gamma})|_{\boldsymbol{\Gamma} = \boldsymbol{\Gamma}_n}, \boldsymbol{\Gamma} \rangle = \mathbb{E}_{\pi \sim \omega}[\mathbb{E}_\pi[\langle \nabla_{\boldsymbol{\Gamma}} \Phi(\boldsymbol{\Gamma})|_{\boldsymbol{\Gamma} = \boldsymbol{\Gamma}_n}, \boldsymbol{\psi}(\boldsymbol{\tau}) \rangle]]$$

for some $\omega$. This implies that

$$\inf_{\boldsymbol{\Gamma} \in \boldsymbol{\Omega}_{\boldsymbol{\psi}}} \langle \nabla_{\boldsymbol{\Gamma}} \Phi(\boldsymbol{\Gamma})|_{\boldsymbol{\Gamma} = \boldsymbol{\Gamma}_n}, \boldsymbol{\Gamma} \rangle = \inf_{\omega \in \triangle_\Pi} \mathbb{E}_{\pi \sim \omega}[\mathbb{E}_\pi[\langle \nabla_{\boldsymbol{\Gamma}} \Phi(\boldsymbol{\Gamma})|_{\boldsymbol{\Gamma} = \boldsymbol{\Gamma}_n}, \boldsymbol{\psi}(\boldsymbol{\tau}) \rangle]]$$

$$= \inf_{\pi \in \Pi_{\exp}} \mathbb{E}_\pi[\langle \Xi_n, \boldsymbol{\psi}(\boldsymbol{\tau}) \rangle]$$

$$= M \cdot \inf_{\pi \in \Pi_{\exp}} \mathbb{E}_\pi[\mathrm{cost}^n(\boldsymbol{\tau})].$$

By (C.1) above, we have that

$$\frac{1}{K} \sum_{k=1}^{K} \boldsymbol{\psi}(\boldsymbol{\tau}_k)$$

is an approximate minimizer of $M \cdot \sup_{\pi \in \Pi_{\exp}} \mathbb{E}_\pi[\mathrm{cost}^n(\boldsymbol{\tau})]$, with approximation tolerance $M(C_\mathcal{R} \log^{p_\mathcal{R}} \frac{2NK}{\delta} \cdot K^{\alpha-1} + \sqrt{\frac{8\log(4N/\delta)}{K}})$. We can therefore apply Lemma C.1 with

$$\epsilon_n = M \cdot \left( C_\mathcal{R} \log^{p_\mathcal{R}} \frac{K}{\delta} \cdot K^{\alpha-1} + \sqrt{\frac{8\log(4N/\delta)}{K}} \right)$$

to get that

$$\Phi(\boldsymbol{\Gamma}_{N+1}) - \min_{\boldsymbol{\Gamma} \in \boldsymbol{\Omega}_\psi} \Phi(\boldsymbol{\Gamma}) \le \frac{\beta R^2 (\log N + 1)}{2(N+1)} + M \cdot \left( C_\mathcal{R} \log^{p_\mathcal{R}} \frac{2NK}{\delta} \cdot K^{\alpha-1} + \sqrt{\frac{8\log(4N/\delta)}{K}} \right).$$

The result then follows since $\boldsymbol{\Gamma}_{N+1} = \frac{1}{K(N+1)} \sum_{n=0}^{N} \sum_{k=1}^{K} \boldsymbol{\psi}(\boldsymbol{\tau}_k^n)$ by Lemma C.2.

$\square$

*Proof of Corollary 1.* By Theorem 5, for any setting of $N$ and $K$, we have that with probability at least $1 - \delta$:

$$\mathrm{tr}\left( \mathcal{H}\left( \frac{1}{K(N+1)} \sum_{n=0}^{N} \sum_{k=1}^{K} \boldsymbol{\psi}(\boldsymbol{\tau}_k^n) + \boldsymbol{\Gamma}_0 \right)^{-1} \right) - \min_{\boldsymbol{\Gamma} \in \boldsymbol{\Omega}} \mathrm{tr}\left( \mathcal{H}(\boldsymbol{\Gamma} + \boldsymbol{\Gamma}_0)^{-1} \right)$$

$$\le \frac{\beta R^2 \log N}{N+1} + \frac{MC_\mathcal{R} \log^{p_\mathcal{R}} \frac{2NK}{\delta}}{K^{1-\alpha}} + M\sqrt{\frac{8\log \frac{4N}{\delta}}{K}}$$

which implies

$$\mathrm{tr}\left( \mathcal{H}\left( \sum_{n=0}^{N} \sum_{k=1}^{K} \boldsymbol{\psi}(\boldsymbol{\tau}_k^n) + T\boldsymbol{\Gamma}_0 \right)^{-1} \right) - \frac{\min_{\boldsymbol{\Gamma} \in \boldsymbol{\Omega}} \mathrm{tr}\left( \mathcal{H}(\boldsymbol{\Gamma} + \boldsymbol{\Gamma}_0)^{-1} \right)}{T}$$

$$\le \frac{\beta R^2 \log N}{T(N+1)} + \frac{MC_\mathcal{R} \log^{p_\mathcal{R}} \frac{2NK}{\delta}}{T\sqrt{K}} + M\frac{1}{T}\sqrt{\frac{8\log \frac{4N}{\delta}}{K}}$$

This gives

$$\mathrm{tr}\left( \mathcal{H}\left( \sum_{n=0}^{N} \sum_{k=1}^{K} \boldsymbol{\psi}(\boldsymbol{\tau}_k^n) + T\boldsymbol{\Gamma}_0 \right)^{-1} \right) \le \frac{\min_{\boldsymbol{\Gamma} \in \boldsymbol{\Omega}} \mathrm{tr}\left( \mathcal{H}(\boldsymbol{\Gamma} + \boldsymbol{\Gamma}_0)^{-1} \right)}{T}$$

$$+ \frac{\beta R^2 \log T}{T(N+1)} + \frac{M(3\log^{1/2} \frac{4T}{\delta} + C_\mathcal{R} \log^{p_\mathcal{R}} \frac{2T}{\delta})}{T\sqrt{K}}.$$

It then remains to bound $R, \beta$, and $M$. By Lemma D.6 of [56], we have that

$$\nabla_{\boldsymbol{\Gamma}} \Phi(\boldsymbol{\Gamma})[\widetilde{\boldsymbol{\Gamma}}] = -\mathrm{tr}\left( \mathcal{H}(\boldsymbol{\Gamma} + \boldsymbol{\Gamma}_0)^{-1} \widetilde{\boldsymbol{\Gamma}}(\boldsymbol{\Gamma} + \boldsymbol{\Gamma}_0)^{-1} \right).$$

We can then compute the second derivative as, using Lemma D.6 of [56]:

$$\nabla_{\boldsymbol{\Gamma}}^2 \Phi(\boldsymbol{\Gamma})[\widetilde{\boldsymbol{\Gamma}}, \bar{\boldsymbol{\Gamma}}] = \frac{\mathrm{d}}{\mathrm{d}t}\left[ -\mathrm{tr}\left( \mathcal{H}(\boldsymbol{\Gamma} + \boldsymbol{\Gamma}_0 + t\bar{\boldsymbol{\Gamma}})^{-1} \widetilde{\boldsymbol{\Gamma}}(\boldsymbol{\Gamma} + \boldsymbol{\Gamma}_0 + t\bar{\boldsymbol{\Gamma}})^{-1} \right) \right]$$

$$= \mathrm{tr}\left( \mathcal{H}(\boldsymbol{\Gamma} + \boldsymbol{\Gamma}_0)^{-1} \bar{\boldsymbol{\Gamma}}(\boldsymbol{\Gamma} + \boldsymbol{\Gamma}_0)^{-1} \widetilde{\boldsymbol{\Gamma}}(\boldsymbol{\Gamma} + \boldsymbol{\Gamma}_0)^{-1} \right)$$

$$+ \operatorname{tr}\left( \mathcal{H}(\boldsymbol{\Gamma} + \boldsymbol{\Gamma}_0)^{-1}\widetilde{\boldsymbol{\Gamma}}(\boldsymbol{\Gamma} + \boldsymbol{\Gamma}_0)^{-1}\bar{\boldsymbol{\Gamma}}(\boldsymbol{\Gamma} + \boldsymbol{\Gamma}_0)^{-1} \right).$$

Recall that $M$ is any bound on

$$\sup_{\boldsymbol{\Gamma} \in \widehat{\boldsymbol{\Omega}}_{\boldsymbol{\psi}}} \sup_{\boldsymbol{\tau} \in \mathcal{T}} |\langle \nabla_{\boldsymbol{\Gamma}} \Phi(\boldsymbol{\Gamma}), \boldsymbol{\psi}(\boldsymbol{\tau}) \rangle|.$$

By the above computation of the gradient, we can bound this as

$$\sup_{\boldsymbol{\Gamma} \in \widehat{\boldsymbol{\Omega}}_{\boldsymbol{\psi}}} \sup_{\boldsymbol{\tau} \in \mathcal{T}} |\langle \nabla_{\boldsymbol{\Gamma}} \Phi(\boldsymbol{\Gamma}), \boldsymbol{\psi}(\boldsymbol{\tau}) \rangle| \leq \sup_{\boldsymbol{\Gamma} \in \widehat{\boldsymbol{\Omega}}_{\boldsymbol{\psi}}} \sup_{\boldsymbol{\tau} \in \mathcal{T}} \left| \operatorname{tr}\left( \mathcal{H}(\boldsymbol{\Gamma} + \boldsymbol{\Gamma}_0)^{-1}\boldsymbol{\psi}(\boldsymbol{\tau})(\boldsymbol{\Gamma} + \boldsymbol{\Gamma}_0)^{-1} \right) \right|$$
$$\leq \|\mathcal{H}\|_{\mathrm{op}} \|\boldsymbol{\Gamma}_0^{-1}\|_{\mathrm{op}}^2 \cdot \sup_{\boldsymbol{\tau} \in \mathcal{T}} \operatorname{tr}(\boldsymbol{\psi}(\boldsymbol{\tau})) \qquad \text{(C.2)}$$
$$\leq D\|\mathcal{H}\|_{\mathrm{op}} \|\boldsymbol{\Gamma}_0^{-1}\|_{\mathrm{op}}^2.$$

To bound $\beta$, by the Mean Value Theorem it suffices to bound the operator norm of $\nabla_{\boldsymbol{\Gamma}}^2 \Phi(\boldsymbol{\Gamma})$. Using the expression above, we can bound this as

$$\sup_{\widetilde{\boldsymbol{\Gamma}}, \bar{\boldsymbol{\Gamma}} \in \widehat{\boldsymbol{\Omega}}_{\boldsymbol{\psi}}} |\nabla_{\boldsymbol{\Gamma}}^2 \Phi(\boldsymbol{\Gamma})[\boldsymbol{\psi}(\boldsymbol{\tau}_1), \boldsymbol{\psi}(\boldsymbol{\tau}_2)]| \leq 2\|\mathcal{H}\|_{\mathrm{op}} \|\boldsymbol{\Gamma}_0^{-1}\|_{\mathrm{op}}^3 \cdot \sup_{\widetilde{\boldsymbol{\Gamma}}, \bar{\boldsymbol{\Gamma}} \in \widehat{\boldsymbol{\Omega}}_{\boldsymbol{\psi}}} \operatorname{tr}(\widetilde{\boldsymbol{\Gamma}}\bar{\boldsymbol{\Gamma}})$$
$$\leq 2D^2 \|\mathcal{H}\|_{\mathrm{op}} \|\boldsymbol{\Gamma}_0^{-1}\|_{\mathrm{op}}^3.$$

Finally, it's straightforward to bound $R \leq 2D$. Putting all of this together gives the result. $\qquad \square$

## C.2 Collecting Full-Rank Data

---

**Algorithm 6** Minimum Eigenvalue Maximization (MINEIG)

---
1: **input**: scale $N$, confidence $\delta$, regret minimization algorithm $\mathbb{A}_{\mathcal{R}}$, exploration policies $\Pi_{\mathrm{exp}}$
2: **for** $j = 1, 2, 3, \ldots$ **do**
3: $\quad N_j \leftarrow \lceil 2^{j/3} \rceil - 1, K_j \leftarrow \lceil 2^{2j/3} \rceil, T_j \leftarrow (N_j + 1)K_j, \lambda_j \leftarrow T_j^{-1/18}, \delta_j \leftarrow \frac{\delta}{4j^2}$
4: $\quad \boldsymbol{\Sigma}_j, \Pi_j \leftarrow \text{DYNAMICOED}(\Phi, N_j, K_j, \delta_j, \mathbb{A}_{\mathcal{R}}, \Pi_{\mathrm{exp}})$ for $\Phi(\boldsymbol{\Gamma}) = \operatorname{tr}((\boldsymbol{\Gamma} + \lambda_j \cdot I)^{-1})$
5: $\quad$ **if** $\lambda_{\min}(\boldsymbol{\Sigma}_j) \geq 12544 D d_{\boldsymbol{\psi}} \log \frac{2N(2 + 32T_j)}{\delta}$ **then**
6: $\quad\quad$ **break**
7: **return** $\Pi_j$

---

In this section, we consider the setting where $\boldsymbol{\psi}(\boldsymbol{\tau}) \in \mathcal{S}_+^{d_{\boldsymbol{\psi}}}$, and our goal is to collect $\{\boldsymbol{\tau}_t\}_{t=1}^T$ such that $\boldsymbol{\psi}(\frac{1}{T}\sum_{t=1}^T \boldsymbol{\psi}(\boldsymbol{\tau}_t)) > 0$. For this to be achievable, we need the following assumption, a generalization of Assumption 3.

**Assumption 12** (Full-Rank Data). *Consider $\boldsymbol{\psi}(\boldsymbol{\tau})$ such that $\boldsymbol{\psi}(\boldsymbol{\tau}) \in \mathbb{S}_+^{d_{\boldsymbol{\psi}}}$. Then we have $\sup_{\boldsymbol{\Gamma} \in \boldsymbol{\Omega}_{\boldsymbol{\psi}}} \lambda_{\min}(\boldsymbol{\Gamma}) \geq \lambda_{\min}^\star$ for some $\lambda_{\min}^\star > 0$.*

Throughout this section we also assume that Assumption 10 is satisfied with $\alpha = 1/2$ (though all results generalize in a straightforward way for $\alpha \neq 1/2$). We have the following result.

**Lemma C.3.** *Under Assumptions 10 to 12, running Algorithm 6 we have that with probability at least $1 - \delta$, it will terminate after collecting at most*

$$\operatorname{poly}\left( d_{\boldsymbol{\psi}}, \frac{1}{\lambda_{\min}^\star}, D, C_{\mathcal{R}}, \log^{p_{\mathcal{R}}} \frac{N}{\delta} \right)$$

*episodes, and return policy set $\Pi$ such that*

$$\lambda_{\min}\left( \sum_{\pi \in \Pi} \boldsymbol{\Gamma}_\pi \right) \geq 6272 D d_{\boldsymbol{\psi}} \log \frac{68N}{\delta}.$$

*Furthermore, if we rerun each policy in $\Pi$ once, the resulting features $\boldsymbol{\Sigma}$ will satisfy, with probability at least $1 - \delta/N$:*

$$\lambda_{\min}(\boldsymbol{\Sigma}) \geq 6272 D d_{\boldsymbol{\psi}} \log \frac{68N}{\delta}.$$

*Proof.* By Lemma C.4 and our choice of $N_j$ and $K_j$ in Algorithm 6, we have that if $\lambda_j \le \frac{\lambda_{\min}^\star}{4d_{\boldsymbol{\psi}}}$ and

$$T_j^{1/3} \ge \widetilde{\Omega}\left(\left(D\lambda_j^{-2}(D^3\lambda_j^{-1} + C_{\mathcal{R}} \cdot \log^{p_{\mathcal{R}}} \frac{1}{\delta_j})\right) \cdot \frac{\lambda_{\min}^\star}{d_{\boldsymbol{\psi}}}\right), \tag{C.3}$$

then $\lambda_{\min}(\boldsymbol{\Sigma}_j) \ge \frac{\lambda_{\min}^\star}{4d_{\boldsymbol{\psi}}} \cdot T_j$ with probability at least $1 - \delta_j$. It follows that, with probability at least $1 - \delta_j$, the if statement on Line 5 will be true once $\lambda_j \le \frac{\lambda_{\min}^\star}{4d_{\boldsymbol{\psi}}}$, (C.3) holds, and

$$\frac{\lambda_{\min}^\star}{4d_{\boldsymbol{\psi}}} \cdot T_j \ge 12544 D d_{\boldsymbol{\psi}} \log \frac{2N(2 + 32T_j)}{\delta}. \tag{C.4}$$

By our choice of $\lambda_j = T_j^{-1/18}$, a sufficient condition to ensure $\lambda_j \le \frac{\lambda_{\min}^\star}{4d_{\boldsymbol{\psi}}}$, (C.3), and (C.4) is

$$T_j \ge \widetilde{\Omega}\left(\max\left\{\left(\frac{d_{\boldsymbol{\psi}}}{\lambda_{\min}^\star}\right)^{18}, \left(\frac{D^4\lambda_{\min}^\star}{d_{\boldsymbol{\psi}}}\right)^6, \left(DC_{\mathcal{R}} \cdot \log^{p_{\mathcal{R}}} \frac{1}{\delta_j} \cdot \frac{\lambda_{\min}^\star}{d_{\boldsymbol{\psi}}}\right)^{9/2}, \frac{Dd_{\boldsymbol{\psi}}^2}{\lambda_{\min}^\star} \cdot \log \frac{NT_j}{\delta}\right\}\right).$$

Since $T_j = \lceil 2^{j/3}\rceil \lceil 2^{2j/3}\rceil \in [2^j, 4 \cdot 2^j]$, it follows that the if statement on Line 5 will be met after running for at most

$$\widetilde{\mathcal{O}}\left(\max\left\{\left(\frac{d_{\boldsymbol{\psi}}}{\lambda_{\min}^\star}\right)^{18}, \left(\frac{D^4\lambda_{\min}^\star}{d_{\boldsymbol{\psi}}}\right)^6, \left(DC_{\mathcal{R}} \cdot \log^{p_{\mathcal{R}}} \frac{1}{\delta_j} \cdot \frac{\lambda_{\min}^\star}{d_{\boldsymbol{\psi}}}\right)^{9/2}, \frac{Dd_{\boldsymbol{\psi}}^2}{\lambda_{\min}^\star} \cdot \log \frac{N}{\delta}\right\}\right) \tag{C.5}$$

episodes.

By Lemma C.5, if $\lambda_{\min}(\boldsymbol{\Sigma}_j) \ge 12544 D d_{\boldsymbol{\psi}} \log \frac{2N(2+32T_j)}{\delta}$ and we rerun all policies in $\Pi_j$, then we will collect data $\widetilde{\boldsymbol{\Sigma}}$ such that $\lambda_{\min}(\widetilde{\boldsymbol{\Sigma}}) \ge \frac{1}{2}\lambda_{\min}(\boldsymbol{\Sigma}_j)$, with probability at least $1 - \delta/2N$. As the if statement on Line 5 will only be true once this is met, it follows that, with probability at least $1 - \delta/2N$, rerunning all policies in $\Pi_j$ once, we will collect data $\boldsymbol{\Sigma}$ which satisfies

$$\lambda_{\min}(\boldsymbol{\Sigma}) \ge \frac{1}{2}\lambda_{\min}(\boldsymbol{\Sigma}_j) \ge 6272 D d_{\boldsymbol{\psi}} \log \frac{2N(2 + 32T_j)}{\delta} \ge 6272 D d_{\boldsymbol{\psi}} \log \frac{68N}{\delta}.$$

The lower bound on $\lambda_{\min}(\sum_{\pi \in \Pi} \boldsymbol{\Gamma}_\pi)$ follows analogously from Lemma C.5.

The result then follows noting that the failure probability of running DYNAMICOED is at most

$$\sum_{j=1}^{\infty} \frac{\delta}{4j^2} \le \delta/2.$$

$\square$

### C.2.1 Supporting Lemmas

**Lemma C.4.** *Under Assumptions 10 to 12, consider running* DYNAMICOED *on the objective*

$$\Phi(\boldsymbol{\Gamma}) = \mathrm{tr}((\boldsymbol{\Gamma} + \lambda \cdot I)^{-1})$$

*with $N = \lceil 2^{i/3}\rceil - 1$ and $K = \lceil 2^{2i/3}\rceil$, for some $\lambda > 0$ and $i$. Let $T := (N+1)K$. Then if $\lambda \le \frac{\lambda_{\min}^\star}{4d_{\boldsymbol{\psi}}}$ and*

$$T^{1/3} \ge \widetilde{\Omega}\left(\left(D\lambda^{-2}(D^3\lambda^{-1} + C_{\mathcal{R}} \cdot \log^{p_{\mathcal{R}}} \frac{1}{\delta})\right) \cdot \frac{\lambda_{\min}^\star}{d_{\boldsymbol{\psi}}}\right), \tag{C.6}$$

*with probability at least $1 - \delta$,*

$$\lambda_{\min}\left(\sum_{t=1}^{T} \boldsymbol{\psi}(\boldsymbol{\tau}_t)\right) \ge \frac{\lambda_{\min}^\star}{4d_{\boldsymbol{\psi}}} \cdot T.$$

*Proof.* Applying Corollary 1 with $\mathcal{H} = I$ and $\mathbf{\Gamma}_0 = \lambda \cdot I$, we have that, with probability at least $1 - \delta$:

$$\text{tr}\left(\left(\sum_{t=1}^{T}\boldsymbol{\psi}(\boldsymbol{\tau}_t) + T\lambda \cdot I\right)^{-1}\right) \leq \frac{1}{T}\cdot\min_{\mathbf{\Gamma}\in\mathbf{\Omega}_{\boldsymbol{\psi}}}\text{tr}((\mathbf{\Gamma}+\lambda\cdot I)^{-1}) + \frac{8D^4\lambda^{-3}}{T(N+1)} + \frac{8D\lambda^{-2}(\log^{1/2}\frac{T}{\delta} + C_{\mathcal{R}}\log^{p_{\mathcal{R}}}\frac{T}{\delta})}{T\sqrt{K}}$$

$$\leq \frac{1}{T}\cdot\min_{\mathbf{\Gamma}\in\mathbf{\Omega}_{\boldsymbol{\psi}}}\text{tr}((\mathbf{\Gamma}+\lambda\cdot I)^{-1}) + \frac{24D^4\lambda^{-3}}{T^{4/3}} + \frac{24D\lambda^{-2}(\log^{1/2}\frac{T}{\delta} + C_{\mathcal{R}}\log^{p_{\mathcal{R}}}\frac{T}{\delta})}{T^{4/3}}$$

where the second inequality follows since

$$T = \lceil 2^{i/3}\rceil\lceil 2^{2i/3}\rceil \leq 4\cdot 2^i$$

which implies $N + 1 = \lceil 2^{i/3}\rceil \geq T^{1/3}/4^{1/3}$ and $K = \lceil 2^{2i/3}\rceil \geq T^{2/3}/4^{2/3}$. If $T$ satisfies (C.6), then we can bound

$$\frac{24D^4\lambda^{-3}}{T^{4/3}} + \frac{24D\lambda^{-2}(\log^{1/2}\frac{T}{\delta} + C_{\mathcal{R}}\log^{p_{\mathcal{R}}}\frac{T}{\delta})}{T^{4/3}} \leq \frac{1}{T}\cdot\frac{d_{\boldsymbol{\psi}}}{\lambda_{\min}^{\star}}.$$

Furthermore, under Assumption 3 there exists some $\mathbf{\Gamma}\in\mathbf{\Omega}_{\boldsymbol{\psi}}$ such that $\mathbf{\Gamma}\succeq\lambda_{\min}^{\star}\cdot I$, so we can upper bound

$$\min_{\mathbf{\Gamma}\in\mathbf{\Omega}_{\boldsymbol{\psi}}}\text{tr}((\mathbf{\Gamma}+\lambda\cdot I)^{-1}) \leq \frac{d_{\boldsymbol{\psi}}}{\lambda_{\min}^{\star}}$$

and we can lower bound

$$\text{tr}\left(\left(\sum_{t=1}^{T}\boldsymbol{\psi}(\boldsymbol{\tau}_t) + T\lambda\cdot I\right)^{-1}\right) \geq \frac{1}{\lambda_{\min}(\sum_{t=1}^{T}\boldsymbol{\psi}(\boldsymbol{\tau}_t)) + T\lambda}.$$

Thus,

$$\frac{1}{\lambda_{\min}(\sum_{t=1}^{T}\boldsymbol{\psi}(\boldsymbol{\tau}_t)) + T\lambda} \leq \frac{1}{T}\cdot\frac{2d_{\boldsymbol{\psi}}}{\lambda_{\min}^{\star}} \implies \lambda_{\min}\left(\sum_{t=1}^{T}\boldsymbol{\psi}(\boldsymbol{\tau}_t)\right) \geq \frac{T\lambda_{\min}^{\star}}{2d_{\boldsymbol{\psi}}} - T\lambda.$$

It follows that if $\lambda \leq \frac{\lambda_{\min}^{\star}}{4d_{\boldsymbol{\psi}}}$, then we have

$$\lambda_{\min}\left(\sum_{t=1}^{T}\boldsymbol{\psi}(\boldsymbol{\tau}_t)\right) \geq T\cdot\frac{\lambda_{\min}^{\star}}{4d_{\boldsymbol{\psi}}}$$

which proves the result. $\qquad\square$

**Lemma C.5.** *Consider running some policies $(\pi_\tau)_{\tau=1}^{T}$, for $\pi_\tau$ $\mathcal{F}_{\tau-1}$-measurable, and collecting covariance $\mathbf{\Sigma}_T = \sum_{t=1}^{T}\boldsymbol{\psi}(\boldsymbol{\tau}_t)$. Then under Assumption 11, as long as*

$$\lambda_{\min}(\mathbf{\Sigma}_T) \geq 12544Dd_{\boldsymbol{\psi}}\log\frac{2+32T}{\delta}$$

*with probability at least $1 - \delta$, if we rerun each $(\pi_\tau)_{\tau=1}^{T}$, we will collect features $\widetilde{\mathbf{\Sigma}}_T$ such that*

$$\lambda_{\min}(\widetilde{\mathbf{\Sigma}}_T) \geq \frac{1}{2}\lambda_{\min}(\mathbf{\Sigma}_T).$$

*Furthermore,*

$$\lambda_{\min}\left(\sum_{\tau=1}^{T}\mathbf{\Gamma}_{\pi_\tau}\right) \geq \frac{1}{2}\lambda_{\min}(\mathbf{\Sigma}_T).$$

*Proof.* This follows from applying Lemma D.7 of [56] to the matrix $\frac{1}{D}\mathbf{\Sigma}_T$. Note that while [56] considers the setting of linear MDPs, the proof of Lemma D.7 of [56] does not make use of the linear MDP assumption, and the proof therefore extends immediately to our setting. Furthermore, though it is not explicitly stated, the lower bound on $\lambda_{\min}(\sum_{\tau=1}^{T}\mathbf{\Gamma}_{\pi_\tau})$ is also proved in Lemma D.7 of [56]. $\qquad\square$

---

**Algorithm 7** Learn Minimizing Exploration Policies (LEARNEXPΠ)

---

1: **input**: $\mathcal{H}$, iterates bound $\widetilde{N}$, confidence $\delta$, regret minimization algorithm $\mathbb{A}_{\mathcal{R}}$, exploration policies $\Pi_{\mathrm{exp}}$
2: **for** $i = 1, 2, 3, \ldots$ **do**
3: $\quad N_i \leftarrow \lceil 2^{i/3} \rceil - 1, K_i \leftarrow \lceil 2^{2i/3} \rceil, T_i \leftarrow (N_i + 1)K_i, \delta_i \leftarrow \delta/4i^2$
4: $\quad \Pi^i_{\mathrm{MINEIG}} \leftarrow \mathrm{MINEIG}(\widetilde{N}T_i, \delta_i, \mathbb{A}_{\mathcal{R}}, \Pi_{\mathrm{exp}})$
5: $\quad$ Run policies in $\Pi^i_{\mathrm{MINEIG}}$ $\lceil T_i/|\Pi^i_{\mathrm{MINEIG}}| \rceil$ times, set $\Gamma^i_0$ to collected features
6: $\quad \Phi(\Gamma) \leftarrow \mathrm{tr}(\mathcal{H}(\Gamma + T_i^{-1}\Gamma^i_0)^{-1})$
7: $\quad \Gamma^i_{\mathrm{fw}}, \Pi^i_{\mathrm{fw}} \leftarrow \mathrm{DYNAMICOED}(\Phi, N_i, K_i, \delta, \mathbb{A}_{\mathcal{R}}, \Pi_{\mathrm{exp}})$
8: $\quad$ **if**

$$\max_{j=1,\ldots,i} |\Pi^j_{\mathrm{MINEIG}}| \le T_i \tag{C.7}$$

$$\frac{16D^4\|\mathcal{H}\|_{\mathrm{op}}\|(T_i^{-1}\Gamma_0)^{-1}\|^3_{\mathrm{op}}}{T_i(N_i+1)} + \frac{16D\|\mathcal{H}\|_{\mathrm{op}}\|(T_i^{-1}\Gamma_0)^{-1}\|^2_{\mathrm{op}}(\log^{1/2}\frac{4T_i}{\delta} + C_{\mathcal{R}}\log^{p_{\mathcal{R}}}\frac{2T_i}{\delta})}{T_i\sqrt{K_i}} \le \mathrm{tr}\left(\mathcal{H}\left(\Gamma^i_{\mathrm{fw}} + \Gamma^i_0\right)^{-1}\right) \tag{C.8}$$

$$\mathrm{tr}(\mathcal{H}) \cdot D\sqrt{2T_i}\sqrt{8d_{\psi}\log(1 + 8\sqrt{2T_i}) + 8\log 1/\delta} \cdot \frac{2}{\lambda_{\min}\left(\Gamma^i_{\mathrm{fw}} + \Gamma^i_0\right)^2} \le \mathrm{tr}\left(\mathcal{H}\left(\Gamma^i_{\mathrm{fw}} + \Gamma^i_0\right)^{-1}\right) \tag{C.9}$$

$$D\sqrt{2T_i}\sqrt{8d_{\psi}\log(1 + 8\sqrt{2T_i}) + 8\log 1/\delta} \le \frac{1}{2}\lambda_{\min}\left(\Gamma^i_{\mathrm{fw}} + \Gamma^i_0\right) \tag{C.10}$$

9: $\quad$ **then**
10: $\quad\quad \Gamma_{\mathrm{out}} \leftarrow \Gamma^i_{\mathrm{fw}} + \Gamma^i_0, \Pi_{\mathrm{out}} \leftarrow \Pi^i_{\mathrm{MINEIG}} \cup (\cup^{\lceil T_i/|\Pi_{\mathrm{MINEIG}}|\rceil}_{j=1} \Pi^i_{\mathrm{fw}})$
11: $\quad\quad$ **return** $\Gamma_{\mathrm{out}}, \Pi_{\mathrm{out}}$

---

## C.3 Rerunning Policies

In this section, we build on the analysis of the DYNAMICOED algorithm to show that, not only do the features collected by DYNAMICOED approximately minimize $\Phi$, but that, under certain conditions, if we rerun the policies that DYNAMICOED ran to collect this data, we will collect a new set of features which also approximately minimizes $\Phi$.

In particular, we specialize this argument to objectives of the form $\Phi(\Gamma) = \mathrm{tr}(\mathcal{H}\Gamma^{-1})$. LEARNEXPΠ (Algorithm 7) proceeds by first calling MINEIG to collect full-rank data, using this data as a regularizer of $\Phi(\Gamma)$, and the running DYNAMICOED on this objective. After meeting a certain termination criteria, it terminates, and returns the policies it has run over its operation.

**Lemma C.6.** *Let $\mathcal{E}_{\mathrm{exp}}$ denote the event that, for all $i = 1, 2, 3, \ldots$, the success event of MINEIG and DYNAMICOED occur, and*

$$\lambda_{\min}(\Gamma^i_0) \ge \lceil T_i/|\Pi^i_{\mathrm{MINEIG}}| \rceil \cdot 6272Dd_{\psi}\log\frac{68\widetilde{N}}{\delta}.$$

*Then if Assumptions 10 to 12 hold, $\mathbb{P}[\mathcal{E}_{\mathrm{exp}}] \ge 1 - \delta$.*

**Lemma C.7.** *Consider rerunning each policy in $\Pi_{\mathrm{out}}$ $N \le \widetilde{N}$ times, and let $\widetilde{\Gamma}$ denote the obtained features. Then, if Assumptions 10 to 12 hold, with probability at least $1 - 3\delta$, on the event $\mathcal{E}_{\mathrm{exp}}$:*

$$\left\|\widetilde{\Gamma} - N \cdot \sum_{\pi \in \Pi_{\mathrm{out}}}\Gamma_{\pi}\right\|_{\mathrm{op}} \le \frac{\sqrt{T_{\mathrm{out}}} \cdot \sqrt{8d_{\psi}\log(1 + 8\sqrt{NT_{\mathrm{out}}}) + 8\log 1/\delta}}{\sqrt{N} \cdot 6272d_{\psi}\log\frac{68\widetilde{N}}{\delta}} \cdot \lambda_{\min}\left(N \cdot \sum_{\pi \in \Pi_{\mathrm{out}}}\Gamma_{\pi}\right), \tag{C.11}$$

$$N \cdot 6272Dd_{\psi}\log\frac{68\widetilde{N}}{\delta} \le \min\left\{\lambda_{\min}\left(N \cdot \sum_{\pi \in \Pi_{\mathrm{out}}}\Gamma_{\pi}\right), \lambda_{\min}(\widetilde{\Gamma})\right\}, \tag{C.12}$$

*and*

$$\mathrm{tr}\left(\mathcal{H}\left(\sum_{\pi \in \Pi_{\mathrm{out}}}\Gamma_{\pi}\right)^{-1}\right) \le \frac{12}{T_{\mathrm{out}}} \cdot \min_{\Gamma \in \Omega}\mathrm{tr}\left(\mathcal{H}\Gamma^{-1}\right) \tag{C.13}$$

*for* $T_{\text{out}} := |\Pi_{\text{out}}|$.

**Lemma C.8.** *On the event $\mathcal{E}_{\text{exp}}$, under Assumptions 10 to 12, we can bound*

$$T_{\text{out}} \le \text{poly}\left(d_{\boldsymbol{\psi}}, \frac{1}{\lambda_{\min}^{\star}}, D, C_{\mathcal{R}}, \log^{p_{\mathcal{R}}} \frac{\widetilde{N}}{\delta}\right).$$

*Furthermore, the total number of episodes collected by Algorithm 7 is bounded by $(16 + 2\log(T_{\text{out}})) \cdot T_{\text{out}}$.*

### C.3.1 Supporting Lemmas and Proofs

**Lemma C.9.** *Under Assumption 11, for any $\boldsymbol{\Gamma} = \mathbb{E}_{\boldsymbol{\tau} \sim \omega}[\boldsymbol{\psi}(\boldsymbol{\tau})]$ and $\mathcal{H} \succeq 0$ we can bound*

$$\text{tr}(\mathcal{H}\boldsymbol{\Gamma}^{-1}) \ge D^{-1} \cdot \text{tr}(\mathcal{H}).$$

*Proof.* By Von Neumann's Trace Inequality we can lower bound

$$\text{tr}(\mathcal{H}\boldsymbol{\Gamma}^{-1}) \ge \lambda_{\min}(\boldsymbol{\Gamma}^{-1}) \cdot \text{tr}(\mathcal{H}) = \|\boldsymbol{\Gamma}\|_{\text{op}}^{-1} \cdot \text{tr}(\mathcal{H}).$$

By our assumption that $\text{tr}(\boldsymbol{\psi}(\boldsymbol{\tau})) \le D$, we can bound $\|\boldsymbol{\Gamma}\|_{\text{op}} \le D$, which proves the result. □

**Lemma C.10.** *Assume $\text{tr}(\boldsymbol{\psi}(\boldsymbol{\tau})) \le D$ for all $\boldsymbol{\tau}$. Let $\boldsymbol{\Gamma}_K$ denote the time-normalized features obtained by playing policies $\{\pi_k\}_{k=1}^{K}$, where $\pi_k$ is $\mathcal{F}_{k-1}$-measurable. Then, with probability at least $1 - \delta$,*

$$\left\|\frac{1}{K}\sum_{k=1}^{K} \boldsymbol{\Gamma}_{\pi_k} - \boldsymbol{\Gamma}_K\right\|_{\text{op}} \le D\sqrt{\frac{8d_{\boldsymbol{\psi}}\log(1 + 8\sqrt{K}) + 8\log 1/\delta}{K}}.$$

*Proof.* This follows from an argument identical to the proof of Lemma C.4 of [56]. While [56] considers the setting of linear MDPs, we note that the proof of Lemma C.4 of [56] nowhere relies on the linear MDP assumption. The result stated here then follows identically as Lemma C.4 of [56], after normalizing $\boldsymbol{\psi}(\boldsymbol{\tau})$ by $D$. □

*Proof of Lemma C.6.* By Lemma C.3, the failure probability of running MINEIG at round $i$ is $\delta_i = \delta/8i^2$, and by Corollary 1 the failure probability of DYNAMICOED at round $i$ is also bounded by $\delta_i = \delta/8i^2$. It follows that the total failure probability of running MINEIG and DYNAMICOED is bounded by

$$\sum_{i=1}^{\infty} \frac{2 \cdot \delta}{8i^2} \le \frac{\delta}{2}.$$

Furthermore, by Lemma C.3, we have that rerunning all policies in $\Pi_{\text{MINEIG}}^i$, we will obtain features $\boldsymbol{\Gamma}$ satisfying, with probability at least $1 - \delta_i/\widetilde{N}T_i$:

$$\lambda_{\min}(\boldsymbol{\Gamma}) \ge 6272 D d_{\boldsymbol{\psi}} \log \frac{68\widetilde{N}}{\delta_i}.$$

Repeating this $\lceil T_i/|\Pi_{\text{MINEIG}}^i|\rceil$ times and union bounding, we have that

$$\lambda_{\min}(\boldsymbol{\Gamma}_0^i) \ge \lceil T_i/|\Pi_{\text{MINEIG}}^i|\rceil \cdot 6272 D d_{\boldsymbol{\psi}} \log \frac{68\widetilde{N}}{\delta_i}$$

with probability at least $1 - \delta/\widetilde{N}T_i \cdot \lceil T_i/|\Pi_{\text{MINEIG}}^i|\rceil \ge 1 - \delta/\widetilde{N}$. □

*Proof of Lemma C.7.* Let $\Pi_{\text{MINEIG}}, \boldsymbol{\Gamma}_0, \Pi_{\text{fw}}, \boldsymbol{\Gamma}_{\text{fw}}$ denote the policies and features obtained on the round at which Algorithm 7 terminates. Let $T_{\text{fw}} = |\Pi_{\text{fw}}|$ denote the number of episodes of DYNAMICOED on the terminating round, and $N_{\text{fw}}, K_{\text{fw}}$ the corresponding values of $N_i$ and $K_i$. Throughout the proof we make use of the fact that at termination of Algorithm 7, all of (C.7)-(C.10) are met.

**Proof of (C.11) and (C.12).** By Lemma C.10 we have that, with probability at least $1 - \delta$:

$$\left\| \widetilde{\boldsymbol{\Gamma}} - N \cdot \sum_{\pi \in \Pi_{\text{out}}} \boldsymbol{\Gamma}_\pi \right\|_{\text{op}} \leq D \sqrt{NT_{\text{out}}} \cdot \sqrt{8 d_{\boldsymbol{\psi}} \log(1 + 8\sqrt{NT_{\text{out}}}) + 8 \log 1/\delta}.$$

On $\mathcal{E}_{\text{exp}}$, by Lemma C.3, we can bound

$$|\Pi_{\text{MinEig}}| \leq \text{poly}\left( d_{\boldsymbol{\psi}}, \frac{1}{\lambda_{\min}^\star}, D, C_{\mathcal{R}}, \log^{p_{\mathcal{R}}} \frac{\widetilde{N} T_{\text{out}}}{\delta} \right)$$

and, furthermore, we can lower bound

$$\lambda_{\min} \left( \sum_{\pi \in \Pi_{\text{MinEig}}} \boldsymbol{\Gamma}_\pi \right) \geq 6272 D d_{\boldsymbol{\psi}} \log \frac{68 \widetilde{N}}{\delta}.$$

Since $\Pi_{\text{MinEig}} \subseteq \Pi_{\text{out}}$, it follows that

$$\lambda_{\min} \left( N \cdot \sum_{\pi \in \Pi_{\text{out}}} \boldsymbol{\Gamma}_\pi \right) \geq N \cdot 6272 D d_{\boldsymbol{\psi}} \log \frac{68 \widetilde{N}}{\delta}.$$

Combining these, we therefore have that, with probability at least $1 - \delta$:

$$\left\| \widetilde{\boldsymbol{\Gamma}} - N \cdot \sum_{\pi \in \Pi_{\text{out}}} \boldsymbol{\Gamma}_\pi \right\|_{\text{op}} \leq \frac{\sqrt{NT_{\text{out}}} \cdot \sqrt{8 d_{\boldsymbol{\psi}} \log(1 + 8\sqrt{NT_{\text{out}}}) + 8 \log 1/\delta}}{N \cdot 6272 d_{\boldsymbol{\psi}} \log \frac{68 \widetilde{N}}{\delta}} \cdot \lambda_{\min} \left( N \cdot \sum_{\pi \in \Pi_{\text{out}}} \boldsymbol{\Gamma}_\pi \right).$$

In addition, also by Lemma C.3, we have that with probability at least $1 - \delta/\widetilde{N} T_{\text{out}} \cdot N \geq 1 - \delta$, that

$$\lambda_{\min}(\widetilde{\boldsymbol{\Gamma}}) \geq N \cdot 6272 D d_{\boldsymbol{\psi}} \log \frac{68 \widetilde{N}}{\delta}.$$

**Proof of (C.13).** By Corollary 1, on $\mathcal{E}_{\text{exp}}$ we have that:

$$\Phi(\boldsymbol{\Gamma}_{\text{fw}}) = \text{tr}\left( \mathcal{H} \left( \boldsymbol{\Gamma}_{\text{fw}} + \boldsymbol{\Gamma}_0 \right)^{-1} \right) \leq \frac{\min_{\boldsymbol{\Gamma} \in \boldsymbol{\Omega}} \text{tr} \left( \mathcal{H}(\boldsymbol{\Gamma} + T_{\text{fw}}^{-1} \boldsymbol{\Gamma}_0)^{-1} \right)}{T_{\text{fw}}}$$
$$+ \frac{8 D^4 \|\mathcal{H}\|_{\text{op}} \|(T_{\text{fw}}^{-1} \boldsymbol{\Gamma}_0)^{-1}\|_{\text{op}}^3}{T_{\text{fw}}(N_{\text{fw}} + 1)} + \frac{8 D \|\mathcal{H}\|_{\text{op}} \|(T_{\text{fw}}^{-1} \boldsymbol{\Gamma}_0)^{-1}\|_{\text{op}}^2 (\log^{1/2} \frac{4 T_{\text{fw}}}{\delta} + C_{\mathcal{R}} \log^{p_{\mathcal{R}}} \frac{2 T_{\text{fw}}}{\delta})}{T_{\text{fw}} \sqrt{K_{\text{fw}}}}.$$

Since $T_{\text{fw}}$ satisfies (C.8), we can bound

$$\frac{8 D^4 \|\mathcal{H}\|_{\text{op}} \|(T_{\text{fw}}^{-1} \boldsymbol{\Gamma}_0)^{-1}\|_{\text{op}}^3}{T_{\text{fw}}(N_{\text{fw}} + 1)} + \frac{8 D \|\mathcal{H}\|_{\text{op}} \|(T_{\text{fw}}^{-1} \boldsymbol{\Gamma}_0)^{-1}\|_{\text{op}}^2 (\log^{1/2} \frac{4 T_{\text{fw}}}{\delta} + C_{\mathcal{R}} \log^{p_{\mathcal{R}}} \frac{2 T_{\text{fw}}}{\delta})}{T_{\text{fw}} \sqrt{K_{\text{fw}}}} \leq \frac{1}{2} \Phi(\boldsymbol{\Gamma}_{\text{fw}}).$$

It follows that

$$\Phi(\boldsymbol{\Gamma}_{\text{fw}}) \leq \frac{\min_{\boldsymbol{\Gamma} \in \boldsymbol{\Omega}} \text{tr} \left( \mathcal{H}(\boldsymbol{\Gamma} + T_{\text{fw}}^{-1} \boldsymbol{\Gamma}_0)^{-1} \right)}{T_{\text{fw}}} + \frac{1}{2} \Phi(\boldsymbol{\Gamma}_{\text{fw}})$$
$$\implies \Phi(\boldsymbol{\Gamma}_{\text{fw}}) \leq 2 \cdot \frac{\min_{\boldsymbol{\Gamma} \in \boldsymbol{\Omega}} \text{tr} \left( \mathcal{H}(\boldsymbol{\Gamma} + T_{\text{fw}}^{-1} \boldsymbol{\Gamma}_0)^{-1} \right)}{T_{\text{fw}}}. \tag{C.14}$$

By Lemma C.10, we have that, with probability at least $1 - \delta$:

$$\left\| \sum_{\pi \in \Pi_{\text{out}}} \boldsymbol{\Gamma}_\pi - (\boldsymbol{\Gamma}_{\text{fw}} + \boldsymbol{\Gamma}_0) \right\|_{\text{op}} \leq D \sqrt{T_{\text{out}}} \sqrt{8 d_{\boldsymbol{\psi}} \log(1 + 8\sqrt{T_{\text{out}}}) + 8 \log 1/\delta}.$$

Since (C.10) is satisfied and $|\Pi_{\text{MinEig}}^i| \leq T_i$, we have

$$D \sqrt{T_{\text{out}}} \sqrt{8 d_{\boldsymbol{\psi}} \log(1 + 8\sqrt{T_{\text{out}}}) + 8 \log 1/\delta} \leq \frac{1}{2} \lambda_{\min} \left( \boldsymbol{\Gamma}_{\text{fw}} + \boldsymbol{\Gamma}_0 \right).$$

By Lemma A.2 it follows that

$$\left\| \left( \sum_{\pi \in \Pi_{\text{out}}} \boldsymbol{\Gamma}_\pi \right)^{-1} - (\boldsymbol{\Gamma}_{\text{fw}} + \boldsymbol{\Gamma}_0)^{-1} \right\|_{\text{op}} \leq D \sqrt{T_{\text{out}}} \sqrt{8 d_{\boldsymbol{\psi}} \log(1 + 8\sqrt{T_{\text{out}}}) + 8 \log 1/\delta} \cdot \frac{2}{\lambda_{\min} \left( \boldsymbol{\Gamma}_{\text{fw}} + \boldsymbol{\Gamma}_0 \right)^2}.$$

This implies that

$$\mathrm{tr}\left(\mathcal{H}\left(\textstyle\sum_{\pi\in\Pi_{\mathrm{out}}}\mathbf{\Gamma}_\pi\right)^{-1}\right) \le \mathrm{tr}\left(\mathcal{H}\left(\mathbf{\Gamma}_{\mathrm{fw}}+\mathbf{\Gamma}_0\right)^{-1}\right)$$
$$+\,\mathrm{tr}(\mathcal{H})\cdot D\sqrt{T_{\mathrm{out}}}\sqrt{8d_\psi\log(1+8\sqrt{T_{\mathrm{out}}})+8\log 1/\delta}\cdot\frac{2}{\lambda_{\min}\left(\mathbf{\Gamma}_{\mathrm{fw}}+\mathbf{\Gamma}_0\right)^2}.$$

Now if

$$\mathrm{tr}(\mathcal{H})\cdot D\sqrt{T_{\mathrm{out}}}\sqrt{8d_\psi\log(1+8\sqrt{T_{\mathrm{out}}})+8\log 1/\delta}\cdot\frac{2}{\lambda_{\min}\left(\mathbf{\Gamma}_{\mathrm{fw}}+\mathbf{\Gamma}_0\right)^2}\le\mathrm{tr}\left(\mathcal{H}\left(\mathbf{\Gamma}_{\mathrm{fw}}+\mathbf{\Gamma}_0\right)^{-1}\right),$$
$$\tag{C.15}$$

we can bound this all by

$$\le 2\mathrm{tr}\left(\mathcal{H}\left(\mathbf{\Gamma}_{\mathrm{fw}}+\mathbf{\Gamma}_0\right)^{-1}\right)\le\frac{4}{T_{\mathrm{fw}}}\cdot\min_{\mathbf{\Gamma}\in\mathbf{\Omega}}\mathrm{tr}\left(\mathcal{H}(\mathbf{\Gamma}+T_{\mathrm{fw}}^{-1}\mathbf{\Gamma}_0)^{-1}\right)$$

where the last inequality follows from (C.14). However, note that (C.15) since (C.9) holds. Finally, note that

$$T_{\mathrm{out}}=T_{\mathrm{fw}}+\lceil T_{\mathrm{fw}}/|\Pi_{\mathrm{MinEig}}|\rceil|\Pi_{\mathrm{MinEig}}|\le 2T_{\mathrm{fw}}+|\Pi_{\mathrm{MinEig}}|\le 3T_{\mathrm{fw}}$$

where the last inequality follows since (C.7) holds. We can therefore upper bound $\frac{4}{T_{\mathrm{fw}}}\le\frac{12}{T_{\mathrm{out}}}$. Putting this together proves the result.

$\square$

*Proof of Lemma C.8.* To bound $T_{\mathrm{out}}$, it suffices to show that (C.7)-(C.10) are satisfied for sufficiently large $T_i$.

On $\mathcal{E}_{\mathrm{exp}}$, by Lemma C.3, we can bound

$$|\Pi_{\mathrm{MinEig}}^i|\le\mathrm{poly}\left(d_\psi,\frac{1}{\lambda_{\min}^\star},D,C_\mathcal{R},\log^{p_\mathcal{R}}\frac{\widetilde{N}T_i}{\delta}\right),$$

so to ensure (C.7) is met it suffices that

$$T_i\ge\mathrm{poly}\left(d_\psi,\frac{1}{\lambda_{\min}^\star},D,C_\mathcal{R},\log^{p_\mathcal{R}}\frac{\widetilde{N}}{\delta}\right).$$

On $\mathcal{E}_{\mathrm{exp}}$, we have

$$\lambda_{\min}(\mathbf{\Gamma}_{\mathrm{fw}}^i+\mathbf{\Gamma}_0^i)\ge\lambda_{\min}(\mathbf{\Gamma}_0^i)\ge\lceil T_i/|\Pi_{\mathrm{MinEig}}^i|\rceil\cdot 6272Dd_\psi\log\frac{68\widetilde{N}}{\delta}.$$

Which also implies

$$\|(T_i^{-1}\mathbf{\Gamma}_0^i)^{-1}\|_{\mathrm{op}}=\frac{T_i}{\lambda_{\min}(\mathbf{\Gamma}_0^i)}\le\frac{T_i}{\lceil T_i/|\Pi_{\mathrm{MinEig}}^i|\rceil}\cdot\frac{1}{6272Dd_\psi\log\frac{68\widetilde{N}}{\delta}}\le\frac{|\Pi_{\mathrm{MinEig}}^i|}{6272Dd_\psi\log\frac{68\widetilde{N}}{\delta}}$$

Furthermore, by Lemma C.9 we can lower bound

$$\mathrm{tr}\left(\mathcal{H}\left(\mathbf{\Gamma}_{\mathrm{fw}}^i+\mathbf{\Gamma}_0^i\right)^{-1}\right)\ge\frac{\mathrm{tr}(\mathcal{H})}{D(T_i+\lceil T_i/|\Pi_{\mathrm{MinEig}}^i|\rceil|\Pi_{\mathrm{MinEig}}^i|)}\ge\frac{\mathrm{tr}(\mathcal{H})}{3DT_i}.$$

Combining these and using that $N_i=\mathcal{O}(T_i^{1/3})$ and $K_i=\mathcal{O}(T_i^{2/3})$, it is easy to see that (C.8)-(C.10) will be met once

$$T_i\ge\mathrm{poly}\left(d_\psi,\frac{1}{\lambda_{\min}^\star},D,C_\mathcal{R},\log^{p_\mathcal{R}}\frac{\widetilde{N}}{\delta}\right).$$

The bound on $T_{\mathrm{out}}$ then follows since $T_i=\lceil 2^{i/3}\rceil\lceil 2^{2i/3}\rceil\in[2^i,4\cdot 2^i]$, so it can be at most a constant larger than the sufficient condition before terminating.

Let $i^\star$ denote the round that Algorithm 7 terminates on. Note that at round $i$, MINEIG runs for at most $|\Pi_{\text{MINEIG}}^i|$, DYNAMICOED runs for at most $T_i$ episodes, and we run for an additional $\lceil T_i/|\Pi_{\text{MINEIG}}^i|\rceil \cdot |\Pi_{\text{MINEIG}}^i|$ episodes on Line 5. In total, then, the number of episodes Algorithm 7 runs for is bounded by

$$\sum_{i=1}^{i^\star}(T_i + |\Pi_{\text{MINEIG}}^i| + \lceil T_i/|\Pi_{\text{MINEIG}}^i|\rceil \cdot |\Pi_{\text{MINEIG}}^i|) \le 2\sum_{i=1}^{i^\star}(T_i + |\Pi_{\text{MINEIG}}^i|)$$

$$\le 16T_{i^\star} + 2\sum_{i=1}^{i^\star}|\Pi_{\text{MINEIG}}^i|$$

where the last inequality follows since $T_i \in [2^i, 4 \cdot 2^i]$. Now note that, since Algorithm 7 only terminates once (C.7) is met, we will have $\max_{j=1,\dots,i^\star}|\Pi_{\text{MINEIG}}^j| \le T_{i^\star}$. This implies that $2\sum_{i=1}^{i^\star}|\Pi_{\text{MINEIG}}^i| \le 2i^\star T_{i^\star} \le 2\log(T_{i^\star}) \cdot T_{i^\star}$. Bounding $T_{i^\star} \le T_{\text{out}}$ gives the result.

$\square$

# D  Smooth Nonlinear Systems

In this section we restrict to the nonlinear regulator system of (1.1). Our goal will be to show that, under our assumptions, the nonlinear regulator system exhibits certain smooth behavior. As we have assumed

$$\Pi^\star = \{\pi^{\boldsymbol{\theta}} \; : \; \boldsymbol{\theta} \in \mathbb{R}^{d_{\boldsymbol{\theta}}}\},$$

it will be convenient to define $\mathcal{J}(\boldsymbol{\theta}; A) := \mathcal{J}(\pi^{\boldsymbol{\theta}}; A)$ and $\boldsymbol{\theta}_\star(A) = \boldsymbol{\theta}_\star$. For the remainder of this section, we will typically use $\boldsymbol{\theta}$ in place of $\pi^{\boldsymbol{\theta}}$. In addition, when considering radius terms such as $r_{\boldsymbol{\theta}}(A_\star)$ and $r_{\text{cost}}(A_\star)$, to simplify results we assume that $r_{\boldsymbol{\theta}}(A_\star) \le 1$ and $r_{\text{cost}}(A_\star) \le 1$. Note that this does not change the validity of the result since, for example, if a result holds with $A \in \mathcal{B}_{\text{F}}(A_\star; r)$ for some $r > r_{\boldsymbol{\theta}}(A_\star)$, it also holds for $A \in \mathcal{B}_{\text{F}}(A_\star; r_{\boldsymbol{\theta}}(A_\star))$. Throughout this section, we let $\nabla_x f(x)[\Delta]$ refer to the directional gradient of $f(x)$ in direction $\Delta$.

We first have the following result, which shows that under our assumptions, the controller loss is differentiable.

**Lemma D.1.** *Under Assumptions 1, 2, 4 and 5, for any $A$ satisfying $A \in \mathcal{B}_{\text{F}}(A_\star; r_{\boldsymbol{\theta}}(A_\star))$, the controller loss $\mathcal{J}(\boldsymbol{\theta}; A)$ is four-times differentiable in $\boldsymbol{\theta}$ and $A$. Furthermore, we can bound*

$$\|\nabla_A^{(i)}\nabla_{\boldsymbol{\theta}}^{(j)}\mathcal{J}(\boldsymbol{\theta}; A)\|_{\text{op}} \le \text{poly}(\|A\|_{\text{op}}, B_{\boldsymbol{\phi}}, L_{\boldsymbol{\phi}}, L_{\boldsymbol{\theta}}, L_{\text{cost}}, \sigma_{\boldsymbol{w}}^{-1}, H, d_{\boldsymbol{x}})$$

*for $i, j \in \{0, 1, 2, 3, 4\}$ satisfying $1 \le i + j \le 3$.*

In this section, we generalize Assumption 6 to the following.

**Assumption 13.** *We assume there exists some $r_{\boldsymbol{\theta}}(A_\star) > 0$ such that, for all $A \in \mathcal{B}_{\text{F}}(A_\star; r_{\boldsymbol{\theta}}(A_\star))$, $\boldsymbol{\theta}_\star(A)$ satisfies:*

- $\nabla_{\boldsymbol{\theta}}\mathcal{J}(\boldsymbol{\theta}; A)|_{\boldsymbol{\theta}=\boldsymbol{\theta}_\star(A)} = 0$,

- $\boldsymbol{\theta}_\star(A)$ *is three-times differentiable in $A$, and we can bound $\|\nabla_A^{(i)}\boldsymbol{\theta}_\star(A)\|_{\text{op}} \le L_{\pi_\star}$ for some $L_{\pi_\star} > 0$ and $i \in \{1, 2, 3\}$.*

The first condition requires that $\boldsymbol{\theta}_\star(A)$ corresponds to a stationary point of the loss. This will be met, for example, by choosing $\boldsymbol{\theta}_\star(A)$ to be a minima (local or global) of $\mathcal{J}(\boldsymbol{\theta}; A)$. It is not obvious, however, that the first and second condition can be simultaneously satisfied. In the following we show that, assuming $\nabla_{\boldsymbol{\theta}}^2 \mathcal{J}(\boldsymbol{\theta}; A_\star)|_{\boldsymbol{\theta}=\boldsymbol{\theta}_\star(A_\star)}$ is full-rank (which will be the case, for example, when $\boldsymbol{\theta}_\star(A_\star)$ is a strict local minimum of $\mathcal{J}(\pi^{\boldsymbol{\theta}}; A_\star)$), there always exists some $\boldsymbol{\theta}_\star(A)$ satisfying both conditions of Assumption 13, with $L_{\pi_\star}$ scaling polynomially in problem parameters, and $r_{\boldsymbol{\theta}}(A_\star)$ scaling inverse polynomially in problem parameters. Note that this definition of $\pi_\star(A)$ is general enough to capture settings where the global minimum of $\mathcal{J}(\pi; A)$ cannot be efficiently computed—it suffices to take $\pi_\star(A)$ a local minimum of the loss.

**Proposition 5.** *Assume that Assumptions 1, 2, 4 and 5 hold and that $\lambda_{\min}(\nabla_{\boldsymbol{\theta}}^2 \mathcal{J}(\boldsymbol{\theta}; A_\star)|_{\boldsymbol{\theta}=\boldsymbol{\theta}_\star(A_\star)}) > 0$. Let $r_{\boldsymbol{\theta}}(A_\star) > 0$ be some value satisfying*

$$r_{\boldsymbol{\theta}}(A_\star) = \min \left\{ r_{\mathrm{cost}}(A_\star), \mathrm{poly}\left(\frac{1}{\lambda_{\min}(\nabla_{\boldsymbol{\theta}}^2 \mathcal{J}(\boldsymbol{\theta}; A_\star)|_{\boldsymbol{\theta}=\boldsymbol{\theta}_\star(A_\star)})}, \|A_\star\|_{\mathrm{op}}, B_{\boldsymbol{\phi}}, L_{\boldsymbol{\phi}}, L_{\boldsymbol{\theta}}, L_{\mathrm{cost}}, \sigma_{\boldsymbol{w}}^{-1}, H, d_{\boldsymbol{x}}\right)^{-1} \right\}.$$

*Then there exists some function $\boldsymbol{\theta}_\star(A)$ such that, for all $A \in \mathcal{B}_{\mathrm{F}}(A_\star; r_{\boldsymbol{\theta}}(A_\star))$:*

- $\nabla_{\boldsymbol{\theta}} \mathcal{J}(\boldsymbol{\theta}; A)|_{\boldsymbol{\theta}=\boldsymbol{\theta}_\star(A)} = 0$,

- $\boldsymbol{\theta}_\star(A)$ *is three-times differentiable in A,*

*and it suffices that we take*

$$L_{\pi_\star} = \mathrm{poly}\left(\frac{1}{\lambda_{\min}(\nabla_{\boldsymbol{\theta}}^2 \mathcal{J}(\boldsymbol{\theta}; A_\star)|_{\boldsymbol{\theta}=\boldsymbol{\theta}_\star(A_\star)})}, \|A_\star\|_{\mathrm{op}}, B_{\boldsymbol{\phi}}, L_{\boldsymbol{\phi}}, L_{\boldsymbol{\theta}}, L_{\mathrm{cost}}, \sigma_{\boldsymbol{w}}^{-1}, H, d_{\boldsymbol{x}}\right).$$

While Proposition 5 shows that there exists some $\boldsymbol{\theta}_\star(A)$ satisfying Assumption 13, it does not directly give a recipe for constructing such a map. The following result shows that under a mild additional assumption, the minimizer of the loss satisfies Assumption 13.

**Proposition 6.** *Let*

$$\boldsymbol{\theta}_\star(A) := \arg\min_{\boldsymbol{\theta} \in \mathbb{R}^{d_{\boldsymbol{\theta}}}} \mathcal{J}(\boldsymbol{\theta}; A).$$

*Then under Assumptions 1, 2 and 4 to 6, there exists some $r_{\boldsymbol{\theta}}(A_\star) > 0$ and $L_{\pi_\star} < \infty$ such that $\boldsymbol{\theta}_\star(A)$ satisfies Assumption 13.*

The scaling of $L_{\pi_\star}$ in Proposition 6 can be shown to match that of Proposition 5, but in general $r_{\boldsymbol{\theta}}(A_\star)$ could be smaller than the value of $r_{\boldsymbol{\theta}}(A_\star)$ given in Proposition 5. In particular, in the setting of Proposition 6, we can only show that $r_{\boldsymbol{\theta}}(A_\star)$ scales with $\min_{\boldsymbol{\theta} \notin \mathcal{B}_2(\boldsymbol{\theta}_\star(A_\star); r)} \mathcal{J}(\boldsymbol{\theta}; A_\star) - \mathcal{J}(\boldsymbol{\theta}_\star(A_\star); A_\star)$ for some $r > 0$ which scales inverse polynomially in problem parameters. While we can show that $\mathcal{J}(\boldsymbol{\theta}; A_\star) - \mathcal{J}(\boldsymbol{\theta}_\star(A_\star); A_\star)$ scales inverse polynomially in problem parameters, including in $\lambda_{\min}(\nabla_{\boldsymbol{\theta}}^2 \mathcal{J}(\boldsymbol{\theta}; A_\star)|_{\boldsymbol{\theta}=\boldsymbol{\theta}_\star(A_\star)})$, for $\boldsymbol{\theta}$ approximately a distance of $r$ from $\boldsymbol{\theta}_\star(A_\star)$, it is possible $\mathcal{J}(\boldsymbol{\theta}; A_\star)$ has some local minimizer $\boldsymbol{\theta}'$ arbitrarily far away from $\boldsymbol{\theta}_\star(A_\star)$, such that $\mathcal{J}(\boldsymbol{\theta}'; A_\star)$ and $\mathcal{J}(\boldsymbol{\theta}_\star(A_\star); A_\star)$ are arbitrarily close, in which case $\Delta^\star$, and therefore $r_{\boldsymbol{\theta}}(A_\star)$, could be arbitrarily small. The failure mode here is that, while $\boldsymbol{\theta}_\star(A_\star)$ may be the global minimum of $\mathcal{J}(\boldsymbol{\theta}; A_\star)$, for $A$ arbitrarily close to $A_\star$, the global minimum of $\mathcal{J}(\boldsymbol{\theta}; A)$ could instead be near $\boldsymbol{\theta}'$, which would render the map $\boldsymbol{\theta}_\star(A)$ discontinuous.

By making further assumptions on $\mathcal{J}(\boldsymbol{\theta}; A_\star)$ which exclude this case, we can obtain a value of $r_{\boldsymbol{\theta}}(A_\star)$ scaling similarly to in Proposition 5. For example, in the following, we show that under the assumption that $\mathcal{J}(\boldsymbol{\theta}; A)$ is convex, this holds.

**Proposition 7.** *Assume that there exists some $r_{\mathrm{conv}}(A_\star) > 0$ such that, for all $A \in \mathcal{B}_{\mathrm{F}}(A_\star; r_{\mathrm{conv}}(A_\star))$, $\mathcal{J}(\boldsymbol{\theta}; A)$ is convex in $\boldsymbol{\theta}$, and set*

$$\boldsymbol{\theta}_\star(A) = \arg\min_{\boldsymbol{\theta} \in \mathbb{R}^{d_{\boldsymbol{\theta}}}} \mathcal{J}(\boldsymbol{\theta}; A).$$

*Then we have that $\boldsymbol{\theta}_\star$ satisfies Assumption 13 with*

$$r_{\boldsymbol{\theta}}(A_\star) = \min \left\{ r_{\mathrm{conv}}(A_\star), r_{\mathrm{cost}}(A_\star), \right.$$

$$\left. \mathrm{poly}\left(\frac{1}{\lambda_{\min}(\nabla_{\boldsymbol{\theta}}^2 \mathcal{J}(\boldsymbol{\theta}; A_\star)|_{\boldsymbol{\theta}=\boldsymbol{\theta}_\star(A_\star)})}, \|A_\star\|_{\mathrm{op}}, B_{\boldsymbol{\phi}}, L_{\boldsymbol{\phi}}, L_{\boldsymbol{\theta}}, L_{\mathrm{cost}}, \sigma_{\boldsymbol{w}}^{-1}, H, d_{\boldsymbol{x}}\right)^{-1} \right\}$$

*and it suffices that we take*

$$L_{\pi_\star} = \mathrm{poly}\left(\frac{1}{\lambda_{\min}(\nabla_{\boldsymbol{\theta}}^2 \mathcal{J}(\boldsymbol{\theta}; A_\star)|_{\boldsymbol{\theta}=\boldsymbol{\theta}_\star(A_\star)})}, \|A_\star\|_{\mathrm{op}}, B_{\boldsymbol{\phi}}, L_{\boldsymbol{\phi}}, L_{\boldsymbol{\theta}}, L_{\mathrm{cost}}, \sigma_{\boldsymbol{w}}^{-1}, H, d_{\boldsymbol{x}}\right).$$

Note that, if $\mathcal{J}(\boldsymbol{\theta}; A)$ is $\mu$-strongly convex in $\boldsymbol{\theta}$ for all $A$ near $A_\star$, we can lower bound $\lambda_{\min}(\nabla_{\boldsymbol{\theta}}^2 \mathcal{J}(\boldsymbol{\theta}; A_\star)|_{\boldsymbol{\theta}=\boldsymbol{\theta}_\star(A_\star)}) \geq \mu$.

**Approximating the Controller Loss.** In order to efficiently direct our exploration, it is convenient to derive a quadratic approximation to the controller loss. The following result shows that, under our assumptions, this is indeed possible.

**Lemma D.2** (Formal Version of Proposition 1). *Under Assumptions 1, 2, 4 to 5 and 13, for $\widehat{A} \in \mathcal{B}_{\mathrm{F}}(A_\star; \min\{r_{\mathrm{cost}}(A_\star), r_{\boldsymbol{\theta}}(A_\star)\})$, we have*

$$\mathcal{J}(\boldsymbol{\theta}_\star(\widehat{A}); A_\star) - \mathcal{J}(\boldsymbol{\theta}_\star(A_\star); A_\star) \leq \mathrm{vec}(\widehat{A} - A_\star)^\top \mathcal{H}(A_\star)\mathrm{vec}(\widehat{A} - A_\star) + M[\widehat{A} - A_\star, \widehat{A} - A_\star, \widehat{A} - A_\star].$$

*for some tensor $M$ such that*

$$\|M[\widehat{A} - A_\star, \widehat{A} - A_\star, \widehat{A} - A_\star]\|_{\mathrm{op}} \leq \mathrm{poly}(L_{\pi_\star}, \|A_\star\|_{\mathrm{op}}, B_\phi, L_\phi, L_{\boldsymbol{\theta}}, L_{\mathrm{cost}}, \sigma_{\boldsymbol{w}}^{-1}, H, d_{\boldsymbol{x}}) \cdot \|\widehat{A} - A_\star\|_{\mathrm{op}}^3.$$

In practice we do not know $\mathcal{H}(A_\star)$ and must estimate it. The following result shows that the distance between $\mathcal{H}(A_\star)$ and $\mathcal{H}(\widehat{A})$ can be bounded.

**Lemma D.3.** *Under Assumptions 1, 2, 4, 5 and 13, and if $\widehat{A} \in \mathcal{B}_{\mathrm{F}}(A_\star; \min\{r_{\mathrm{cost}}(A_\star), r_{\boldsymbol{\theta}}(A_\star)\})$, we can bound*

$$\|\mathcal{H}(A_\star) - \mathcal{H}(\widehat{A})\|_{\mathrm{op}} \leq \mathrm{poly}(L_{\pi_\star}, \|A_\star\|_{\mathrm{op}}, B_\phi, L_\phi, L_{\boldsymbol{\theta}}, L_{\mathrm{cost}}, \sigma_{\boldsymbol{w}}^{-1}, H, d_{\boldsymbol{x}}) \cdot \|\widehat{A} - A_\star\|_{\mathrm{op}}.$$

## D.1 Proof of Smoothness of Nonlinear System

We let $f_{\boldsymbol{w}}(\cdot)$ denote the density of the noise (which, by assumption, is simply an isotropic Gaussian density). We let $f_{A,\boldsymbol{\theta}}(\cdot)$ denote the density over trajectories induced by playing controller $\boldsymbol{\theta}$ on system $A$. We will overload notation somewhat and let $f_{A,\boldsymbol{\theta}}(\boldsymbol{x}_{h+1} \mid \boldsymbol{\tau}_{1:h})$ denote the density over $\boldsymbol{x}_{h+1}$ induced by playing controller $\boldsymbol{\theta}$ given trajectory $\boldsymbol{\tau}_{1:h}$. Note that $f_{A,\boldsymbol{\theta}}(\boldsymbol{x}_{h+1} \mid \boldsymbol{\tau}_{1:h}) = f_{\boldsymbol{w}}(\boldsymbol{x}_{h+1} - A\phi(\boldsymbol{x}_h^{\boldsymbol{\tau}}, \pi_h^{\boldsymbol{\theta}}(\boldsymbol{\tau}_{1:h})))$ and

$$f_{A,\boldsymbol{\theta}}(\boldsymbol{\tau}) = \prod_{h=1}^{H} f_{A,\boldsymbol{\theta}}(\boldsymbol{x}_{h+1} \mid \boldsymbol{\tau}_{1:h}).$$

Throughout this section we let $\boldsymbol{x}_h^{\boldsymbol{\tau}}$ (resp. $\boldsymbol{u}_h^{\boldsymbol{\tau}}$) denote the state (resp. input) at step $h$ of trajectory $\boldsymbol{\tau}$. Under our regularity assumptions (Assumptions 1, 2, 4 and 5) and since the noise is Gaussian, we can swap derivatives and integrals, which we make use of throughout the following proofs.

*Proof of Lemma D.1.* Let $\mathrm{cost}(\boldsymbol{\tau})$ denote the cost of trajectory $\boldsymbol{\tau}$. Then we have

$$\mathcal{J}(\boldsymbol{\theta}; A) = \int \mathrm{cost}(\boldsymbol{\tau}) f_{A,\boldsymbol{\theta}}(\boldsymbol{\tau})\mathrm{d}\boldsymbol{\tau}.$$

Let $A_{\boldsymbol{t}} := A + t_1\Delta_1^A + t_2\Delta_2^A + t_3\Delta_3^A$ and $\boldsymbol{\theta}_{\boldsymbol{s}} := \boldsymbol{\theta} + s_1\Delta_1^{\boldsymbol{\theta}} + s_2\Delta_2^{\boldsymbol{\theta}} + s_3\Delta_3^{\boldsymbol{\theta}}$, for some $\Delta_i^A$ and $\Delta_j^{\boldsymbol{\theta}}$, which we assume satisfy $\|\Delta_i^A\|_{\mathrm{op}}, \|\Delta_j^{\boldsymbol{\theta}}\|_{\mathrm{op}} \leq 1$. Rather than differentiating $\mathcal{J}(\boldsymbol{\theta}; A)$ with respect to $\boldsymbol{\theta}$ or $A$, we will differentiate $\mathcal{J}(\boldsymbol{\theta}_{\boldsymbol{s}}; A_{\boldsymbol{t}})$ with respect to some $x_1, x_2, x_3, x_4 \in \{t_1, t_2, t_3, s_1, s_2, s_3\}$. Note that, for example,

$$\frac{\mathrm{d}}{\mathrm{d}t_1}\mathcal{J}(\boldsymbol{\theta}_{\boldsymbol{s}}; A_{\boldsymbol{t}})|_{\boldsymbol{t}=\boldsymbol{s}=0} = \nabla_A\mathcal{J}(\boldsymbol{\theta}, A)[\Delta_1^A],$$

i.e. the directional gradient of $\mathcal{J}(\boldsymbol{\theta}, A)$ with respect to $A$ in direction $\Delta_1^A$, and that this similarly holds for gradients with respect to other $t_i, s_j$, or higher-order derivatives. Thus, if we can show that $\mathcal{J}(\boldsymbol{\theta}_{\boldsymbol{s}}; A_{\boldsymbol{t}})$ is differentiable with respect to any $x_1, x_2, x_3, x_4 \in \{t_1, t_2, t_3, s_1, s_2, s_3\}$, and this holds for any choice of $\Delta_i^A, \Delta_j^{\boldsymbol{\theta}}$, then we have that $\mathcal{J}(\boldsymbol{\theta}, A)$ is four-times differentiable with respect to $\boldsymbol{\theta}$ and $A$. Furthermore, we can bound the operator norm of $\nabla_A\mathcal{J}(\boldsymbol{\theta}, A)$, by bounding the value of $\frac{\mathrm{d}}{\mathrm{d}t_1}\mathcal{J}(\boldsymbol{\theta}_{\boldsymbol{s}}; A_{\boldsymbol{t}})|_{\boldsymbol{t}=\boldsymbol{s}=0}$ for all $\Delta_1^A$ satisfying $\|\Delta_1^A\|_{\mathrm{op}} \leq 1$ (and we can similarly bound the operator norm of the higher order derivatives of $\mathcal{J}(\boldsymbol{\theta}, A)$).

$\mathcal{J}(\boldsymbol{\theta}; A)$ **is Differentiable.** Let $x_1, x_2, x_3, x_4 \in \{t_1, t_2, t_3, s_1, s_2, s_3\}$. We have

$$\frac{\mathrm{d}}{\mathrm{d}x_1}\mathcal{J}(\boldsymbol{\theta}_{\boldsymbol{s}}; A_{\boldsymbol{t}}) = \frac{\mathrm{d}}{\mathrm{d}x_1}\int \mathrm{cost}(\boldsymbol{\tau}) f_{A_{\boldsymbol{t}},\boldsymbol{\theta}_{\boldsymbol{s}}}(\boldsymbol{\tau})\mathrm{d}\boldsymbol{\tau}$$

$$= \int \frac{f_{A_t,\boldsymbol{\theta}_s}(\boldsymbol{\tau})}{f_{A_t,\boldsymbol{\theta}_s}(\boldsymbol{\tau})} \frac{\mathrm{d}}{\mathrm{d}x_1} f_{A_t,\boldsymbol{\theta}_s}(\boldsymbol{\tau})\mathrm{cost}(\boldsymbol{\tau})\mathrm{d}\boldsymbol{\tau}$$

$$= \int \frac{\mathrm{d}}{\mathrm{d}x_1} \log f_{A_t,\boldsymbol{\theta}_s}(\boldsymbol{\tau}) \cdot \mathrm{cost}(\boldsymbol{\tau}) f_{A_t,\boldsymbol{\theta}_s}(\boldsymbol{\tau})\mathrm{d}\boldsymbol{\tau}.$$

Differentiating this gives

$$\frac{\mathrm{d}}{\mathrm{d}x_2}\frac{\mathrm{d}}{\mathrm{d}x_1}\mathcal{J}(\boldsymbol{\theta}_s; A_t) = \frac{\mathrm{d}}{\mathrm{d}x_2}\int \frac{\mathrm{d}}{\mathrm{d}x_1}\log f_{A_t,\boldsymbol{\theta}_s}(\boldsymbol{\tau})\cdot\mathrm{cost}(\boldsymbol{\tau})f_{A_t,\boldsymbol{\theta}_s}(\boldsymbol{\tau})\mathrm{d}\boldsymbol{\tau}$$

$$= \int \left(\frac{\mathrm{d}}{\mathrm{d}x_1}\log f_{A_t,\boldsymbol{\theta}_s}(\boldsymbol{\tau})\right)\left(\frac{\mathrm{d}}{\mathrm{d}x_2}\log f_{A_t,\boldsymbol{\theta}_s}(\boldsymbol{\tau})\right)\cdot\mathrm{cost}(\boldsymbol{\tau})f_{A_t,\boldsymbol{\theta}_s}(\boldsymbol{\tau})\mathrm{d}\boldsymbol{\tau}$$

$$+ \int \frac{\mathrm{d}}{\mathrm{d}x_2}\frac{\mathrm{d}}{\mathrm{d}x_1}\log f_{A_t,\boldsymbol{\theta}_s}(\boldsymbol{\tau})\cdot\mathrm{cost}(\boldsymbol{\tau})f_{A_t,\boldsymbol{\theta}_s}(\boldsymbol{\tau})\mathrm{d}\boldsymbol{\tau},$$

and

$$\frac{\mathrm{d}}{\mathrm{d}x_3}\frac{\mathrm{d}}{\mathrm{d}x_2}\frac{\mathrm{d}}{\mathrm{d}x_1}\mathcal{J}(\boldsymbol{\theta}_s; A_t) = \int \left(\frac{\mathrm{d}}{\mathrm{d}x_1}\log f_{A_t,\boldsymbol{\theta}_s}(\boldsymbol{\tau})\right)\left(\frac{\mathrm{d}}{\mathrm{d}x_2}\log f_{A_t,\boldsymbol{\theta}_s}(\boldsymbol{\tau})\right)\left(\frac{\mathrm{d}}{\mathrm{d}x_3}\log f_{A_t,\boldsymbol{\theta}_s}(\boldsymbol{\tau})\right)\cdot\mathrm{cost}(\boldsymbol{\tau})f_{A_t,\boldsymbol{\theta}_s}(\boldsymbol{\tau})\mathrm{d}\boldsymbol{\tau}$$

$$+ \int \left(\frac{\mathrm{d}}{\mathrm{d}x_3}\frac{\mathrm{d}}{\mathrm{d}x_1}\log f_{A_t,\boldsymbol{\theta}_s}(\boldsymbol{\tau})\right)\left(\frac{\mathrm{d}}{\mathrm{d}x_2}\log f_{A_t,\boldsymbol{\theta}_s}(\boldsymbol{\tau})\right)\cdot\mathrm{cost}(\boldsymbol{\tau})f_{A_t,\boldsymbol{\theta}_s}(\boldsymbol{\tau})\mathrm{d}\boldsymbol{\tau}$$

$$+ \int \left(\frac{\mathrm{d}}{\mathrm{d}x_1}\log f_{A_t,\boldsymbol{\theta}_s}(\boldsymbol{\tau})\right)\left(\frac{\mathrm{d}}{\mathrm{d}x_3}\frac{\mathrm{d}}{\mathrm{d}x_2}\log f_{A_t,\boldsymbol{\theta}_s}(\boldsymbol{\tau})\right)\cdot\mathrm{cost}(\boldsymbol{\tau})f_{A_t,\boldsymbol{\theta}_s}(\boldsymbol{\tau})\mathrm{d}\boldsymbol{\tau}$$

$$+ \int \frac{\mathrm{d}}{\mathrm{d}x_3}\frac{\mathrm{d}}{\mathrm{d}x_2}\frac{\mathrm{d}}{\mathrm{d}x_1}\log f_{A_t,\boldsymbol{\theta}_s}(\boldsymbol{\tau})\cdot\mathrm{cost}(\boldsymbol{\tau})f_{A_t,\boldsymbol{\theta}_s}(\boldsymbol{\tau})\mathrm{d}\boldsymbol{\tau}$$

$$+ \int \left(\frac{\mathrm{d}}{\mathrm{d}x_2}\frac{\mathrm{d}}{\mathrm{d}x_1}\log f_{A_t,\boldsymbol{\theta}_s}(\boldsymbol{\tau})\right)\left(\frac{\mathrm{d}}{\mathrm{d}x_3}\log f_{A_t,\boldsymbol{\theta}_s}(\boldsymbol{\tau})\right)\cdot\mathrm{cost}(\boldsymbol{\tau})f_{A_t,\boldsymbol{\theta}_s}(\boldsymbol{\tau})\mathrm{d}\boldsymbol{\tau}.$$

The fourth derivative of $\mathcal{J}(\boldsymbol{\theta}; A)$ can be similarly calculated by differentiating $\frac{\mathrm{d}}{\mathrm{d}x_3}\frac{\mathrm{d}}{\mathrm{d}x_2}\frac{\mathrm{d}}{\mathrm{d}x_1}\mathcal{J}(\boldsymbol{\theta}_s; A_t)$; we omit it for brevity. We have

$$\log f_{A_t,\boldsymbol{\theta}_s}(\boldsymbol{\tau}) = \log \prod_{h=1}^{H} f_{A_t,\boldsymbol{\theta}_s}(\boldsymbol{x}_{h+1}^{\boldsymbol{\tau}} \mid \boldsymbol{\tau}_{1:h})$$

$$= \sum_{h=1}^{H} \log f_{\boldsymbol{w}}(\boldsymbol{x}_{h+1}^{\boldsymbol{\tau}} - A_t\boldsymbol{\phi}(\boldsymbol{x}_h^{\boldsymbol{\tau}}, \pi_h^{\boldsymbol{\theta}_s}(\boldsymbol{\tau}_{1:h})))$$

$$= \sum_{h=1}^{H} -\frac{1}{2\sigma_{\boldsymbol{w}}^2}\|\boldsymbol{x}_{h+1}^{\boldsymbol{\tau}} - A_t\boldsymbol{\phi}(\boldsymbol{x}_h^{\boldsymbol{\tau}}, \pi_h^{\boldsymbol{\theta}_s}(\boldsymbol{\tau}_{1:h}))\|_2^2 + C$$

for some $C$ which does not depend on $\boldsymbol{t}$ or $\boldsymbol{s}$. Given that $\boldsymbol{\phi}(\boldsymbol{x}, \boldsymbol{u})$ is four-times differentiable in $\boldsymbol{u}$ and $\pi_h^{\boldsymbol{\theta}_s}(\boldsymbol{\tau}_{1:h})$ is four-times differentiable in $\boldsymbol{x}$ (which hold by Assumption 4 and Assumption 5), it is clear that $\log f_{A_t,\boldsymbol{\theta}_s}(\boldsymbol{\tau})$ is four-times differentiable in $t_i$ or $s_i$, regardless of the choice of $\Delta_i^A$ or $\Delta_i^{\boldsymbol{\theta}}$. This proves the first result.

**Norm Bounds on Gradient.** Note that

$$\frac{\mathrm{d}}{\mathrm{d}t_i}\log f_{A_t,\boldsymbol{\theta}_s}(\boldsymbol{\tau})|_{\boldsymbol{t}=\boldsymbol{s}=0} = \sum_{h=1}^{H} \frac{1}{\sigma_{\boldsymbol{w}}^2}(\boldsymbol{x}_{h+1}^{\boldsymbol{\tau}} - A\boldsymbol{\phi}(\boldsymbol{x}_h^{\boldsymbol{\tau}}, \pi_h^{\boldsymbol{\theta}}(\boldsymbol{\tau}_{1:h})))^\top \cdot \Delta_i^A\boldsymbol{\phi}(\boldsymbol{x}_h^{\boldsymbol{\tau}}, \pi_h^{\boldsymbol{\theta}}(\boldsymbol{\tau}_{1:h})),$$

$$\frac{\mathrm{d}}{\mathrm{d}s_i}\log f_{A_t,\boldsymbol{\theta}_s}(\boldsymbol{\tau})|_{\boldsymbol{t}=\boldsymbol{s}=0} = \sum_{h=1}^{H} \frac{1}{\sigma_{\boldsymbol{w}}^2}(\boldsymbol{x}_{h+1}^{\boldsymbol{\tau}} - A\boldsymbol{\phi}(\boldsymbol{x}_h^{\boldsymbol{\tau}}, \pi_h^{\boldsymbol{\theta}}(\boldsymbol{\tau}_{1:h})))^\top \cdot A\nabla_{\boldsymbol{u}}\boldsymbol{\phi}(\boldsymbol{x}_h^{\boldsymbol{\tau}}, \pi_h^{\boldsymbol{\theta}}(\boldsymbol{\tau}_{1:h})) \cdot \nabla_{\boldsymbol{\theta}}\pi_h^{\boldsymbol{\theta}}(\boldsymbol{\tau}_{1:h}) \cdot \Delta_i^{\boldsymbol{\theta}}.$$

Furthermore, differentiating these expressions further with respect to $t_j$ or $s_j$ will simply yield higher-order derivates of $\boldsymbol{\phi}(\boldsymbol{x}, \boldsymbol{u})$ and $\pi_h^{\boldsymbol{\theta}}(\boldsymbol{\tau}_{1:h})$. Using the norm bounds on the gradient of $\boldsymbol{\phi}(\boldsymbol{x}, \boldsymbol{u})$ and $\pi_h^{\boldsymbol{\theta}}(\boldsymbol{\tau}_{1:h})$ given in Assumption 4 and Assumption 5, and the norm bound of $\boldsymbol{\phi}(\boldsymbol{x}, \boldsymbol{u})$ given in Assumption 1, we can then bound

$$\|\nabla_A^{(i)}\nabla_{\boldsymbol{\theta}}^{(j)}\log f_{A,\boldsymbol{\theta}}(\boldsymbol{\tau})\|_{\mathrm{op}} \le \mathrm{poly}(\|A\|_{\mathrm{op}}, B_{\boldsymbol{\phi}}, L_{\boldsymbol{\phi}}, L_{\boldsymbol{\theta}}, \sigma_{\boldsymbol{w}}^{-1}) \cdot \sum_{h=1}^{H}(1 + \|\boldsymbol{x}_{h+1}^{\boldsymbol{\tau}} - A\boldsymbol{\phi}(\boldsymbol{x}_h^{\boldsymbol{\tau}}, \pi_h^{\boldsymbol{\theta}}(\boldsymbol{\tau}_{1:h}))\|_2)$$

for $i, j \in \{0, 1, 2, 3, 4\}$ satisfying $1 \le i + j \le 4$ (where we have used the fact noted above that, to bound the operator norm of $\nabla_A^{(i)} \nabla_{\boldsymbol{\theta}}^{(j)} \log f_{A,\boldsymbol{\theta}}(\boldsymbol{\tau})$, it suffices to bound the directional gradient in every direction). It follows that we can bound

$$\|\nabla_A^{(i)} \nabla_{\boldsymbol{\theta}}^{(j)} \mathcal{J}(\boldsymbol{\theta}; A)\|_{\mathrm{op}}$$

$$\le \mathrm{poly}(\|A\|_{\mathrm{op}}, B_{\boldsymbol{\phi}}, L_{\boldsymbol{\phi}}, L_{\boldsymbol{\theta}}, \sigma_{\boldsymbol{w}}^{-1}) \cdot \int \left( \sum_{h=1}^{H} (1 + \|\boldsymbol{x}_{h+1}^{\boldsymbol{\tau}} - A\boldsymbol{\phi}(\boldsymbol{x}_h^{\boldsymbol{\tau}}, \pi_h^{\boldsymbol{\theta}}(\boldsymbol{\tau}_{1:h})) \|_2) \right)^4 \cdot \mathrm{cost}(\boldsymbol{\tau}) f_{A,\boldsymbol{\theta}}(\boldsymbol{\tau}) \mathrm{d}\boldsymbol{\tau}$$

$$\overset{(a)}{\le} \mathrm{poly}(\|A\|_{\mathrm{op}}, B_{\boldsymbol{\phi}}, L_{\boldsymbol{\phi}}, L_{\boldsymbol{\theta}}, \sigma_{\boldsymbol{w}}^{-1}) \cdot \sqrt{\int \mathrm{cost}(\boldsymbol{\tau})^2 f_{A,\boldsymbol{\theta}}(\boldsymbol{\tau}) \mathrm{d}\boldsymbol{\tau}}$$

$$\cdot \sqrt{\int \left( \sum_{h=1}^{H} (1 + \|\boldsymbol{x}_{h+1}^{\boldsymbol{\tau}} - A\boldsymbol{\phi}(\boldsymbol{x}_h^{\boldsymbol{\tau}}, \pi_h^{\boldsymbol{\theta}}(\boldsymbol{\tau}_{1:h})) \|_2) \right)^8 f_{A,\boldsymbol{\theta}}(\boldsymbol{\tau}) \mathrm{d}\boldsymbol{\tau}}$$

$$\overset{(b)}{\le} \mathrm{poly}(\|A\|_{\mathrm{op}}, B_{\boldsymbol{\phi}}, L_{\boldsymbol{\phi}}, L_{\boldsymbol{\theta}}, L_{\mathrm{cost}}, \sigma_{\boldsymbol{w}}^{-1}, H, d_{\boldsymbol{x}})$$

where $(a)$ follows from Cauchy-Schwarz, and $(b)$ follows from Lemma A.1 and Assumption 2, since we have assumed $A \in \mathcal{B}_{\mathrm{F}}(A_\star; r_{\mathrm{cost}}(A_\star))$. $\qquad \square$

*Proof of Proposition 5.* **Existence and Differentiability of $\boldsymbol{\theta}_\star$.** By Lemma D.1 we have that $\mathcal{J}(\boldsymbol{\theta}; A)$ is four-times differentiable in its arguments. By the Implicit Function Theorem, since $\lambda_{\min}(\nabla_{\boldsymbol{\theta}}^2 \mathcal{J}(\boldsymbol{\theta}; A_\star)|_{\boldsymbol{\theta}=\boldsymbol{\theta}_\star(A_\star)}) > 0$ by assumption, we have that there exists some $r'_{\boldsymbol{\theta}}(A_\star) > 0$ and unique function $\boldsymbol{\theta}_\star(A)$ defined on $\mathcal{B}_{\mathrm{F}}(A_\star; r'_{\boldsymbol{\theta}}(A_\star))$ such that $\nabla_{\boldsymbol{\theta}} \mathcal{J}(\boldsymbol{\theta}; A)|_{\boldsymbol{\theta}=\boldsymbol{\theta}_\star(A)} = 0$, and $\boldsymbol{\theta}_\star(A)$ is three-times differentiable (note that, while the Implicit Function Theorem is typically stated to give that the resulting function is only one-time differentiable, it can be extended to $k$-times differentiable, assuming the implicit equation is $k$-times differentiable [60]).

By Lemma D.1 and the continuity of eigenvalues, it follows that for $A$ close enough to $A_\star$, we have $\lambda_{\min}(\nabla_{\boldsymbol{\theta}}^2 \mathcal{J}(\boldsymbol{\theta}; A)|_{\boldsymbol{\theta}=\boldsymbol{\theta}_\star(A)}) \ge \frac{1}{2} \lambda_{\min}(\nabla_{\boldsymbol{\theta}}^2 \mathcal{J}(\boldsymbol{\theta}; A_\star)|_{\boldsymbol{\theta}=\boldsymbol{\theta}_\star(A_\star)}) > 0$. We can therefore apply the Implicit Function Theorem as above to any $A$ satisfying this, to get that there exists some unique $\widetilde{\boldsymbol{\theta}}_\star(A')$ defined for all $A'$ near $A$ such that $\nabla_{\boldsymbol{\theta}} \mathcal{J}(\boldsymbol{\theta}; A')|_{\boldsymbol{\theta}=\widetilde{\boldsymbol{\theta}}_\star(A')} = 0$ and $\widetilde{\boldsymbol{\theta}}_\star(A')$ is differentiable. By the uniqueness of $\boldsymbol{\theta}_\star(A)$ on $\mathcal{B}_{\mathrm{F}}(A_\star; r'_{\boldsymbol{\theta}}(A_\star))$, it follows that any $\widetilde{\boldsymbol{\theta}}_\star(A')$ defined in this way must be identical to $\boldsymbol{\theta}_\star(A)$ on $\mathcal{B}_{\mathrm{F}}(A_\star; r'_{\boldsymbol{\theta}}(A_\star))$ (assuming the regions on which they are defined overlaps). We can therefore define $\boldsymbol{\theta}_\star(A)$ to simply be the extension of $\boldsymbol{\theta}_\star(A)$ to all such $\widetilde{\boldsymbol{\theta}}_\star(A)$, defined for all $A$ near $A_\star$ such that $\lambda_{\min}(\nabla_{\boldsymbol{\theta}}^2 \mathcal{J}(\boldsymbol{\theta}; A)|_{\boldsymbol{\theta}=\boldsymbol{\theta}_\star(A)}) \ge \frac{1}{2} \lambda_{\min}(\nabla_{\boldsymbol{\theta}}^2 \mathcal{J}(\boldsymbol{\theta}; A_\star)|_{\boldsymbol{\theta}=\boldsymbol{\theta}_\star(A_\star)})$, and will have that $\boldsymbol{\theta}_\star(A)$ is three-times differentiable and satisfies $\nabla_{\boldsymbol{\theta}} \mathcal{J}(\boldsymbol{\theta}; A)|_{\boldsymbol{\theta}=\boldsymbol{\theta}_\star(A)} = 0$ for all such $A$.

We then choose $r_{\boldsymbol{\theta}}(A_\star)$ to be defined such that, for all $A \in \mathcal{B}_{\mathrm{F}}(A_\star; r_{\boldsymbol{\theta}}(A_\star))$, we have $\lambda_{\min}(\nabla_{\boldsymbol{\theta}}^2 \mathcal{J}(\boldsymbol{\theta}; A)|_{\boldsymbol{\theta}=\boldsymbol{\theta}_\star(A)}) \ge \frac{1}{2} \lambda_{\min}(\nabla_{\boldsymbol{\theta}}^2 \mathcal{J}(\boldsymbol{\theta}; A_\star)|_{\boldsymbol{\theta}=\boldsymbol{\theta}_\star(A_\star)})$. By Lemma D.1, we know that $\nabla_{\boldsymbol{\theta}}^2 \mathcal{J}(\boldsymbol{\theta}; A)$ is continuous and furthermore we know that eigenvalues are continuous. Using the gradient bounds given in Lemma D.1 to bound the Lipschitz constant of $\nabla_{\boldsymbol{\theta}}^2 \mathcal{J}(\boldsymbol{\theta}; A)$, it follows that we can take

$$r_{\boldsymbol{\theta}}(A_\star) = \mathrm{poly}\left( \frac{1}{\lambda_{\min}(\nabla_{\boldsymbol{\theta}}^2 \mathcal{J}(\boldsymbol{\theta}; A_\star)|_{\boldsymbol{\theta}=\boldsymbol{\theta}_\star(A_\star)})}, \|A_\star\|_{\mathrm{op}}, B_{\boldsymbol{\phi}}, L_{\boldsymbol{\phi}}, L_{\boldsymbol{\theta}}, L_{\mathrm{cost}}, \sigma_{\boldsymbol{w}}^{-1}, H, d_{\boldsymbol{x}} \right)^{-1}.$$

**Bounding Norm of Gradients.** Fix $A \in \mathcal{B}_{\mathrm{F}}(A_\star; r_{\boldsymbol{\theta}}(A_\star))$. We know that $\boldsymbol{\theta}_\star(A)$ satisfies

$$\nabla_{\boldsymbol{\theta}} \mathcal{J}(\boldsymbol{\theta}; A)|_{\boldsymbol{\theta}=\boldsymbol{\theta}_\star(A)} = 0.$$

We wish to differentiate $\boldsymbol{\theta}_\star(A)$ with respect to $A$, and bound the magnitude of up to the third derivative. Similar to the proof of Lemma D.1, we let $A_{\boldsymbol{t}} := A + t_1 \Delta_1^A + t_2 \Delta_2^A + t_3 \Delta_3^A$ for some $\Delta_i^A$ satisfying $\|\Delta_i^A\|_{\mathrm{op}} \le 1$. As noted in the proof of Lemma D.1, we have

$$\frac{\mathrm{d}}{\mathrm{d}t_i} \boldsymbol{\theta}_\star(A_{\boldsymbol{t}})|_{\boldsymbol{t}=0} = \nabla_A \boldsymbol{\theta}_\star(A)[\Delta_i^A]$$

(and similarly for higher-order derivatives). Thus, to show the result, it suffices to show that $\boldsymbol{\theta}_\star(A_{\boldsymbol{t}})$ is differentiable in $t_1, t_2, t_3$ for all $\Delta_i^A$, and to bound the magnitude of this derivative for all $\Delta_i^A$ with $\|\Delta_i^A\|_{\mathrm{op}} \leq 1$. We have

$$\frac{\mathrm{d}}{\mathrm{d}t_1}\nabla_{\boldsymbol{\theta}}\mathcal{J}(\boldsymbol{\theta}; A_{\boldsymbol{t}})|_{\boldsymbol{\theta}=\boldsymbol{\theta}_\star(A_{\boldsymbol{t}})}\big|_{\boldsymbol{t}=0} = 0$$

$$\implies \underbrace{\nabla_{A'}\nabla_{\boldsymbol{\theta}}\mathcal{J}(\boldsymbol{\theta}; A')|_{\boldsymbol{\theta}=\boldsymbol{\theta}_\star(A), A'=A}[\Delta_1^A]}_{=:G_1(A,\Delta_1^A)} + \nabla_{\boldsymbol{\theta}}^2\mathcal{J}(\boldsymbol{\theta}; A)|_{\boldsymbol{\theta}=\boldsymbol{\theta}_\star(A)} \cdot \nabla_A\boldsymbol{\theta}_\star(A)[\Delta_1^A] = 0 \quad \text{(D.1)}$$

which implies

$$\nabla_A\boldsymbol{\theta}_\star(A)[\Delta_1^A] = -\left(\nabla_{\boldsymbol{\theta}}^2\mathcal{J}(\boldsymbol{\theta}; A)|_{\boldsymbol{\theta}=\boldsymbol{\theta}_\star(A)}\right)^{-1} \cdot G_1(A, \Delta_1^A)$$

which is well-defined since we have assumed that $\nabla_{\boldsymbol{\theta}}^2\mathcal{J}(\boldsymbol{\theta}; A)|_{\boldsymbol{\theta}=\boldsymbol{\theta}_\star(A)}$ is full-rank, and $\mathcal{J}$ is differentiable in both its arguments by Lemma D.1. To compute the second derivative of $\boldsymbol{\theta}_\star$, we differentiate through (D.1) which gives

$$\frac{\mathrm{d}}{\mathrm{d}t_2}\left(G_1(A_{\boldsymbol{t}}, \Delta_1^A) + \nabla_{\boldsymbol{\theta}}^2\mathcal{J}(\boldsymbol{\theta}; A_{\boldsymbol{t}})|_{\boldsymbol{\theta}=\boldsymbol{\theta}_\star(A_{\boldsymbol{t}})} \cdot \nabla_A\boldsymbol{\theta}_\star(A_{\boldsymbol{t}})[\Delta_1^A]\right)\big|_{\boldsymbol{t}=0} = 0$$

$$\implies \underbrace{\frac{\mathrm{d}}{\mathrm{d}t_2}\left(G_1(A_{\boldsymbol{t}}, \Delta_1^A) + \nabla_{\boldsymbol{\theta}}^2\mathcal{J}(\boldsymbol{\theta}; A_{\boldsymbol{t}})|_{\boldsymbol{\theta}=\boldsymbol{\theta}_\star(A_{\boldsymbol{t}})} \cdot \nabla_A\boldsymbol{\theta}_\star(A)[\Delta_1^A]\right)\big|_{\boldsymbol{t}=0}}_{=:G_2(A,\Delta_1^A,\Delta_2^A)}$$

$$+ \nabla_{\boldsymbol{\theta}}^2\mathcal{J}(\boldsymbol{\theta}; A)|_{\boldsymbol{\theta}=\boldsymbol{\theta}_\star(A)} \cdot \nabla_A^2\boldsymbol{\theta}_\star(A)[\Delta_1^A, \Delta_2^A] = 0.$$

Note that $G_2(A, \Delta_1^A, \Delta_2^A)$ involves at most a third-order derivative of $\mathcal{J}(\boldsymbol{\theta}; A)$ and first-order derivative of $\boldsymbol{\theta}_\star(A)$, both of which we know exist by Lemma D.1 and what we showed above. This then further implies

$$\nabla_A^2\boldsymbol{\theta}_\star(A)[\Delta_1^A, \Delta_2^A] = -\left(\nabla_{\boldsymbol{\theta}}^2\mathcal{J}(\boldsymbol{\theta}; A)|_{\boldsymbol{\theta}=\boldsymbol{\theta}_\star(A)}\right)^{-1} \cdot G_2(A, \Delta_1^A, \Delta_2^A),$$

which is well-defined since we have assumed that $\nabla_{\boldsymbol{\theta}}^2\mathcal{J}(\boldsymbol{\theta}; A)|_{\boldsymbol{\theta}=\boldsymbol{\theta}_\star(A)}$ is full-rank. Finally, we compute

$$\frac{\mathrm{d}}{\mathrm{d}t_3}\left(G_2(A_{\boldsymbol{t}}, \Delta_1^A, \Delta_2^A) + \nabla_{\boldsymbol{\theta}}^2\mathcal{J}(\boldsymbol{\theta}; A_{\boldsymbol{t}})|_{\boldsymbol{\theta}=\boldsymbol{\theta}_\star(A_{\boldsymbol{t}})} \cdot \nabla_A^2\boldsymbol{\theta}_\star(A_{\boldsymbol{t}})[\Delta_1^A, \Delta_2^A]\right)\big|_{\boldsymbol{t}=0} = 0$$

$$\implies \underbrace{\frac{\mathrm{d}}{\mathrm{d}t_3}\left(G_2(A_{\boldsymbol{t}}, \Delta_1^A, \Delta_2^A) + \nabla_{\boldsymbol{\theta}}^2\mathcal{J}(\boldsymbol{\theta}; A_{\boldsymbol{t}})|_{\boldsymbol{\theta}=\boldsymbol{\theta}_\star(A_{\boldsymbol{t}})} \cdot \nabla_A^2\boldsymbol{\theta}_\star(A)[\Delta_1^A, \Delta_2^A]\right)\big|_{\boldsymbol{t}=0}}_{=:G_3(A,\Delta_1^A,\Delta_2^A,\Delta_3^A)}$$

$$+ \nabla_{\boldsymbol{\theta}}^2\mathcal{J}(\boldsymbol{\theta}; A)|_{\boldsymbol{\theta}=\boldsymbol{\theta}_\star(A)} \cdot \nabla_A^3\boldsymbol{\theta}_\star(A)[\Delta_1^A, \Delta_2^A, \Delta_3^A] = 0.$$

Note that $G_3(A, \Delta_1^A, \Delta_2^A, \Delta_3^A)$ involves at most a fourth-order derivative of $\mathcal{J}(\boldsymbol{\theta}; A)$ and second-order derivative of $\boldsymbol{\theta}_\star(A)$, both of which we know exist by Lemma D.1 and what we showed above. We therefore have

$$\nabla_A^3\boldsymbol{\theta}_\star(A)[\Delta_1^A, \Delta_2^A, \Delta_3^A] = -\left(\nabla_{\boldsymbol{\theta}}^2\mathcal{J}(\boldsymbol{\theta}; A)|_{\boldsymbol{\theta}=\boldsymbol{\theta}_\star(A)}\right)^{-1} \cdot G_3(A, \Delta_1^A, \Delta_2^A, \Delta_3^A)$$

which is well-defined since we have assumed that $\nabla_{\boldsymbol{\theta}}^2\mathcal{J}(\boldsymbol{\theta}; A)|_{\boldsymbol{\theta}=\boldsymbol{\theta}_\star(A)}$ is full-rank. As each of these expressions is defined for all choice of $\Delta_i^A$, the differentiability of $\boldsymbol{\theta}_\star(A)$ follows.

Note that the above expressions for $\nabla_A\boldsymbol{\theta}_\star(A)[\Delta_1^A], \nabla_A^2\boldsymbol{\theta}_\star(A)[\Delta_1^A, \Delta_2^A]$, and $\nabla_A^3\boldsymbol{\theta}_\star(A)[\Delta_1^A, \Delta_2^A, \Delta_3^A]$ all depend on at most a fourth derivative of $\mathcal{J}(\boldsymbol{\theta}; A)$, as well as $\left(\nabla_{\boldsymbol{\theta}}^2\mathcal{J}(\boldsymbol{\theta}; A)|_{\boldsymbol{\theta}=\boldsymbol{\theta}_\star(A)}\right)^{-1}$. The norm bounds are then a direct consequence of Lemma D.1.

$\square$

*Proof of Proposition 6.* By Lemma D.1 we have that $\mathcal{J}(\boldsymbol{\theta}; A)$ is four-times differentiable in its arguments. Since we have assumed $\nabla_{\boldsymbol{\theta}}^2\mathcal{J}(\boldsymbol{\theta}; A_\star)|_{\boldsymbol{\theta}=\boldsymbol{\theta}_\star(A_\star)} \succ 0$, by the Implicit Function Theorem [60], it follows that there exists some $r'_{\boldsymbol{\theta}} > 0$ and mapping $\widetilde{\boldsymbol{\theta}}(A)$ such that, for all $A \in \mathcal{B}_{\mathrm{F}}(A_\star; r'_{\boldsymbol{\theta}})$, $\nabla_{\boldsymbol{\theta}}\mathcal{J}(\boldsymbol{\theta}; A)|_{\boldsymbol{\theta}=\widetilde{\boldsymbol{\theta}}(A)} = 0$, and $\widetilde{\boldsymbol{\theta}}(A)$ is three-times differentiable.

Our goal is now to show that $\widetilde{\boldsymbol{\theta}}(A) = \boldsymbol{\theta}_\star(A)$ for $A$ close enough to $A_\star$. By the continuity of eigenvalues, $\mathcal{J}(\boldsymbol{\theta}; A)$, and $\widetilde{\boldsymbol{\theta}}(A)$, we have that there exists some $r$ and $r_{\boldsymbol{\theta}}''$ such that, for all $\boldsymbol{\theta} \in \mathcal{B}_{\mathrm{F}}(\boldsymbol{\theta}_\star(A_\star); r)$ and $A \in \mathcal{B}_{\mathrm{F}}(A_\star; r_{\boldsymbol{\theta}}'')$, we have $\nabla_{\boldsymbol{\theta}}^2 \mathcal{J}(\boldsymbol{\theta}; A) \succ 0$ and, furthermore, $\widetilde{\boldsymbol{\theta}}(A) \in \mathcal{B}_2(\boldsymbol{\theta}_\star(A_\star); r/2)$ for all $A \in \mathcal{B}_{\mathrm{F}}(A_\star; r_{\boldsymbol{\theta}}'')$. This implies that $\widetilde{\boldsymbol{\theta}}(A)$ is strict local minimum of $\mathcal{J}(\boldsymbol{\theta}; A)$ and, in particular, that

$$\mathcal{J}(\boldsymbol{\theta}; A) > \mathcal{J}(\widetilde{\boldsymbol{\theta}}(A); A), \quad \forall \boldsymbol{\theta} \in \mathcal{B}_2(\boldsymbol{\theta}_\star(A_\star); r), \boldsymbol{\theta} \neq \widetilde{\boldsymbol{\theta}}(A).$$

Let $\Delta^\star := \min_{\boldsymbol{\theta} \notin \mathcal{B}_2(\boldsymbol{\theta}_\star(A_\star); r)} \mathcal{J}(\boldsymbol{\theta}; A_\star) - \mathcal{J}(\boldsymbol{\theta}_\star(A_\star); A_\star)$ and note that, since we have assumed the global minimum of $\mathcal{J}(\boldsymbol{\theta}; A_\star)$ is unique, we have $\Delta^\star > 0$.

Fix some $A \in \mathcal{B}_{\mathrm{F}}(A_\star; r_{\boldsymbol{\theta}}'')$ and assume that $\widetilde{\boldsymbol{\theta}}(A)$ is not the global minimum of $\mathcal{J}(\boldsymbol{\theta}; A)$. This implies that $\boldsymbol{\theta}_\star(A)$, the global minimum of $\mathcal{J}(\boldsymbol{\theta}; A)$, is outside of $\mathcal{B}_2(\boldsymbol{\theta}_\star(A_\star); r)$. Furthermore, by the continuity of $\mathcal{J}(\boldsymbol{\theta}; A)$, we have, for some $L, L' > 0$,

$$\begin{aligned}
\mathcal{J}(\boldsymbol{\theta}_\star(A); A_\star) &\leq \mathcal{J}(\boldsymbol{\theta}_\star(A); A) + L\|A - A_\star\|_{\mathrm{F}} \\
&\leq \mathcal{J}(\widetilde{\boldsymbol{\theta}}(A); A) + L\|A - A_\star\|_{\mathrm{F}} \\
&\leq \mathcal{J}(\widetilde{\boldsymbol{\theta}}(A); A_\star) + 2L\|A - A_\star\|_{\mathrm{F}} \\
&\leq \mathcal{J}(\boldsymbol{\theta}_\star(A_\star); A_\star) + 2L\|A - A_\star\|_{\mathrm{F}} + L\|\widetilde{\boldsymbol{\theta}}(A) - \boldsymbol{\theta}_\star(A_\star)\|_2 \\
&\leq \mathcal{J}(\boldsymbol{\theta}_\star(A_\star); A_\star) + L'\|A - A_\star\|_{\mathrm{F}}.
\end{aligned}$$

This implies that

$$L'r_{\boldsymbol{\theta}}'' \geq L'\|A - A_\star\|_{\mathrm{F}} \geq \mathcal{J}(\boldsymbol{\theta}_\star(A); A_\star) - \mathcal{J}(\boldsymbol{\theta}_\star(A_\star); A_\star) \geq \Delta^\star.$$

However, for $r_{\boldsymbol{\theta}}''$ small enough, this is a contradiction. Thus, it follows that $\widetilde{\boldsymbol{\theta}}(A)$ is the global minimum of $\mathcal{J}(\boldsymbol{\theta}; A)$, so $\widetilde{\boldsymbol{\theta}}(A) = \boldsymbol{\theta}_\star(A)$.

The result then follows since we already have that $\widetilde{\boldsymbol{\theta}}(A)$ is three-times differentiable and satisfies $\nabla_{\boldsymbol{\theta}} \mathcal{J}(\boldsymbol{\theta}; A)|_{\boldsymbol{\theta}=\widetilde{\boldsymbol{\theta}}(A)} = 0$, and by taking $r_{\boldsymbol{\theta}}(A_\star)$ to be the minimum of $r_{\boldsymbol{\theta}}'$ and $r_{\boldsymbol{\theta}}''$. The boundedness of $L_{\pi_\star}$ follows as in the proof of Proposition 5.

$\square$

*Proof of Proposition 7.* Note that, by convexity and the KKT conditions, the solutions to $\arg\min_{\boldsymbol{\theta} \in \mathbb{R}^{d_{\boldsymbol{\theta}}}} \mathcal{J}(\boldsymbol{\theta}; A)$ are described by

$$\nabla_{\boldsymbol{\theta}} \mathcal{J}(\boldsymbol{\theta}; A)|_{\boldsymbol{\theta}=\boldsymbol{\theta}_\star(A)} = 0.$$

Thus, an equivalent definition for $\boldsymbol{\theta}_\star(A)$ is that it satisfies $\nabla_{\boldsymbol{\theta}} \mathcal{J}(\boldsymbol{\theta}; A)|_{\boldsymbol{\theta}=\boldsymbol{\theta}_\star(A)} = 0$. Assumption 13 can then be shown to hold by an argument analogous to Proposition 5. $\square$

*Proof of Lemma D.2.* Let $A(t) = t\widehat{A} + (1-t)A_\star$ and $g(t) := \mathcal{J}(\boldsymbol{\theta}_\star(A(t)); A_\star)$. By Lemma D.1 and under Assumption 5, we have that both $\mathcal{J}(\boldsymbol{\theta}; A)$ and $\boldsymbol{\theta}_\star(A)$ are three-times differentiable for all $A = t\widehat{A} + (1-t)A_\star, t \in [0, 1]$, so it follows that $g(t)$ is three-times differentiable in $t$. We can therefore apply Taylor's Theorem to expand $g(1)$ about the point $t = 0$ to get:

$$\begin{aligned}
g(1) = g(0) &+ \nabla_{\boldsymbol{\theta}} \mathcal{J}(\boldsymbol{\theta}; A_\star)|_{\boldsymbol{\theta}=\boldsymbol{\theta}_\star(A_\star)} \cdot \nabla_A \boldsymbol{\theta}_\star(A)|_{A=A_\star}[\widehat{A} - A_\star] \\
&+ \nabla_A \boldsymbol{\theta}_\star(A)|_{A=A_\star}^\top \nabla_{\boldsymbol{\theta}}^2 \mathcal{J}(\boldsymbol{\theta}; A_\star)|_{\boldsymbol{\theta}=\boldsymbol{\theta}_\star(A_\star)} \nabla_A \boldsymbol{\theta}_\star(A)|_{A=A_\star}[\widehat{A} - A_\star, \widehat{A} - A_\star] \\
&+ \nabla_{\boldsymbol{\theta}} \mathcal{J}(\boldsymbol{\theta}; A_\star)|_{\boldsymbol{\theta}=\boldsymbol{\theta}_\star(A_\star)} \cdot \nabla_A^2 \boldsymbol{\theta}_\star(A)|_{A=A_\star}[\widehat{A} - A_\star, \widehat{A} - A_\star] \\
&+ \nabla_A^3 \mathcal{J}(\boldsymbol{\theta}_\star(A); A_\star)|_{A=A'}[\widehat{A} - A_\star, \widehat{A} - A_\star, \widehat{A} - A_\star]
\end{aligned}$$

where $A' = A(t')$ for some $t' \in [0, 1]$. Under Assumption 13, we have that $\nabla_{\boldsymbol{\theta}} \mathcal{J}(\boldsymbol{\theta}; A_\star)|_{\boldsymbol{\theta}=\boldsymbol{\theta}_\star(A_\star)} = 0$, which implies that, plugging in the definition of $g(1)$ and $g(0)$,

$$\begin{aligned}
\mathcal{J}(\boldsymbol{\theta}_\star(\widehat{A}); A_\star) = \mathcal{J}(\boldsymbol{\theta}_\star(A_\star); A_\star) &+ \nabla_A \boldsymbol{\theta}_\star(A)|_{A=A_\star}^\top \nabla_{\boldsymbol{\theta}}^2 \mathcal{J}(\boldsymbol{\theta}; A_\star)|_{\boldsymbol{\theta}=\boldsymbol{\theta}_\star(A_\star)} \nabla_A \boldsymbol{\theta}_\star(A)|_{A=A_\star}[\widehat{A} - A_\star, \widehat{A} - A_\star] \\
&+ \nabla_A^3 \mathcal{J}(\boldsymbol{\theta}_\star(A); A_\star)|_{A=A'}[\widehat{A} - A_\star, \widehat{A} - A_\star, \widehat{A} - A_\star].
\end{aligned}$$

We can bound
$$|\nabla_A^3 \mathcal{J}(\boldsymbol{\theta}_\star(A); A_\star)|_{A=A'}[\widehat{A} - A_\star, \widehat{A} - A_\star, \widehat{A} - A_\star]| \le \|\nabla_A^3 \mathcal{J}(\boldsymbol{\theta}_\star(A); A_\star)|_{A=A'}\|_{\mathrm{op}} \cdot \|\widehat{A} - A_\star\|_{\mathrm{op}}^3.$$

The expression for $\nabla_A^3 \mathcal{J}(\boldsymbol{\theta}_\star(A); A_\star)$ contains up to the third derivative of both $\mathcal{J}(\boldsymbol{\theta}; A_\star)$ and $\boldsymbol{\theta}_\star(A)$. By Lemma D.1 and under Assumption 13, since $A' \in \mathcal{B}_{\mathrm{F}}(A_\star; \min\{r_{\mathrm{cost}}(A_\star), r_{\boldsymbol{\theta}}(A_\star)\})$ by construction, we can then bound
$$\|\nabla_A^3 \mathcal{J}(\boldsymbol{\theta}_\star(A); A_\star)|_{A=A'}\|_{\mathrm{op}} \le \mathrm{poly}(L_{\pi_\star}, \|A_\star\|_{\mathrm{op}}, B_{\boldsymbol{\phi}}, L_{\boldsymbol{\phi}}, L_{\boldsymbol{\theta}}, L_{\mathrm{cost}}, \sigma_{\boldsymbol{w}}^{-1}, H, d_{\boldsymbol{x}}).$$

The result follows by the definition of $\mathcal{H}(A_\star)$. $\qquad\square$

*Proof of Lemma D.3.* Recall that $\mathcal{H}(\widehat{A}) = \nabla_A^2 \mathcal{J}(\boldsymbol{\theta}_\star(A); \widehat{A})\big|_{A=\widehat{A}}$. To prove this, we will use that this is differentiable by Lemma D.1, and will apply Taylor's Theorem.

First, note that by Taylor's Theorem we have
$$\nabla_A^2 \mathcal{J}(\boldsymbol{\theta}_\star(A); \widehat{A})|_{A=\widehat{A}} = \nabla_A^2 \mathcal{J}(\boldsymbol{\theta}_\star(A); \widehat{A})|_{A=A_\star} + \nabla_A^3 \mathcal{J}(\boldsymbol{\theta}_\star(A); \widehat{A})|_{A=A'}[\widehat{A} - A_\star]$$

for $A' = t\widehat{A} + (1-t)A_\star$ for some $t \in [0,1]$. The third derivative of $\mathcal{J}(\boldsymbol{\theta}_\star(A); \widehat{A})$ will involve up to the third derivative of both $\mathcal{J}(\boldsymbol{\theta}; \widehat{A})$ and $\boldsymbol{\theta}_\star(A)$, so using Lemma D.1 and Assumption 13, since $A' \in \mathcal{B}_{\mathrm{F}}(A_\star; \min\{r_{\mathrm{cost}}(A_\star), r_{\boldsymbol{\theta}}(A_\star)\})$ by assumption, we can bound
$$\|\nabla_A^3 \mathcal{J}(\boldsymbol{\theta}_\star(A); \widehat{A})|_{A=A'}[\widehat{A} - A_\star]\|_{\mathrm{op}} \le \mathrm{poly}(L_{\pi_\star}, \|A_\star\|_{\mathrm{op}}, B_{\boldsymbol{\phi}}, L_{\boldsymbol{\phi}}, L_{\boldsymbol{\theta}}, L_{\mathrm{cost}}, \sigma_{\boldsymbol{w}}^{-1}, H, d_{\boldsymbol{x}}) \cdot \|\widehat{A} - A_\star\|_{\mathrm{op}}.$$

Next, we wish to relate $\nabla_A^2 \mathcal{J}(\boldsymbol{\theta}_\star(A); \widehat{A})|_{A=A_\star}$ to $\nabla_A^2 \mathcal{J}(\boldsymbol{\theta}_\star(A); A_\star)|_{A=A_\star} = \mathcal{H}(A_\star)$. Again applying Taylor's Theorem, we have
$$\nabla_A^2 \mathcal{J}(\boldsymbol{\theta}_\star(A); \widehat{A})|_{A=A_\star} = \nabla_A^2 \mathcal{J}(\boldsymbol{\theta}_\star(A); A_\star)|_{A=A_\star} + \nabla_{A'}\nabla_A^2 \mathcal{J}(\boldsymbol{\theta}_\star(A); A')|_{A=A_\star, A'=A''}[\widehat{A} - A_\star]$$

for $A'' = t\widehat{A} + (1-t)A_\star$ for some $t \in [0,1]$. By Lemma D.1 and Assumption 13, we can bound
$$\|\nabla_{A'}\nabla_A^2 \mathcal{J}(\boldsymbol{\theta}_\star(A); A')|_{A=A_\star, A'=A''}[\widehat{A} - A_\star]\|_{\mathrm{op}}$$
$$\le \mathrm{poly}(L_{\pi_\star}, \|A_\star\|_{\mathrm{op}}, L_{\boldsymbol{\phi}}, L_{\boldsymbol{\theta}}, L_{\mathrm{cost}}, \sigma_{\boldsymbol{w}}^{-1}, H, d_{\boldsymbol{x}}) \cdot \|\widehat{A} - A_\star\|_{\mathrm{op}}.$$

The result follows. $\qquad\square$

**Lemma D.4.** *Under Assumptions 1, 2, 4, 5 and 13, for all $A \in \mathcal{B}_{\mathrm{F}}(A_\star; \min\{r_{\mathrm{cost}}(A_\star), r_{\boldsymbol{\theta}}(A_\star)\})$, we can bound*
$$\|\mathcal{H}(A)\|_{\mathrm{op}} \le \mathrm{poly}(\|A_\star\|_{\mathrm{op}}, B_{\boldsymbol{\phi}}, L_{\boldsymbol{\phi}}, L_{\boldsymbol{\theta}}, L_{\mathrm{cost}}, L_{\pi_\star}, \sigma_{\boldsymbol{w}}^{-1}, H, d_{\boldsymbol{x}})$$

*Proof.* Recall that $\mathcal{H}(\widehat{A}) = \nabla_A^2 \mathcal{J}(\boldsymbol{\theta}_\star(A); \widehat{A})\big|_{A=\widehat{A}}$. The bound then follows from Lemma D.1 and Assumption 13. $\qquad\square$

# E High-Probability Regret Bounds in Nonlinear Systems

In this section, we modify the proof the main result of [17] slightly to show a high probability regret bound for $\mathrm{LC}^3$. For the sake of brevity, we omit details that are identical to the proof given in [17]. We will need the following assumption.

**Assumption 14** (Bounded Cost). *We assume that, for all trajectories $\boldsymbol{\tau}$, we have $\mathrm{cost}(\boldsymbol{\tau}) \le c_{\max}$.*

We adopt the notation used in this work, modifying somewhat the notation from [17]. In particular, we let $\mathcal{J}(\pi; A)$ denote the expected cost of playing policy $\pi$ under system $A$, and we set
$$\boldsymbol{\Sigma}_t = \sum_{s=1}^{t} \sum_{h=1}^{H} \boldsymbol{\phi}(\boldsymbol{x}_h^t, \boldsymbol{u}_h^t)\boldsymbol{\phi}(\boldsymbol{x}_h^t, \boldsymbol{u}_h^t)^\top + \lambda I$$

denote the covariates obtained by the first $t$ episodes of $\mathrm{LC}^3$ (plus a regularizer). We let $\pi^t$ denote the policy played at episode $t$ of $\mathrm{LC}^3$. For a policy set $\Pi$, we define regret as
$$\mathcal{R}_T(\Pi) := \sum_{t=1}^{T} \mathcal{J}(\pi^t; A_\star) - T \cdot \min_{\pi \in \Pi} \mathcal{J}(\pi; A_\star).$$

We will also denote $\pi_\star := \arg\min_{\pi \in \Pi} \mathcal{J}(\pi; A_\star)$.

In addition to these notational changes, we modify $\text{LC}^3$ slightly to use the parameter

$$\beta^t := \sqrt{\lambda} B_A + \sqrt{8 d_{\boldsymbol{x}} \log 5 + 8 \log(T \det(\boldsymbol{\Sigma}_t) \det(\boldsymbol{\Sigma}_0)^{-1}/\delta)}$$

in the construction of the confidence set, $\text{BALL}^t$.

Besides the aforementioned changes, in the following proofs we adopt the same notation as [17]. We have the following result.

**Theorem 6.** *Under Assumptions 1 and 14 and with any policy class $\Pi$, with probability at least $1 - \delta$, $\text{LC}^3$ has regret bounded as*

$$\mathcal{R}_T(\Pi) \leq C \cdot c_{\max} H \sqrt{d_{\boldsymbol{\phi}} \cdot (d_{\boldsymbol{\phi}} + d_{\boldsymbol{x}} + B_A + \log \frac{1}{\delta}) \cdot T} \cdot \log\left(1 + B_{\boldsymbol{\phi}} HT/\sigma_{\boldsymbol{w}}\right)$$

*for a universal constant $C$.*

*Proof of Theorem 6.* By Lemma E.2, we have that the event $\mathcal{E}_1$ holds with probability at least $1 - \delta$. We therefore assume $\mathcal{E}_1$ holds for the remainder of the proof.

By the definition of the confidence set in $\text{LC}^3$, on $\mathcal{E}_1$ we have that $A_\star$ is the in confidence set for all $t \leq T$. It follows that on $\mathcal{E}_1$,

$$
\begin{aligned}
\mathcal{R}_T &= \sum_{t=1}^T \left[ \mathcal{J}(\pi^t; A_\star) - \mathcal{J}(\pi_\star; A_\star) \right] \\
&\overset{(a)}{\leq} \sum_{t=1}^T \left[ \mathcal{J}(\pi^t; A_\star) - \mathcal{J}(\pi_\star; \widehat{A}^t) \right] \\
&\overset{(b)}{\leq} \sum_{t=1}^T c_{\max} \cdot \mathbb{E}_{A_\star, \pi^t} \left[ \sum_{h=1}^H \min \left\{ \frac{1}{\sigma_{\boldsymbol{w}}} \| (A_\star - \widehat{A}^t) \cdot \boldsymbol{\phi}(\boldsymbol{x}_h, \boldsymbol{u}_h) \|_2, 1 \right\} \right]
\end{aligned}
\tag{E.1}
$$

where $(a)$ follows from the optimistic property of $\text{LC}^3$ when $A_\star \in \text{BALL}^t$, and $(b)$ follows from Lemma E.1. On $\mathcal{E}_1$, we have

$$
\begin{aligned}
\| (A_\star - \widehat{A}^t) \boldsymbol{\phi}(\boldsymbol{x}_h, \boldsymbol{u}_h) \|_2 &\leq \| (A_\star - \widehat{A}^t) \boldsymbol{\Sigma}_t^{1/2} \|_2 \| \boldsymbol{\Sigma}_t^{-1/2} \boldsymbol{\phi}(\boldsymbol{x}_h, \boldsymbol{u}_h) \|_2 \\
&\leq \left( \| (A_\star - \bar{A}^t) \boldsymbol{\Sigma}_t^{1/2} \|_2 + \| (\bar{A}^t - \widehat{A}^t) \boldsymbol{\Sigma}_t^{1/2} \|_2 \right) \cdot \| \boldsymbol{\Sigma}_t^{-1/2} \boldsymbol{\phi}(\boldsymbol{x}_h, \boldsymbol{u}_h) \|_2 \\
&\leq 2\beta^t \| \boldsymbol{\phi}(\boldsymbol{x}_h, \boldsymbol{u}_h) \|_{\boldsymbol{\Sigma}_t^{-1}}
\end{aligned}
$$

where the last inequality follows from the definition of $\text{BALL}^t$ since $\widehat{A}^t \in \text{BALL}^t$ by construction, and by the definition of $\mathcal{E}_1$. This gives

$$(\text{E.1}) \leq \sum_{t=1}^T c_{\max} \cdot \mathbb{E}_{A_\star, \pi^t} \left[ \sum_{h=1}^H \min \left\{ \frac{2\beta^t}{\sigma_{\boldsymbol{w}}} \| \boldsymbol{\phi}(\boldsymbol{x}_h, \boldsymbol{u}_h) \|_{\boldsymbol{\Sigma}_t^{-1}}, 1 \right\} \right].$$

By Lemma E.3, with probability $1 - \delta$ we can bound this as

$$\leq \underbrace{\frac{2 c_{\max} \beta^T}{\sigma_{\boldsymbol{w}}} \cdot \sum_{t=1}^T \sum_{h=1}^H \min \left\{ \| \boldsymbol{\phi}(\boldsymbol{x}_h^t, \boldsymbol{u}_h^t) \|_{\boldsymbol{\Sigma}_t^{-1}}, 1 \right\}}_{(a)} + 4 c_{\max} H \sqrt{T \log 1/\delta}.$$

By Cauchy-Schwarz, we can bound $(a)$ as

$$(a) \leq \frac{2 c_{\max} \beta^T}{\sigma_{\boldsymbol{w}}} \cdot \sqrt{T} \sqrt{\sum_{t=1}^T \sum_{h=1}^H \min \left\{ \| \boldsymbol{\phi}(\boldsymbol{x}_h^t, \boldsymbol{u}_h^t) \|_{\boldsymbol{\Sigma}_t^{-1}}^2, 1 \right\}}.$$

We have

$$\sum_{t=1}^{T}\sum_{h=1}^{H}\min\left\{\|\phi(\boldsymbol{x}_h^t,\boldsymbol{u}_h^t)\|_{\boldsymbol{\Sigma}_t^{-1}}^2,1\right\}H \le \sum_{t=1}^{T}\min\left\{\sum_{h=1}^{H}\|\phi(\boldsymbol{x}_h^t,\boldsymbol{u}_h^t)\|_{\boldsymbol{\Sigma}_t^{-1}}^2,1\right\}$$
$$\le 2H\log(\det(\boldsymbol{\Sigma}_T)\det(\boldsymbol{\Sigma}_0)^{-1})$$

where the last inequality uses Lemma B.6 of [17]. Putting all of this together, we have shown that with probability at least $1-2\delta$, we have

$$\mathcal{R}_T \le \frac{2c_{\max}\beta^T}{\sigma_{\boldsymbol{w}}}\cdot\sqrt{T}\cdot\sqrt{2H\log(\det(\boldsymbol{\Sigma}_T)\det(\boldsymbol{\Sigma}_0)^{-1})} + 4c_{\max}H\sqrt{T\log 1/\delta}.$$

It remains to bound $\beta^T$ and $\log(\det(\boldsymbol{\Sigma}_T)\det(\boldsymbol{\Sigma}_0)^{-1})$. We have $\boldsymbol{\Sigma}_0 = \lambda I$, so $\det(\boldsymbol{\Sigma}_0) = \lambda^{d_\phi}$. Furthermore, if $\|\phi(\boldsymbol{x},\boldsymbol{u})\|_2 \le B_\phi$, then we can bound $\det(\boldsymbol{\Sigma}_T) \le (\lambda + B_\phi^2 TH)^{d_\phi}$. Putting this together we have

$$\log(\det(\boldsymbol{\Sigma}_T)\det(\boldsymbol{\Sigma}_0)^{-1}) \le d_\phi \cdot \log(1 + B_\phi^2 TH/\lambda).$$

Recalling that

$$\beta^T = \sqrt{\lambda}B_A + \sigma_{\boldsymbol{w}}\sqrt{8d_{\boldsymbol{x}}\log 5 + 8\log(T\det(\boldsymbol{\Sigma}_T)\det(\boldsymbol{\Sigma}_0)^{-1}/\delta)}$$

we can similarly bound

$$\beta^T/\sigma_{\boldsymbol{w}} \le \sqrt{\lambda}B_A/\sigma_{\boldsymbol{w}} + \sqrt{8d_{\boldsymbol{x}}\log 5 + 8d_\phi \cdot \log(1 + B_\phi^2 TH/\lambda) + 8\log(T/\delta)}$$
$$\le \sqrt{\lambda}B_A/\sigma_{\boldsymbol{w}} + c\sqrt{d_{\boldsymbol{x}} + d_\phi\log(1 + B_\phi TH/\lambda) + \log 1/\delta}.$$

Choosing $\lambda = \sigma_{\boldsymbol{w}}^2$ completes the proof. $\qquad\square$

## E.1 Supporting Lemmas

**Lemma E.1.** *Under Assumption 14, we can bound*

$$\mathcal{J}(\pi;A_\star) - \mathcal{J}(\pi;A) \le c_{\max}\cdot\mathbb{E}_{A_\star,\pi}\left[\sum_{h=1}^{H}\min\left\{\frac{1}{\sigma_{\boldsymbol{w}}}\|(A_\star - A)\phi(\boldsymbol{x}_h,\boldsymbol{u}_h)\|_2,1\right\}\right].$$

*Proof.* Following the proof of Lemma B.3 of [17], and adopting the same notation, we have

$$\mathcal{J}(\pi;A_\star) - \mathcal{J}(\pi;A) \le \sum_{h=1}^{H}\mathbb{E}_{A_\star,\pi}\left[\sqrt{A_h}\min\left\{\frac{1}{\sigma_{\boldsymbol{w}}}\|(A_\star - A)\phi(\boldsymbol{x}_h,\boldsymbol{u}_h)\|_2,1\right\}\right].$$

Under Assumption 14 we have $A_h \le c_{\max}^2$. Plugging this in gives the result. $\qquad\square$

**Lemma E.2.** *Let $\beta^t := \sqrt{\lambda}B_A + \sigma_{\boldsymbol{w}}\sqrt{8d_{\boldsymbol{x}}\log 5 + 8\log(T\det(\boldsymbol{\Sigma}_t)\det(\boldsymbol{\Sigma}_0)^{-1}/\delta)}$ and let $\mathcal{E}_1$ denote the event*

$$\mathcal{E}_1 := \left\{\forall t \le T \ : \ \left\|(\bar{A}^t - A_\star)\boldsymbol{\Sigma}_t^{1/2}\right\|_{\mathrm{op}} \le \beta^t\right\}.$$

*Then running $\mathrm{LC}^3$ we have $\mathbb{P}_{A_\star}[\mathcal{E}_1] \ge 1-\delta$.*

*Proof.* The proof of Lemma B.5 of [17] shows that with probability at least $1-\delta$,

$$\left\|(\bar{A}^t - A_\star)\boldsymbol{\Sigma}_t^{1/2}\right\|_{\mathrm{op}} \le \sqrt{\lambda}\|A_\star\|_{\mathrm{op}} + \sigma_{\boldsymbol{w}}\sqrt{8d_{\boldsymbol{x}}\log 5 + 8\log(\det(\boldsymbol{\Sigma}_t)\det(\boldsymbol{\Sigma}_0)^{-1}/\delta)}.$$

The result then follows from this, since $\|A_\star\|_{\mathrm{op}} \le B_A$, and a union bound. $\qquad\square$

**Lemma E.3.** *With probability* $1 - \delta$*, we have*

$$\sum_{t=1}^{T} \mathbb{E}_{A_\star, \pi^t} \left[ \sum_{h=1}^{H} \min \left\{ \frac{2\beta^t}{\sigma_{\boldsymbol{w}}} \| \boldsymbol{\phi}(\boldsymbol{x}_h, \boldsymbol{u}_h) \|_{\boldsymbol{\Sigma}_t^{-1}}, 1 \right\} \right] \leq \frac{2\beta^T}{\sigma_{\boldsymbol{w}}} \sum_{t=1}^{T} \sum_{h=1}^{H} \min \left\{ \| \boldsymbol{\phi}(\boldsymbol{x}_h^t, \boldsymbol{u}_h^t) \|_{\boldsymbol{\Sigma}_t^{-1}}, 1 \right\}$$
$$+ 4H\sqrt{T \log 1/\delta}.$$

*Proof.* This is an immediate consequence of Azuma-Hoeffding, since $\sum_{h=1}^{H} \min \left\{ \frac{2\beta^t}{\sigma_{\boldsymbol{w}}} \| \boldsymbol{\phi}(\boldsymbol{x}_h, \boldsymbol{u}_h) \|_{\boldsymbol{\Sigma}_t^{-1}}, 1 \right\} \leq H$ almost surely, and from upper bounding

$$\min \left\{ \frac{2\beta^t}{\sigma_{\boldsymbol{w}}} \| \boldsymbol{\phi}(\boldsymbol{x}_h^t, \boldsymbol{u}_h^t) \|_{\boldsymbol{\Sigma}_t^{-1}}, 1 \right\} \leq \frac{2\beta^T}{\sigma_{\boldsymbol{w}}} \min \left\{ \| \boldsymbol{\phi}(\boldsymbol{x}_h^t, \boldsymbol{u}_h^t) \|_{\boldsymbol{\Sigma}_t^{-1}}, 1 \right\}.$$

$\square$

# F    Lower Bounds on Learning in Nonlinear Systems

In this section, we assume that $\boldsymbol{\theta}_\star$ and $\pi_\star$ correspond to the global minimizer:

$$\boldsymbol{\theta}_\star(A) := \arg\min_{\boldsymbol{\theta} \in \mathbb{R}^{d_{\boldsymbol{\theta}}}} \mathcal{J}(\boldsymbol{\theta}; A), \quad \pi_\star(A) := \arg\min_{\pi \in \Pi^\star} \mathcal{J}(\pi; A). \tag{F.1}$$

Here we formally state the additional assumptions needed in Section 4.2, and provide a formal version of Theorem 2.

**Assumption 15.** *There exists some* $r_\mu(A_\star) > 0$ *such that, for all* $A \in \mathcal{B}_{\mathrm{F}}(A_\star, r_\mu(A_\star))$*,* $\pi_\star(A)$ *is unique and, furthermore, there exists some* $\mu > 0$ *such that*

$$\mathcal{J}(\boldsymbol{\theta}; A) \geq \mathcal{J}(\boldsymbol{\theta}_\star(A); A) + \tfrac{\mu}{2} \| \boldsymbol{\theta} - \boldsymbol{\theta}^\star(A) \|_2^2.$$

Assumption 15 will be satisfied in cases where $\mathcal{J}(\pi^{\boldsymbol{\theta}}; A)$ is strongly convex in $\boldsymbol{\theta}$, but may hold even when this is not the case. Intuitively, it requires that our controller class is not overparameterized—moving $\boldsymbol{\theta}$ away from its optimal value will cause the loss to increase. We will additionally make the following regularity assumptions on policies in $\Pi_{\mathrm{exp}}$ and their induced covariates set, $\boldsymbol{\Omega}$.

**Assumption 16.** *There exists some* $\underline{\lambda} > 0$ *such that, for each* $\boldsymbol{\Lambda} \in \boldsymbol{\Omega}$*, we have* $\lambda_{\min}(\boldsymbol{\Lambda}) \geq \underline{\lambda}$*.*

Assumption 16 requires that *every* exploration policy we consider excites all directions in $\boldsymbol{\phi}$ space (in contrast, Assumption 3 only assumes there *exists* some distribution over policies in $\Pi_{\mathrm{exp}}$ which excite all directions). We remark that this assumption is relatively mild if Assumption 3 holds. As we show in Appendix C.2, under Assumption 3, a mixture over policies, $\omega$, satisfying $\lambda_{\min}(\mathbb{E}_{\pi \sim \omega}[\boldsymbol{\Lambda}_\pi]) > 0$ can be learned using only a number of samples scaling polynomially in problem parameters. Given $\omega$, a policy class $\Pi_{\mathrm{exp}}$ satisfying Assumption 16 can be obtained by simply mixing $\omega$ with every other exploration policy. We are now ready to state our main lower bound.

**Theorem 7** (Formal Version of Theorem 2)**.** *Under Assumptions 1, 2, 4, 5, 13, 15 and 16 and if* $\pi_\star$ *is defined as in* (F.1)*, as long as* $T \geq C_{\mathrm{lb}}$*, for any* $\omega_{\mathrm{exp}} \in \triangle_{\Pi_{\mathrm{exp}}}$*, we have*

$$\min_{\widehat{\pi}} \max_{A \in \mathcal{B}_T} \mathbb{E}_{\mathfrak{D}_T \sim A, \omega_{\mathrm{exp}}}[\mathcal{J}(\widehat{\pi}(\mathfrak{D}_T); A) - \mathcal{J}(\pi_\star(A); A)] \geq \frac{\sigma_{\boldsymbol{w}}^2}{3T} \cdot \min_{\boldsymbol{\Lambda} \in \boldsymbol{\Omega}} \mathrm{tr}(\mathcal{H}(A_\star)\check{\boldsymbol{\Lambda}}^{-1}) - \frac{C_{\mathrm{lb}}}{T^{5/4}}$$

*for* $\mathcal{B}_T := \{ A \ : \ \| A - A_\star \|_{\mathrm{F}}^2 \leq 5 d_{\boldsymbol{x}} d_{\boldsymbol{\phi}} / (\underline{\lambda} d_{\boldsymbol{x}} T H)^{5/6} \}$*,* $\mathbb{E}_{\mathfrak{D}_T \sim A, \omega_{\mathrm{exp}}}[\cdot] = \mathbb{E}_{\pi \sim \omega_{\mathrm{exp}}}[\mathbb{E}_{\mathfrak{D}_T \sim A, \pi}[\cdot]]$ *denotes the expectation over trajectories generated by running policies* $\pi$ *drawn according to* $\omega_{\mathrm{exp}}$ *on system* $A$ *for* $T$ *episodes,* $\widehat{\pi}$ *any mapping from observations to policies in* $\Pi^\star$*, and*

$$C_{\mathrm{lb}} := \mathrm{poly}\left( d_{\boldsymbol{\phi}}, d_{\boldsymbol{x}}, H, \| A_\star \|_{\mathrm{op}}, B_{\boldsymbol{\phi}}, L_{\boldsymbol{\phi}}, L_{\boldsymbol{\theta}}, L_{\mathrm{cost}}, L_{\pi_\star}, \sigma_{\boldsymbol{w}}, \sigma_{\boldsymbol{w}}^{-1}, \tfrac{1}{\underline{\lambda}}, \tfrac{1}{\mu}, \tfrac{1}{r_{\mathrm{cost}}(A_\star)}, \tfrac{1}{r_{\boldsymbol{\theta}}(A_\star)}, \tfrac{1}{r_\mu(A_\star)} \right).$$

*Proof of Theorem 7.* This proof follows immediately from Lemma F.1, by lower bounding the right-hand side of (F.2) by the min over all policies in $\Pi$.  $\square$

**Lemma F.1.** *Under Assumptions 1, 2, 4, 5, 13, 15 and 16 and if $\boldsymbol{\theta}_\star$ is defined as in (F.1), as long as*

$$T \geq \mathrm{poly}(\|A_\star\|_{\mathrm{op}}, L_\phi, L_{\boldsymbol{\theta}}, L_{\pi_\star}, L_{\mathrm{cost}}, \sigma_{\boldsymbol{w}}, \sigma_{\boldsymbol{w}}^{-1}, B_\phi, H, d_{\boldsymbol{x}}, d_\phi, \underline{\lambda}^{-1}, \mu^{-1}, r_{\mathrm{cost}}(A_\star)^{-1}, r_{\boldsymbol{\theta}}(A_\star)^{-1}, r_\mu(A_\star)^{-1}),$$

*for any $\omega_{\mathrm{exp}} \in \triangle_\Pi$, we have*

$$\min_{\widehat{\boldsymbol{\theta}}} \max_{A \in \mathcal{B}_T} \mathbb{E}_{\mathfrak{D}_T \sim A, \omega_{\mathrm{exp}}}[\mathcal{J}(\widehat{\boldsymbol{\theta}}(\mathfrak{D}_T); A) - \mathcal{J}(\boldsymbol{\theta}_\star(A); A)] \geq \frac{\sigma_{\boldsymbol{w}}^2}{3T} \cdot \mathrm{tr}(\mathcal{H}(A_\star)\mathbb{E}_{\pi \sim \omega_{\mathrm{exp}}}[\check{\boldsymbol{\Lambda}}_\pi]^{-1}) - \frac{C_{\mathrm{lb}}}{T^{5/4}} \tag{F.2}$$

*for $\mathcal{B}_T := \{A \ : \ \|A - A_\star\|_{\mathrm{F}}^2 \leq 5d_{\boldsymbol{x}}d_\phi/(\underline{\lambda}TH)^{5/6}\}$, where $\mathbb{E}_{\mathfrak{D}_T \sim A, \omega_{\mathrm{exp}}}[\cdot] = \mathbb{E}_{\pi \sim \omega_{\mathrm{exp}}}[\mathbb{E}_{\mathfrak{D}_T \sim A, \pi}[\cdot]]$ denotes the expectation over trajectories generated by running policies $\pi$ drawn according to $\omega_{\mathrm{exp}}$ on system $A$ for $T$ episodes, and*

$$C_{\mathrm{lb}} := \mathrm{poly}(\|A_\star\|_{\mathrm{op}}, L_\phi, L_{\boldsymbol{\theta}}, L_{\pi_\star}, L_{\mathrm{cost}}, \sigma_{\boldsymbol{w}}, \sigma_{\boldsymbol{w}}^{-1}, B_\phi, H, d_{\boldsymbol{x}}, d_\phi, \underline{\lambda}^{-1}, \mu^{-1}, r_{\mathrm{cost}}(A_\star)^{-1}, r_{\boldsymbol{\theta}}(A_\star)^{-1}, r_\mu(Ast)^{-1}).$$

*Proof.* This result is a direct consequence of Theorem 6.1 of [2]—to obtain the result we must only verify that the assumptions of this result are met. We verify each assumption below.

**Verifying Assumption 3 of [2].** Part 1 of Assumption 3 of [2] is met by Assumption 15 within diameter $r_\mu(A_\star)$. Furthermore, under Assumptions 1, 2, 4, 5 and 13 and by Lemma D.1, the additional parts of Assumption 3 of [2] are also met with diameter $\min\{r_{\mathrm{cost}}(A_\star), r_{\boldsymbol{\theta}}(A_\star)\}$ and smoothness constant $\mathrm{poly}(\|A_\star\|_{\mathrm{op}}, L_\phi, L_{\boldsymbol{\theta}}, L_{\pi_\star}, L_{\mathrm{cost}}, \sigma_{\boldsymbol{w}}^{-1}, H, d_{\boldsymbol{x}})$.

**Verifying Assumption 4 and Assumption 5 of [2].** Assumption 4 of [2] is immediately met by Assumption 16. Furthermore, Assumption 5 is met by Lemma F.2 with $c_{\mathrm{cov}} = 1$, $L_{\mathrm{cov}}(\theta_\star, \gamma^2) = \frac{H^2 B_\phi^3}{\sigma_{\boldsymbol{w}}^2} \cdot \mathrm{poly}(d_{\boldsymbol{x}})$, and $C_{\mathrm{cov}} = 0$.

Given that these assumptions are met, the result follows noting that, if we run for $T$ episodes, then the effective horizon is $d_{\boldsymbol{x}}TH$ (using the mapping from the setting of (1.1) to the martingale regression setting described in Appendix A.1.1). Note that the final bound scales with $\frac{1}{T}$ instead of $\frac{1}{d_{\boldsymbol{x}}TH}$ as we are able to bring the $d_{\boldsymbol{x}}H$ factor into the $\check{\boldsymbol{\Lambda}}_{\pi_{\mathrm{exp}}}$ term, since $\boldsymbol{\Lambda}_{\pi_{\mathrm{exp}}}$ is not normalized by $d_{\boldsymbol{x}}H$. $\qquad\square$

**Lemma F.2.** *Under Assumption 1, for any policy distribution $\omega \in \triangle_{\Pi_{\mathrm{exp}}}$ and $A, A'$, we have*

$$\mathbb{E}_{\pi \sim \omega}[\boldsymbol{\Lambda}_{A,\pi}] \preceq \mathbb{E}_{\pi \sim \omega}[\boldsymbol{\Lambda}_{A',\pi}] + \frac{H^2 B_\phi^3}{\sigma_{\boldsymbol{w}}^2} \cdot \mathrm{poly}(d_{\boldsymbol{x}}) \cdot \|A - A'\|_{\mathrm{F}} \cdot I.$$

*Proof.* We will prove that the desired bound follows for a particular $\pi \in \Pi_{\mathrm{exp}}$, which immediately implies that it holds for $\omega \in \triangle_{\Pi_{\mathrm{exp}}}$. By definition we have

$$\boldsymbol{\Lambda}_{A,\pi} = \int \left(\sum_{h=1}^H \boldsymbol{\phi}(\boldsymbol{x}_h^{\boldsymbol{\tau}}, \boldsymbol{u}_h^{\boldsymbol{\tau}})\boldsymbol{\phi}(\boldsymbol{x}_h^{\boldsymbol{\tau}}, \boldsymbol{u}_h^{\boldsymbol{\tau}})^\top\right) \cdot f_{A,\pi}(\boldsymbol{\tau})\mathrm{d}\boldsymbol{\tau}$$

and

$$f_{A,\pi}(\boldsymbol{\tau}) = \prod_{h=1}^H f_A(\boldsymbol{x}_{h+1}^{\boldsymbol{\tau}} \mid \boldsymbol{x}_h^{\boldsymbol{\tau}}, \boldsymbol{u}_h^{\boldsymbol{\tau}})\pi_h(\boldsymbol{u}_h^{\boldsymbol{\tau}} \mid \boldsymbol{x}_h^{\boldsymbol{\tau}}).$$

Fix some $\boldsymbol{v} \in \mathcal{S}^{d_\phi - 1}$, and note that, given the expression above, we have

$$\boldsymbol{v}^\top \boldsymbol{\Lambda}_{A,\pi} \boldsymbol{v} = \int \sum_{h=1}^H (\boldsymbol{v}^\top \boldsymbol{\phi}(\boldsymbol{x}_h^{\boldsymbol{\tau}}, \boldsymbol{u}_h^{\boldsymbol{\tau}}))^2 \cdot f_{A,\pi}(\boldsymbol{\tau})\mathrm{d}\boldsymbol{\tau}.$$

It follows that

$$\nabla_A \boldsymbol{v}^\top \boldsymbol{\Lambda}_{A,\pi} \boldsymbol{v} = \int \sum_{h=1}^H (\boldsymbol{v}^\top \boldsymbol{\phi}(\boldsymbol{x}_h^{\boldsymbol{\tau}}, \boldsymbol{u}_h^{\boldsymbol{\tau}}))^2 \cdot \nabla_A f_{A,\pi}(\boldsymbol{\tau})\mathrm{d}\boldsymbol{\tau}$$

$$= \int \sum_{h=1}^{H} (\boldsymbol{v}^\top \boldsymbol{\phi}(\boldsymbol{x}_h^\boldsymbol{\tau}, \boldsymbol{u}_h^\boldsymbol{\tau}))^2 \cdot f_{A,\pi}(\boldsymbol{\tau}) \nabla_A \log f_{A,\pi}(\boldsymbol{\tau}) \mathrm{d}\boldsymbol{\tau}.$$

As in the proof of Lemma D.1, we have, for any $\Delta$,

$$\nabla_A \log f_{A,\pi}(\boldsymbol{\tau})[\Delta] = \sum_{h=1}^{H} \frac{1}{\sigma_{\boldsymbol{w}}^2} (\boldsymbol{x}_{h+1}^\boldsymbol{\tau} - A\boldsymbol{\phi}(\boldsymbol{x}_h^\boldsymbol{\tau}, \boldsymbol{u}_h^\boldsymbol{\tau})) \cdot \Delta \boldsymbol{\phi}(\boldsymbol{x}_h^\boldsymbol{\tau}, \boldsymbol{u}_h^\boldsymbol{\tau}),$$

so we can bound

$$\|\nabla_A \log f_{A,\pi}(\boldsymbol{\tau})\|_{\mathrm{op}} \le \frac{B_\phi}{\sigma_{\boldsymbol{w}}^2} \sum_{h=1}^{H} \|\boldsymbol{x}_{h+1}^\boldsymbol{\tau} - A\boldsymbol{\phi}(\boldsymbol{x}_h^\boldsymbol{\tau}, \boldsymbol{u}_h^\boldsymbol{\tau})\|_2.$$

Furthermore, we can also bound $(\boldsymbol{v}^\top \boldsymbol{\phi}(\boldsymbol{x}_h^\boldsymbol{\tau}, \boldsymbol{u}_h^\boldsymbol{\tau}))^2 \le B_\phi^2$. We therefore have

$$\|\nabla_A \boldsymbol{v}^\top \boldsymbol{\Lambda}_{A,\pi} \boldsymbol{v}\|_{\mathrm{op}} \le H B_\phi^3 \int \sum_{h=1}^{H} \|\boldsymbol{x}_{h+1}^\boldsymbol{\tau} - A\boldsymbol{\phi}(\boldsymbol{x}_h^\boldsymbol{\tau}, \boldsymbol{u}_h^\boldsymbol{\tau})\|_2 \cdot f_{A,\pi}(\boldsymbol{\tau}) \mathrm{d}\boldsymbol{\tau}$$

$$\le \frac{H^2 B_\phi^3}{\sigma_{\boldsymbol{w}}^2} \cdot \mathrm{poly}(d_{\boldsymbol{x}})$$

where the last inequality follows from Lemma A.1. It follows from the Mean Value Theorem that

$$|\boldsymbol{v}^\top \boldsymbol{\Lambda}_{A,\pi} \boldsymbol{v} - \boldsymbol{v}^\top \boldsymbol{\Lambda}_{A',\pi} \boldsymbol{v}| \le \frac{H^2 B_\phi^3}{\sigma_{\boldsymbol{w}}^2} \cdot \mathrm{poly}(d_{\boldsymbol{x}}) \cdot \|A - A'\|_{\mathrm{F}}.$$

As this holds for all $\boldsymbol{v} \in \mathcal{S}^{d-1}$, it follows that

$$\|\boldsymbol{\Lambda}_{A,\pi} - \boldsymbol{\Lambda}_{A',\pi}\|_{\mathrm{op}} \le \frac{H^2 B_\phi^3}{\sigma_{\boldsymbol{w}}^2} \cdot \mathrm{poly}(d_{\boldsymbol{x}}) \cdot \|A - A'\|_{\mathrm{F}}.$$

$\square$

# G    Additional Experimental Details

In this section, we provide additional details on our experimental results presented in Section 6. All experiments were run on a machine with 56 Intel(R) Xeon(R) CPU E5-2690 v4 @ 2.60GHz CPUs, and 64GB RAM. All code was implemented in `PyTorch`.

## G.1    Details on Problem Settings and Controller Parameterizations

We first expand on the precise definitions of the systems considered. As noted in Section 6, for the drone and car examples we set $H = 50$, and for the system of Section 1.1 we set $H = 10$. In addition, for all examples the noise is distributed as $\boldsymbol{w}_h \sim \mathcal{N}(0, 0.1 \cdot I)$. In all cases we set $\gamma^2 = 10H$ (where $\gamma^2$ is a bound on $\mathbb{E}_{\pi_{\exp}}[\sum_{h=1}^{H} \boldsymbol{u}_h^\top \boldsymbol{u}_h]$), and we therefore let $\Pi_{\exp}$ denote the set of all policies satisfying $\mathbb{E}_{\pi_{\exp}}[\sum_{h=1}^{H} \boldsymbol{u}_h^\top \boldsymbol{u}_h] \le \gamma^2$.

### G.1.1    System of Section 1.1 (Figure 1)

The dynamics for this system are given by

$$\boldsymbol{x}_{h+1} = 0.8\boldsymbol{x}_h + \boldsymbol{u}_h - \sum_{i=1}^{10} 3\boldsymbol{\phi}_i(\boldsymbol{x}_h) + \boldsymbol{w}_h$$

for $\boldsymbol{\phi}_i(\boldsymbol{x}) = \max\{1 - 100(\boldsymbol{x} - c_i)^2, 0\}$, and $\mathrm{cost}(\boldsymbol{x}, \boldsymbol{u}) = (\boldsymbol{x} - c_1)^2 + 100^{-1} \cdot \boldsymbol{u}^2$. We set

$$c_1 = 10, c_2 = -14, c_3 = -11, c_4 = -8, c_5 = -5, c_6 = -2, c_7 = 1, c_8 = 4, c_9 = 7.$$

This then corresponds to a system in the form (1.1) with

$$A_\star = [0.8, 1, -3, \dots, -3], \quad \boldsymbol{\phi}(\boldsymbol{x}, \boldsymbol{u}) = [\boldsymbol{x}, \boldsymbol{u}, \boldsymbol{\phi}_1(\boldsymbol{x}), \dots, \boldsymbol{\phi}_{10}(\boldsymbol{x})].$$

For this system, we parameterize our controller class $\Pi^\star$ as, for any $\pi^{\boldsymbol{\theta}} \in \Pi^\star$ with parameter $\boldsymbol{\theta}$,

$$\pi^{\boldsymbol{\theta}}(\boldsymbol{x}) = \boldsymbol{\theta}_1 \boldsymbol{x} + \sum_{i=1}^{10} \boldsymbol{\theta}_{i+1} \phi_i(\boldsymbol{x}) + \boldsymbol{\theta}_{12}.$$

Note that the form of this controller lets us simply "match" the parameters of the system, and cancel undesirable parameters. Given this, for this system we let $\pi_\star(A)$ be the controller which sets $\boldsymbol{\theta}_{1:11}$ to cancel the dynamics of the system $A$, and set $\boldsymbol{\theta}_{12} = c_1 = 10$.

See Appendix G.1.3 for details on the computation of $\mathcal{H}(A)$ on this system.

### G.1.2 Drone System (Figure 2)

The dynamics of this system are given by

$$\boldsymbol{x}_{h+1} = \begin{bmatrix} 1 & 0 & 0 & 0.1 & 0 & 0 \\ 0 & 1 & 0 & 0 & 0.1 & 0 \\ 0 & 0 & 1 & 0 & 0 & 0.1 \\ 0 & 0 & 0 & 1 & 0 & 0 \\ 0 & 0 & 0 & 0 & 1 & 0 \\ 0 & 0 & 0 & 0 & 0 & 1 \end{bmatrix} \boldsymbol{x}_h + \begin{bmatrix} 0 & 0 & 0 \\ 0 & 0 & 0 \\ 0 & 0 & 0 \\ 0.1 & 0 & 0 \\ 0 & 0.1 & 0 \\ 0 & 0 & 0.1 \end{bmatrix} \boldsymbol{u}_h + \begin{bmatrix} 0 \\ 0 \\ 0 \\ 0 \\ 0 \\ -0.98 \end{bmatrix} + \boldsymbol{w}_h. \qquad \text{(G.1)}$$

Here we interpret $[\boldsymbol{x}]_{1:3}$ as the $x, y$, and $z$ positions, respectively, and $[\boldsymbol{x}]_{4:6}$ as the $x, y, z$ velocities. This system is therefore equivalent to three double integrator systems, with an affine term (which we interpret as "gravity") affecting only the $z$ coordinate. We set the cost to

$$\text{cost}(\boldsymbol{x}, \boldsymbol{u}) = \frac{0.1}{5} \cdot \sum_{i=1}^{d_{\boldsymbol{x}}} [\boldsymbol{x}]_i^2 + \frac{1}{5} \cdot [\boldsymbol{u}]_1^2 + [\boldsymbol{u}]_2^2 + [\boldsymbol{u}]_3^2$$

This then corresponds to a system in the form (1.1) with

$$A_\star = \begin{bmatrix} 1 & 0 & 0 & 0.1 & 0 & 0 & 0 & 0 & 0 & 0 \\ 0 & 1 & 0 & 0 & 0.1 & 0 & 0 & 0 & 0 & 0 \\ 0 & 0 & 1 & 0 & 0 & 0.1 & 0 & 0 & 0 & 0 \\ 0 & 0 & 0 & 1 & 0 & 0 & 0.1 & 0 & 0 & 0 \\ 0 & 0 & 0 & 0 & 1 & 0 & 0 & 0.1 & 0 & 0 \\ 0 & 0 & 0 & 0 & 0 & 1 & 0 & 0 & 0.1 & -0.98 \end{bmatrix}, \quad \phi(\boldsymbol{x}, \boldsymbol{u}) = [\boldsymbol{x}, \boldsymbol{u}, 1].$$

For this system, we parameterize our controller class $\Pi^\star$ as, for any $\pi^{\boldsymbol{\theta}} \in \Pi^\star$ with parameter $\boldsymbol{\theta}$,

$$\pi_h^{\boldsymbol{\theta}}(\boldsymbol{x}) = \boldsymbol{\theta}_h^{\text{fb}} \boldsymbol{x} + \boldsymbol{\theta}_h^{\text{offset}}$$

where $\boldsymbol{\theta}_h^{\text{fb}} \in \mathbb{R}^{3 \times 6}$ is the state-feedback portion of the controller, and $\boldsymbol{\theta}_h^{\text{offset}} \in \mathbb{R}^3$ is an offset term. It can be shown that the optimal controller for a system of the form (G.1) can be parameterized in this way [23]. Furthermore, the optimal parameters can be computed in closed-form. As such, for this system we set $\pi_\star(A)$ to be with the optimal parameters, computed using this closed-form solution.

In addition to computing the optimal controller in closed-form, we can also compute the cost of a controller, $\mathcal{J}(\pi; A)$, in closed-form. To compute $\mathcal{H}(A)$ in this example, we then simply apply the `torch.autograd.functional.hessian` function to $\mathcal{J}(\pi_\star(A); A)$.

### G.1.3 Car System (Figure 3)

The dynamics of this system are given by

$$\boldsymbol{x}_{h+1} = \begin{bmatrix} 1 & 0 & 0.1 & 0 & 0 & 0 \\ 0 & 1 & 0 & 0.1 & 0 & 0 \\ 0 & 0 & 1 & 0 & 0 & 0 \\ 0 & 0 & 0 & 1 & 0 & 0 \\ 0 & 0 & 0 & 0 & 1 & 0.1 \\ 0 & 0 & 0 & 0 & 0 & 1 \end{bmatrix} \boldsymbol{x}_h + \begin{bmatrix} 0 & 0 \\ 0 & 0 \\ 0.1 \cdot \cos([\boldsymbol{x}_h]_5) & 0 \\ 0.1 \cdot \sin([\boldsymbol{x}_h]_5) & 0 \\ 0 & 0 \\ 0 & 0.1 \end{bmatrix} \boldsymbol{u}_h + \boldsymbol{w}_h \qquad \text{(G.2)}$$

where $[\boldsymbol{x}_h]_5$ denotes the 5th element of $\boldsymbol{x}_h$. Here we interpret $[\boldsymbol{x}_h]_1$ as the $x$ position, $[\boldsymbol{x}_h]_2$ as the $y$ position, $[\boldsymbol{x}_h]_3$ as the $x$ velocity, $[\boldsymbol{x}_h]_4$ as the $y$ velocity, $[\boldsymbol{x}_h]_5$ as the angle of orientation (that is, the direction the car is facing), and $[\boldsymbol{x}_h]_6$ as the angular velocity. The first control dimension, then, corresponds to the "gas", the power given to the car to move forward or backward, and the second control dimension corresponds to altering the direction of the steering wheel. Similar to the drone system, we set the cost to

$$\text{cost}(\boldsymbol{x}, \boldsymbol{u}) = \begin{bmatrix} \boldsymbol{x} \\ \boldsymbol{u} \end{bmatrix}^\top Q \begin{bmatrix} \boldsymbol{x} \\ \boldsymbol{u} \end{bmatrix} \quad \text{with} \quad Q = \frac{0.1 \cdot I + \boldsymbol{v}_1 \boldsymbol{v}_1^\top + \boldsymbol{v}_2 \boldsymbol{v}_2^\top}{\|0.1 \cdot I + \boldsymbol{v}_1 \boldsymbol{v}_1^\top + \boldsymbol{v}_2 \boldsymbol{v}_2^\top\|_{\text{op}}}$$

for some $\boldsymbol{v}_1, \boldsymbol{v}_2$. To write this in the form of (1.1), in order to make the problem more challenging we choose an overparameterized $\boldsymbol{\phi}(\boldsymbol{x}, \boldsymbol{u})$:

$$\boldsymbol{\phi}(\boldsymbol{x}, \boldsymbol{u}) = \begin{bmatrix} \boldsymbol{x}, \boldsymbol{u}, \cos([\boldsymbol{x}]_5), \sin([\boldsymbol{x}]_5), [\boldsymbol{u}]_1 \cdot \cos([\boldsymbol{x}]_5), [\boldsymbol{u}]_1 \cdot \sin([\boldsymbol{x}]_5), [\boldsymbol{u}]_2 \cdot \cos([\boldsymbol{x}]_5), [\boldsymbol{u}]_2 \cdot \sin([\boldsymbol{x}]_5) \end{bmatrix}$$

and set

$$A_\star = \begin{bmatrix} 1 & 0 & 0.1 & 0 & 0 & 0 & 0 & 0 & 0 & 0 & 0 & 0 & 0 & 0 \\ 0 & 1 & 0 & 0.1 & 0 & 0 & 0 & 0 & 0 & 0 & 0 & 0 & 0 & 0 \\ 0 & 0 & 1 & 0 & 0 & 0 & 0 & 0 & 0 & 0 & 0.1 & 0 & 0 & 0 \\ 0 & 0 & 0 & 1 & 0 & 0 & 0 & 0 & 0 & 0 & 0 & 0.1 & 0 & 0 \\ 0 & 0 & 0 & 0 & 1 & 0.1 & 0 & 0 & 0 & 0 & 0 & 0 & 0 & 0 \\ 0 & 0 & 0 & 0 & 0 & 1 & 0 & 0.1 & 0 & 0 & 0 & 0 & 0 & 0 \end{bmatrix}.$$

For the car system, the controller class $\Pi^\star$ is a hierarchical controller parameterized by some $\boldsymbol{\theta} \in \mathbb{R}^4$. This controller first uses PD control to compute a "goal input", the direction we would like to modify the state in, as:

$$\boldsymbol{u}_{\text{goal}}(\boldsymbol{x}) = -\boldsymbol{\theta}_1 [\boldsymbol{x}]_{1:2} - \boldsymbol{\theta}_2 [\boldsymbol{x}]_{3:4}.$$

Given the underactuated structure of the system in (G.2), we cannot directly push the state in the direction of $\boldsymbol{u}_{\text{goal}}(\boldsymbol{x})$. Instead, we set $\boldsymbol{u}$ to the following:

$$\begin{bmatrix} \boldsymbol{u}_{\text{goal}}(\boldsymbol{x})^\top \begin{bmatrix} \cos([\boldsymbol{x}]_5) \\ \sin([\boldsymbol{x}]_5) \end{bmatrix} \\ -\boldsymbol{\theta}_3([\boldsymbol{x}]_5 - \beta_{\text{goal}}(\boldsymbol{x})) - \boldsymbol{\theta}_4 [\boldsymbol{x}]_6 \end{bmatrix} \quad \text{for} \quad \beta_{\text{goal}}(\boldsymbol{x}) = \tan^{-1}([\boldsymbol{u}_{\text{goal}}(\boldsymbol{x})]_2 / [\boldsymbol{u}_{\text{goal}}(\boldsymbol{x})]_1).$$

Given the complex form of this controller and the dynamics, there does not exist a closed-form way to set $\boldsymbol{\theta}$ optimally. Instead, for this system, we rely on a simple random search procedure to compute $\pi_\star(A)$. To find an optimal controller for system $A$, we randomly sample parameters $\boldsymbol{\theta}$, compute the cost they incur on system $A$, and then set $\pi_\star(A)$ to the randomly generated controller with lowest cost. Note that this procedure is not differentiable, but we require $\pi_\star(A)$ is differentiable. To remedy this, in situations where a differentiable $\pi_\star(A)$ is needed (in particular, in the computation of $\mathcal{H}(A)$), rather than returning a single controller, we return the softmin distribution over all controllers sampled, weighting each controller by its estimated cost. As the softmin distribution can be differentiated, this parameterization of $\pi_\star(A)$ is differentiable.

For this system, there does not exist a closed-form expression for $\mathcal{J}(\pi; A)$ and, as such, to compute $\mathcal{J}(\pi; A)$, we simply perform many roll-outs of policy $\pi$ on system $A$ and average the cost. Given this and the search-based implementation of $\pi_\star(A)$ outlined above, we found that computing the hessian $\mathcal{H}(A)$ using the `torch.autograd.functional.hessian` as in Appendix G.1.2 was very memory-intensive. Instead, we computed the Jacobian $G(A) := \nabla_{A'} \mathcal{J}(\pi_\star(A'); A)|_{A'=A}$, and then, in place of $\mathcal{H}(A)$, we use $G(A)G(A)^\top$. To compute $G(A)$, we use the `torch.autograd.functional.jacobian` function. While using $G(A)G(A)^\top$ in place of $\mathcal{H}(A)$ is not justified by our theoretical analysis, if we are in settings where $\pi_\star(A)$ is not precisely the minimum of $\mathcal{J}(\pi; A)$ (which will likely be the case here since we are relying on a sampling-based implementation of $\pi_\star(A)$, which will incur some small error), then we argue that this is a reasonable metric to use. In particular, in this setting, the approximation of $\mathcal{J}(\pi_\star(\widehat{A}); A_\star)$ given in Proposition 1 should have an additional first-order term of the form $G(A)^\top \text{vec}(A_\star - \widehat{A})$. As we can upper bound

$$G(A)^\top \text{vec}(A_\star - \widehat{A}) \leq \sqrt{\|\text{vec}(A_\star - \widehat{A})\|_{G(A_\star)G(A_\star)^\top}^2},$$

optimizing for the metric $G(A)G(A)^\top$ instead of $\mathcal{H}(A)$ can be seen as minimizing the first-order Taylor-approximation of the excess loss. Intuitively, this metric quantifies the sensitivity of the loss to particular parameters in $A_\star$, and in practice we found that optimizing this metric produced significant improvements over existing methods. The implementation of the example from Section 1.1 relied on this same approximation.

## G.2 Implementation Details

For all methods considered, our implementation follows the basic structure of Algorithm 1: at every epoch, we explore so as to minimize some exploration objective, form an estimate of $A_\star$ on the collected data, and then compute $\pi_\star(\widehat{A}_t)$ on our estimate. Our main experimental results (Figures 1 to 3) show the loss of $\pi_\star(\widehat{A}_t)$ as the time horizon $t$ increases. For each method, to collect an initial set of data, we begin each trial by exploring randomly for some fixed number of episodes (10 for the drone example, 100 for the car example). The first point in each plot then corresponds to the performance after this initial random exploration. Each aspect of our implementation is modular, and any given component can be easily replaced. Below we highlight our implementation of the exploration routine, and choice of exploration objective, for the various approaches we consider.

### G.2.1 Implementation of DYNAMICOED

Implementing the exploration procedure, DYNAMICOED, requires access to a regret minimization oracle. While in principle the $LC^3$ algorithm of [17] could be applied to this problem to give such an oracle, the $LC^3$ algorithm requires access to a computation oracle which is not clear how to implement in practice. To remedy this, we implement a Thompson Sampling-inspired modification to the $LC^3$ algorithm of [17].

The primary computational challenge of implementing the $LC^3$ algorithm is the computation of the *optimistic* policy:

$$\underset{\pi \in \Pi_{\mathrm{exp}}}{\arg\min} \ \underset{A \in \mathrm{BALL}^t}{\min} \ \mathcal{J}^{\mathrm{exp}}(\pi; A)$$

where $\mathcal{J}^{\mathrm{exp}}(\pi; A)$ denotes the exploration cost that is minimized in $LC^3$ (i.e. the expected cost on the cost function $\mathrm{cost}_h^n(\boldsymbol{x}, \boldsymbol{u}) \leftarrow \frac{1}{M} \cdot \boldsymbol{\phi}(\boldsymbol{x}, \boldsymbol{u})^\top (\Xi_n) \boldsymbol{\phi}(\boldsymbol{x}, \boldsymbol{u})$ set in DYNAMICOED), and $\mathrm{BALL}^t$ the confidence set for $A_\star$ at iteration $t$.

To avoid solving this optimization, we adopt a Thompson Sampling-inspired variation of this procedure. In particular, at iteration $t$, we sample $\widetilde{A}_t \sim \mathcal{N}(\widehat{A}_t, \boldsymbol{\Lambda}_t^{-1})$. Standard Thompson Sampling would then compute $\arg\min_{\pi \in \Pi_{\mathrm{exp}}} \mathcal{J}^{\mathrm{exp}}(\pi; \widetilde{A}_t)$, but even this can be challenging, so we instead rely on a sampling MPC-inspired approach. Given that we are at state $\boldsymbol{x}_h$ and have played inputs $\boldsymbol{u}_1, \ldots, \boldsymbol{u}_{h-1}$, we aim to approximately solve the following optimization:

$$\min_{\boldsymbol{u}_h, \boldsymbol{u}_{h+1}, \ldots, \boldsymbol{u}_H \in \mathbb{R}^{d_{\boldsymbol{u}}}} \sum_{h'=h}^{H} \mathrm{cost}_h^n(\boldsymbol{x}_{h'}, \boldsymbol{u}_{h'})$$

$$\text{s.t.} \quad \boldsymbol{x}_{h+1} = \widetilde{A}_t \boldsymbol{\phi}(\boldsymbol{x}_h, \boldsymbol{u}_h), \sum_{h=1}^{H} \boldsymbol{u}_h^\top \boldsymbol{u}_h \leq \gamma^2. \tag{G.3}$$

To solve this approximately, we sample many possible $\boldsymbol{u}$ randomly, compute the value of the objective of (G.3) on the trajectories induced by these $\boldsymbol{u}$, and finally choose the input that minimizes this objective. Rather than playing the entire sequence of chosen inputs, however, we simply play the first input in the sequence, observe the new state on the actual system, and re-solve (G.3) on this new state. Note that the implementation of $LC^3$ used for the experiments given in [17] relies on a similar Thompson Sampling-based approximation to the $LC^3$ algorithm.

### G.2.2 Implementation of Uniform Exploration

The goal of the procedure we have referred to as `Uniform Exploration` is to collect data which will result in the estimation error, $\|A_\star - \widehat{A}\|_{\mathrm{op}}$, being minimized, the goal of the method given in [1]. It can be shown that this is equivalent to maximizing $\lambda_{\min}(\boldsymbol{\Lambda}_T)$, so this method reduces to choosing

inputs that maximize $\lambda_{\min}(\boldsymbol{\Lambda}_T)$. To implement this procedure, we rely on the same sampling-based MPC approach as we outlined above, with the primary difference being that instead of minimizing the objective of (G.3), we choose the inputs that maximize

$$\lambda_{\min}\left(\boldsymbol{\Lambda}_t + \sum_{h=1}^{H} \boldsymbol{\phi}(\boldsymbol{x}_h, \boldsymbol{u}_h)\boldsymbol{\phi}(\boldsymbol{x}_h, \boldsymbol{u}_h)^{\top}\right),$$

where $\boldsymbol{\Lambda}_t$ denote the covariates we have obtained so far at iteration $t$. While very similar in spirit to the algorithm of [1], the implementation details are somewhat different than the algorithm proposed in that work. We found that in practice our implementation performed better than directly implementing (a sampling-based variant of) the algorithm from [1], and all reported results for `Uniform Exploration` are therefore on this version.

### G.2.3    Exploring via Cost Minimization

A natural point of comparison to our methods would be to forsake the system identification phase entirely, and simply run standard policy optimization algorithms such as TRPO or PPO [61, 62], to obtain a controller $\widehat{\pi}_T$. The primary difficulty with these approaches in the settings we consider is that these algorithms are *on-policy*, meaning that they primarily roll out trajectories using their current estimate of the optimal policy, $\widehat{\pi}_t$, and using the collected data to do policy improvement on $\widehat{\pi}_t$. In contrast, our setting is *off-policy*, in the sense that the learner must explore by playing policies in $\Pi_{\exp}$, but return some policy in $\Pi^{\star}$. Since in the settings we consider $\Pi_{\exp} \neq \Pi^{\star}$, on-policy approaches are simply exploring very differently, and therefore cannot be compared with directly.

This is particularly an issue in our setting where *stability* may come into play. Indeed, it may be the case that some controller in $\Pi^{\star}$ will destabilize the system, and cause the norm of the state to increase exponentially in $h$. Inducing such trajectories significantly improves one's ability to perform system identification as the signal-to-noise ratio also then increases exponentially. However, to induce this trajectory with a state-feedback controller, the power of the input played by this controller will also increase exponentially in $h$. Since we choose $\Pi_{\exp}$ to include only policies with bounded power, this is not a fair comparison (and, furthermore, is likely not an algorithm one would want to run in practice).

While direct comparison with such approaches is therefore not possible, it is possible to compare against algorithms that, instead of collecting data that minimizes or maximizes objectives such as $\mathrm{tr}(\mathcal{H}\check{\boldsymbol{\Lambda}}^{-1})$ or $\lambda_{\min}(\boldsymbol{\Lambda})$, instead simply aims to play policies minimizing $\mathcal{J}(\pi; A)$. In principle, such algorithms are similar to approaches such as TRPO in how they perform their exploration—both collect data by aiming to minimize the actual cost we are attempting to find a controller to minimize.

To implement this approach, we rely on a sampling-based MPC algorithm similar to that described in Appendix G.2.1, but where the goal is now to solve

$$\min_{\pi \in \Pi_{\exp}} \mathcal{J}(\pi; A).$$

Note that the key difference between this approach and approaches such as TRPO is that we still only play $\pi \in \Pi_{\exp}$. Using this objective to induce exploration, we then simply estimate $A_{\star}$ on this collected data, and return $\pi_{\star}(\widehat{A})$. The results of this approach on the drone system are given in Figure 4 (with this cost minimization approach denoted as `Cost Minimization Exploration`). As this illustrates, this approach is significantly worse than Algorithm 1, and is also outperformed by `Uniform Exploration` or `Random Exploration` when the number of episodes is large enough.

### G.3    Additional Results

Finally, in this section we present versions of Figures 1 to 3 with error bars in Figures 5 to 7. In all figures, errors bars denote one standard error.

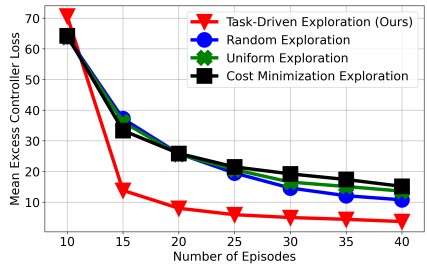

Figure 4: Performance on drone with LC$^3$ Exploration

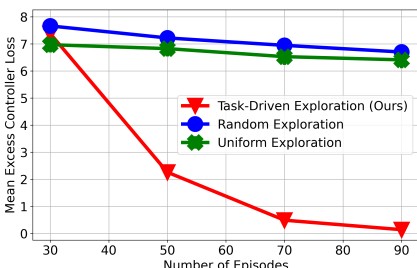

Figure 5: Mean excess controller loss on instance of Section 1.1 with error bars

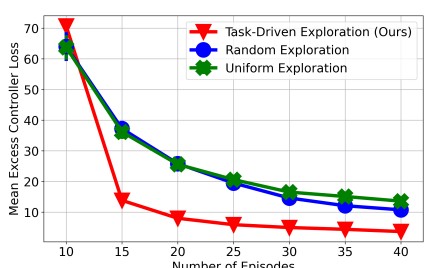

Figure 6: Mean excess controller loss on drone with error bars

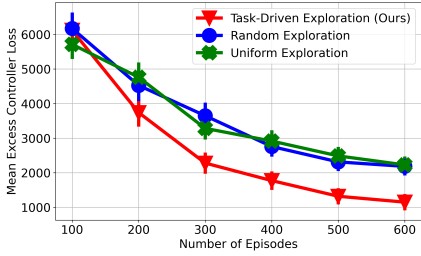

Figure 7: Mean excess controller loss on car with error bars

