# OpenReview forum: "Optimal Exploration for Model-Based RL in Nonlinear Systems"
_NeurIPS.cc/2023/Conference — NeurIPS 2023 spotlight_

### Official Review · Reviewer_vdw9 · 2023-06-14

**Soundness:** 3 good
**Presentation:** 3 good
**Contribution:** 3 good
**Rating:** 6
**Confidence:** 2

**Summary:**

This paper addresses the problem of exploration when learning to control non linear systems. In particular, they derive a task-driven exploration method which focuses on learning the system parameters that are relevant for the specific task that a controller is trying to achieve.

To derive the algorithm, they rely on a quantity which estimates how much the task loss vary, under a certainty-equivalence controller and the current model estimate. This quantity is the key to estimating what parameters in the model are relevant for the task at hand. It can be used by existing exploration algorithms (DynamicOED) as a minimization target.

The exploration routine is presented as a second contribution which extends an algorithm originally designed for linear MDPs to non-linear systems.


**Strengths:**

The premises of the article that all parameters of the system might not be useful to know for the task of the controller makes a lot of sense, and the problem of task driven exploration is interesting and relevant.

Overall, there is novelty as they extend a lot of existing concepts to a much broader class of systems (non-linear) and the contribution seems significant. However, it would be quite hard to implement the proposed algorithm from reading just the paper itself.

The introduction of the motivating example and associated discussion is quite useful to understand the problem space.

The related work section helps to position the scope of the paper and is mixing both RL and more traditional control.

The experiment section show that the proposed algorithm outperforms non task-driven exploration methods. The set of baselines is quite limited but as the concept is quite specific to this article, I believe they are a sound choice.


**Weaknesses:**

Although the paper is well structured and written, the main weakness is the presentation. Since the paper introduces a lot of mathematical concepts and assumptions, it is useful to help the reader by describing intuitively what they mean.

The assumptions presented are justified by citing previous work but some description would be useful to make the paper more self contained. Specifically assumption 3, it would be useful if the authors provide an intuitive justification.

Assumption 8 lacks some general explanation. How does the choice of the underlying algorithm affect the performance? and why is the cost different than the goal $\Phi$?

A lot of the concepts are derived from reference [2] which essentially solves the problem in the case of linear systems. It makes the contribution more incremental, however the jump from linear to non-linear is still significant.

Theorem 1 and 2 introduce bounds on the exploration loss which are claimed to be novel, it would still be useful to compare with the bounds for algorithms that are not task dependent. Is there a trade-off?
Since section 5 is presented as a potentially independent contribution, it would have been useful to add more background about the experiment design problem and clearly highlight the difference with previous work.

The notation in dynamicOED is hard to follow, the explanation helps only a little. It would be quite hard to implement the algorithm just from reading the paper. It refers a lot to the LC3 algorithm which is not explained (this is certainly not my main expertise).

The systems use in the experiments have fairly small scales (6 dimensions for the state). One could wonder whether the approach scale in the number of dimensions. It is mention a linear dependency with the system parameter for the convergence. What about the computational cost of the algorithm?

In the experimental section, it is mentioned that MPC sampling based methods are used for exploration. How do we know that they verify assumption 8 which is key for convergence?

Maybe the author could have tried linearizing the systems and applied the linear version of the algorithm that is cited.


**Questions:**

- Is there any physical system that would be like the motivating example?

- What is D-optimal design? What to the optimization targets presented as example represent at line 270 in section 5?

- Why is dynamicOED returning several exploration policies and not just one per step in algorithm 1?

- How is the optimal controller $\pi_*$ obtained to compute the excess controller loss?



**Limitations:**

They assume a specific form of non-linear system which could limit the applicability of the solution (at least the theoretical results), however a few references are mentioned indicating that many systems fit into this category.

The bounds scale polynomially in the number of system parameters, it is not commented whether it is desirable or not. Couldn’t systems have a large number of parameters?

---

> ### Author Rebuttal · Authors · 2023-08-09
>
> We would like to thank the reviewer for their comments and will work to incorporate the feedback into the final version. We address specific comments below.
>
> > *“However, it would be quite hard to implement the proposed algorithm...” “The notation in dynamicOED is hard to follow, the explanation helps only a little...”*
>
> While the algorithm description in the main body is somewhat brief due to space constraints, we provided many more experimental details in Appendix G, which we believe will be helpful in implementing our algorithm. Furthermore, we plan to release code for our approach, and will work to clarify notation in the final version.
>
> > *“Although the paper is well structured and written, the main weakness is the presentation...”*
>
> We were somewhat limited by space constraints in the original submission, but will plan to add more explanation as space permits in the final version. We would be happy to provide explanations to any particular points the reviewer has.
>
> > *“The assumptions presented are justified by citing previous work but some description would be useful...”*
>
> Intuitively, Assumption 3 states that every feature direction can be reached, and is relevant to learning a good controller. If this assumption is not satisfied, then $\phi$ is overspecified, and could be projected down to a lower-dimensional space without changing the behavior. If our smoothness assumptions (Assumptions 4 and 5) are not met, small changes in the system parameters could lead to arbitrarily large changes in the cost of the learned controller. To ensure that a (reasonably) good controller can be learned on an estimate of the system parameters, we must assume that the cost of a controller varies smoothly as the parameters of the system change. We will add these intuitions in the extra page.
>
> > *“Assumption 8 lacks some general explanation...”*
>
> As shown in Theorem 3, the parameters the algorithm employed by Assumption 8 has ($C_{\mathcal{R}},p_{\mathcal{R}}$) affect the convergence rate of our experiment design procedure—besides this, the choice of algorithm does not affect the performance. The cost in Assumption 8 is different from the goal cost as we do not apply the algorithm of Assumption 8 to optimize the goal cost directly—instead, we apply it to a cost which incentives exploration (and the data collected during exploration is then used to solve the actual goal). This is illustrated and described in in Section 5.1.
>
> > *“A lot of the concepts are derived from reference [2]...”*
>
> We emphasize that moving from linear to nonlinear systems is usually very non-trivial, which is the case here, and to achieve our guarantee in the nonlinear setting requires non-trivial analysis. For example, in linear systems, random excitation is sufficient, while in nonlinear systems random excitation is not guaranteed to excite each direction, and estimates of system parameters could be arbitrarily bad. To address this required developing exploration techniques that navigate within the system to ensure all directions are excited.
>
> > *“Theorem 1 and 2 introduce bounds on the exploration loss which are claimed to be novel...”*
>
> Only several works that we are aware of can be directly compared with our results in Theorem 1 and 2. In particular, [1,2] both provide guarantees in this setting, but their rates translate to suboptimality guarantees that scale as $O(1/\sqrt{T})$, significantly slower than our $O(1/T)$ rate. The work of [3] does not give an end-to-end guarantee on controller loss (and furthermore they consider the non-episodic setting). However, applying their estimation guarantee in Proposition 1, we obtain a bound scaling as roughly
> $O( || \mathcal{H}(A_\star) ||\_{op} \cdot \min [ d_x, d\_{\phi}] \cdot \frac{H (d_x + d_\phi)}{\lambda_{\min}^\star T})$, which could be significantly worse than our bound, as we  demonstrate to be the case experimentally in Section 1.1 (we emphasize again that these are not directly comparable though—to obtain a direct comparison the results of [3] would need to be reworked in the episodic setting we consider). We will add further discussion on this in the final version.
>
> [1] Kakade, Sham, et al. "Information theoretic regret bounds for online nonlinear control." Advances in Neural Information Processing Systems 33 (2020): 15312-15325.
>
> [2] Song, Yuda, and Wen Sun. "Pc-mlp: Model-based reinforcement learning with policy cover guided exploration." International Conference on Machine Learning. PMLR, 2021.
>
> [3] Mania, Horia, Michael I. Jordan, and Benjamin Recht. "Active learning for nonlinear system identification with guarantees." arXiv preprint arXiv:2006.10277 (2020).
>
> > *“The systems use in the experiments have fairly small scales (6 dimensions for the state)...”*
>
> We refer the reviewer to point 1., 3., and 4. of our response to all reviewers, and to Appendix G for additional experimental details. As noted, we believe our contribution is primarily theoretical, but we believe our experimental results do provide empirical evidence that task-driven exploration yields non-trivial gains on realistic systems, and motivates future experimental work in this direction. Furthermore, efficient approximations of our algorithm exist, which we believe could be scaled to higher-dimensional systems without issue.
>
> > *“In the experimental section, it is mentioned that MPC sampling based methods are used for exploration. How do we know that they verify assumption 8 which is key for convergence?”*
>
> In practice our MPC procedure may not have a guarantee satisfying Assumption 8 directly. However, we see this as an advantage: our experimental results demonstrate that our approach can be effectively applied even when our assumptions are not necessarily met.

---

> > ### Comment · Reviewer_vdw9 · 2023-08-10
> > **Thank you for the clarifications, please add the explanation about the assumptions.**
> >
> > I acknowledge reading the rebuttal of the authors and thank them for answering all my questions clearly.
> >
> > I recommend that you add the explanation about assumption 3 and assumption 8 to the paper to improve the presentation.
> >
> > The discussion of theorem 1 and 2 and their relations to related work could also be added.

---

### Official Review · Reviewer_DNZf · 2023-07-05

**Soundness:** 3 good
**Presentation:** 3 good
**Contribution:** 3 good
**Rating:** 6
**Confidence:** 3

**Summary:**

This paper considers the problem of task-relevant system identification. It starts with the motivation that not all parameters in the dynamics model are relevant to the given task; thus, the exploration should be placed to prioritize the identification of the relevant dynamics parameters, and thus eventually leading to improved efficiency of the model-based policy learning.

The authors begin with translating the task loss to the dynamics prediction loss, which is weighted by the cost hessian at ground truth dynamics model. The main result of this paper is to show that this dynamics loss can be approximated tightly by DynamicOED method. The proposed method is validated using some toy problems.

Overall, the paper is well-motivated and presented. This paper did a great job in explaining the theorems. Although the interest of parametric dynamics model is restricted to weighted features, the main results of this paper may be of interest to the reinforcement learning community.


**Strengths:**

See my comments in Summary

**Weaknesses:**

- In the motivating example, the explanation of task-irrelevant or relevant parameters are confusing. Shouldn't $a_{3:12}$ be irrelevant to learning the optimal controller, while $a_{1:2}$ critical?
- Please further explain "In practice, though they may not formally satisfy 290 the guarantee of Assumption 8, deep RL approaches could be used."
- I think Section 5 a bit deviates from the main problem formulation, especially starting with a new system Equ. (5), although I understand the authors are trying to introduce DynamicOED generally for a broad interest of the readers.
- The authors could give some perspectives on the promise of the results applied to more general dynamics.
- I strongly suggest adding more or complex tasks to demonstrate the efficiency of the proposed methods over non-optimal exploration model-based policy learning.


**Questions:**

See my comments in Weaknesses.

**Limitations:**

See my comments in Weaknesses.

---

> ### Author Rebuttal · Authors · 2023-08-09
>
> We would like to thank the reviewer for their comments and will work to incorporate the feedback into the final version. We address specific comments below.
>
> > *“In the motivating example, the explanation of task-irrelevant or relevant parameters are confusing. Shouldn't $a_{3:12}$ be irrelevant to learning the optimal controller, while $a_{1:2}$ critical?”*
>
> $a_1$, $a_2$, and $a_3$ are all critical to learning the optimal controller—$a_1$ and $a_2$ since they are active everywhere, and $a_3$ since it is active at the point the optimal controller seeks to reach (and so therefore significantly influences the dynamics at the optimal point). $a_1$ and $a_2$ would be learned efficiently be essentially any approach, however—our approach is able to outperform existing approaches by targeting its exploration $a_3$ (while simultaneously learning $a_1$ and $a_2$). We will add additional explanation to the final version to make this clear.
>
> > *“Please further explain "In practice, though they may not formally satisfy 290 the guarantee of Assumption 8, deep RL approaches could be used."”*
>
> Our approach is very modular—for example, the policy parameterization and optimizer used, as well as the regret minimization oracle for DynamicOED could be replaced without changing our end-to-end guarantee. One could therefore instantiate these with a variety of methods, including deep RL approaches (e.g. PPO with neural network function approximation could be run on the estimated system to obtain a good policy). While these approaches may lack the theoretical guarantees necessary for our formal results to hold, we expect that, if they are used effectively, they could still yield strong empirical results.
>
> > *“I think Section 5 a bit deviates from the main problem formulation, especially starting with a new system Equ. (5), although I understand the authors are trying to introduce DynamicOED generally for a broad interest of the readers.”*
>
> As we hoped to make the results of Section 5 as applicable as possible to a wide range of systems (and make clear that the results in this section do not require that our system be of the form (1.1)), we chose to make this section as general as possible. We will make clear in the final version, however, its connections to systems of the form (1.1).
>
> > *“The authors could give some perspectives on the promise of the results applied to more general dynamics.”*
>
> We refer the reviewer to point 2. of our response to all reviewers—the dynamics considered are quite general and can encompass nearly any continuous dynamical system. We believe extending our results to even more general settings is an interesting future direction, but is beyond the scope of this work. In particular, extending to more general systems would require new estimation procedures, as the least-squares estimator is not necessarily optimal in such settings, which could significantly affect how the observations translate to estimation error and therefore to controller loss, introducing non-trivial analysis challenges.
>
> > *“I strongly suggest adding more or complex tasks to demonstrate the efficiency of the proposed methods over non-optimal exploration model-based policy learning.”*
>
> We refer the reviewer to points 1. and 3. of our response to all reviewers. As stated there our contribution is primarily theoretical, but we believe our experimental results do provide empirical evidence that task-driven exploration yields non-trivial gains on realistic systems, and motivates future experimental work in this direction.

---

### Official Review · Reviewer_o9V7 · 2023-07-05

**Soundness:** 4 excellent
**Presentation:** 4 excellent
**Contribution:** 4 excellent
**Rating:** 8
**Confidence:** 3

**Summary:**

The paper proposes an active exploration algorithm for systems of the form $x_{k+1} = A \phi(x_k, u_k) + w_k$ with known features $\phi$ and unknown matrix $A$, where $w_k \sim \mathcal{N}(0,1)$. The problem is reduced to minimising $\mathrm{tr}(\mathcal{H}(A_\star)\check{\Lambda}_T^{-1})$ where $\mathcal{H}(A_\star)$ is the Hessian at the optimal $A_\star$ (which is iteratively estimated) and $\check{\Lambda}_T^{-1}$ is roughly the expected covariance $\phi\phi^T$ over $T$ episodes of length $H$. Thorough analysis is presented. The algorithm achieves instance-optimal rate. Examples on simulated systems are provided.

**Strengths:**

Very strong theoretical paper, solving optimal exploration for a class of linear-in-features systems. Originality, quality, clarity, are excellent. Significance is strong, but since the features are assumed known, it limits the applicability of the algorithm.


**Weaknesses:**

- The class of systems is quite restricted. If one needs to learn the non-linear features $\phi(x, u)$ first, then one could just as well use the collected data to estimate $A$ afterwards. It would be interesting to see an extension where features are learned as well.
- The computational complexity is unclear. Does the algorithm run real-time? Is the numerical implementation stable?
- There is no Discussion/Conclusion/Limitations section.
- Empirical comparisons are only done with respect to very naive baselines (random and uniform exploration). At least some variance or entropy minimising active exploration could be considered, as e.g., in [1]. But for this paper it is fine, the theoretical contributions are already sufficient, no further experiments are necessary.

[1] Schultheis, M., Belousov, B., Abdulsamad, H., & Peters, J. (2020). Receding horizon curiosity. In Conference on robot learning (pp. 1278-1288). PMLR.

**Questions:**

Could one further improve the algorithm for control-affine systems? I.e., $x' = A\phi(x) + Bu$ with unknown $A$ and $B$?

**Limitations:**

No. The authors are encouraged to add at least one paragraph of Conclusion, which also discusses the limitations.

---

> ### Author Rebuttal · Authors · 2023-08-09
>
> We would like to thank the reviewer for their comments and will work to incorporate the feedback into the final version. We address specific comments below.
>
> > *“The class of systems is quite restricted. If one needs to learn the non-linear features  first, then one could just as well use the collected data to estimate  afterwards. It would be interesting to see an extension where features are learned as well.”*
>
> We refer the reviewer to point 2. in our response to all reviewers. We agree that there are interesting cases where one may want to learn the features of the system, but leave fully addressing this for future work. We remark, however, that in many settings (e.g. inverted pendulum) the features (i.e., structure from physics) are known, and all that must be learned is the coefficient. Furthermore, in many settings it may be possible to learn an expressive set of features from one task and transfer them to another; therefore, we do not believe the assumption that the features are known is particularly restrictive (though is nonetheless an interesting direction for future work).
>
> > *“The computational complexity is unclear. Does the algorithm run real-time? Is the numerical implementation stable?”*
>
> We refer the reviewer to point 4. in our response to all reviewers for discussion on computational complexity. The current implementation is essentially real-time—the primary computational burden is in the policy optimization step and computation of the hessian. While these cannot be done in real-time currently, they only need to be performed a limited number of times (on order $\log T$) and can be performed offline between episodes. Furthermore, as noted, our approach is very modular and these more computationally expensive components could be replaced with more computationally efficient procedures if desired—we did not optimize for this. The implementation is numerically stable and we plan to release code for it.
>
> > *“There is no Discussion/Conclusion/Limitations section.”*
>
> We have omitted this due to space constraints but will add for the final version. In particular, we will make sure to highlight some of the limitations mentioned by reviewers here.
>
> > *“Empirical comparisons are only done with respect to very naive baselines (random and uniform exploration). At least some variance or entropy minimising active exploration could be considered, as e.g., in [1]. But for this paper it is fine, the theoretical contributions are already sufficient, no further experiments are necessary.”*
>
> We refer the reviewer to points 1. and 3. of our response to all reviewers. As stated there (and noted by the reviewer) our contribution is primarily theoretical, but we believe our experimental results do provide empirical evidence that task-driven exploration yields non-trivial gains on realistic systems, and motivates future experimental work in this direction. Furthermore, we believe the “Uniform” baseline we compare against is very strong (see point 3. for more explanation).
>
> > *“Could one further improve the algorithm for control-affine systems? I.e., $x' = A \phi(x) + Bu$ with unknown $A$ and $B$?”*
>
> As our result is optimal for any choice of $\phi$, in the control-affine case, where $\phi(x,u) = [\phi(x),u]$, our statistical complexity cannot be improved (since it is already optimal). However, there might exist more computationally efficient implementations in such control-affine settings.

---

> > ### Comment · Reviewer_o9V7 · 2023-08-10
> > **Rebuttal Acknowledged**
> >
> > Thank you for your answers and clarifications

---

### Official Review · Reviewer_oMB3 · 2023-07-06

**Soundness:** 3 good
**Presentation:** 3 good
**Contribution:** 2 fair
**Rating:** 4
**Confidence:** 2

**Summary:**

This paper tackles the challenge of controlling unknown nonlinear dynamical systems in reinforcement learning and control theory. The authors propose a novel algorithm, inspired by recent work in linear systems, focusing on the most critical parameters for learning a low-cost controller on the actual system. This method efficiently explores the system to reduce uncertainty in a specific, task-dependent metric. They demonstrate the algorithm's effectiveness on nonlinear robotic systems and provide a theoretical lower bound showing that their approach learns a controller at a near instance optimal rate. This work could have significant implications for reinforcement learning and control theory.

**Strengths:**

The author presents an algorithm which achieves the instance-optimal rate, with controller loss matching the lower bound on the loss of any sufficiently regular control rule.

**Weaknesses:**

The regularity assumptions in 3.1 should be addressed clearly and the author could give a toy example to illustrate when these assumptions are satisfied.

**Questions:**

The formulation of the nonlinear system in (1.1) seems restricted, can you explain more about the formulation?

**Limitations:**

The simulations are conducted on low dimension systems and the experiment setting should be stated more clearly. Besides, this work faces a high computational burden.

---

> ### Author Rebuttal · Authors · 2023-08-09
>
> We would like to thank the reviewer for their comments and will work to incorporate the feedback into the final version. We address specific comments below.
>
> > *"The regularity assumptions in 3.1 should be addressed clearly and the author could give a toy example to illustrate when these assumptions are satisfied."*
>
> We would like to emphasize that, as noted, the majority of the assumptions in Section 3.1 are standard in the theoretical literature [1,2]. These will all be met for classes of problems such as LQR. We also want to emphasize that, intuitively, such assumptions are needed to learn efficiently. For example, if our smoothness assumptions (Assumptions 4 and 5) are not met, small changes in the system parameters could lead to arbitrarily large changes in the cost of the learned controller. To ensure that a (reasonably) good controller can be learned on an estimate of the system parameters, we must assume that the cost of a controller varies smoothly as the parameters of the system change.
>
> [1] Mania, Horia, Michael I. Jordan, and Benjamin Recht. "Active learning for nonlinear system identification with guarantees." arXiv preprint arXiv:2006.10277 (2020).
>
> [2] Kakade, Sham, et al. "Information theoretic regret bounds for online nonlinear control." Advances in Neural Information Processing Systems 33 (2020): 15312-15325.
>
> > *"The formulation of the nonlinear system in (1.1) seems restricted, can you explain more about the formulation?"*
>
> We refer the reviewer to point 2. of our response to all reviewers. As stated there, system (1.1) is actually very general and can encompass essentially any continuous dynamical system.
>
> > *"The simulations are conducted on low dimension systems and the experiment setting should be stated more clearly. Besides, this work faces a high computational burden."*
>
> We refer the reader to point 1., 3., and 4. of our response to all reviewers, and to Appendix G for additional experimental details. As noted, we believe our contribution is primarily theoretical (with a focus on statistical optimality rather than computational complexity), but we believe our experimental results do provide empirical evidence that task-driven exploration yields non-trivial gains on realistic systems, and motivates future experimental work in this direction. Furthermore, efficient approximations of our algorithm exist, which we believe could be scaled to higher-dimensional systems without issue.

---

### Official Review · Reviewer_KGe1 · 2023-07-26

**Soundness:** 3 good
**Presentation:** 4 excellent
**Contribution:** 3 good
**Rating:** 6
**Confidence:** 2

**Summary:**

The paper seeks to quantify formally in the settings of nonlinear dynamical systems (a) which parameters are most relevant to learning a good controller and (b) the best exploration strategy to minimize uncertainty in such parameters. This paper draws theoretical insights where minimizing the controller loss in nonlinear systems translates to estimating the system parameters in a particular task-dependent metric. In light of this characterization, the paper presents an algorithm that achieves the instance-optimal rate for efficient exploration, with controller loss matching a proven lower bound. Several numerical experiments are conducted on nonlinear systems to validate the analysis and the approach.




**Strengths:**

- **S1** The paper is very well written. The problem definitions and contributions are clear.
- **S2** For a particular class of nonlinear systems of the form (Eqn. 1.1), the paper characterizes how minimizing the controller loss in nonlinear systems translates to estimating the system parameters in a task-dependent metric, and provides a lower bound on the loss of any control rule.
- **S3** The paper proposes an algorithm that achieves the instance-optimal rate, with controller loss matching the proven lower bound.
- **S4** The proposed approah demonstrates improvements compared to the baselines in several numerical experiments.

**Weaknesses:**

- **W1** The paper could benefit from having stronger experimental results. The authors could include more simulated environments that fit into the particular class of nonlinear systems, both in simulation and the real world, such as [1]. The authors could also consider comparing with more baseline methods, such as [2] and [3], from the deep model-based RL community.
- **W2** Could the authors elaborate on how theoretical insights from the paper relate to problems in practical applications of the particular class of nonlinear systems (1.1)? I believe this would improve the paper's impact on the broader robotics, control, and RL communities. For instance, could applications in safety-critical scenarios be benefited from these insights (e.g., flight maneuvers in [1])?

[1] O’Connell et al., "Neural-fly enables rapid learning for agile flight in strong winds." Science Robotics, 7(66):eabm6597, 2022.

[2] Kaiser et al., "Model-based reinforcement learning for atari."  arXiv preprint arXiv:1903.00374, 2019.

[3] Hafner et al., "Dream to Control: Learning Behaviors by Latent Imagination." arXiv preprint arXiv:1912.01603, 2020.

**Questions:**

- The authors mentioned in line 378 that they believe the approach will also scale to deep model-based RL settings. Could the authors explain how their approach can be scaled with deep RL?


**Limitations:**

- The authors target a particular class of nonlinear systems characterized in (1.1). This characterization would determine what kind of systems the theoretical insights and the approach can be directly related to.
- The authors provide experimental analysis on simulated domains. It would be interesting to see the results in real-world environments.

---

> ### Author Rebuttal · Authors · 2023-08-09
>
> We would like to thank the reviewer for their comments and will work to incorporate the feedback into the final version. We address specific comments below.
>
> > *Response to W1*
>
> We refer the reviewer to points 1. and 3. of our response to all reviewers. As stated there our contribution is primarily theoretical, but we believe our experimental results do provide empirical evidence that task-driven exploration yields non-trivial gains on realistic systems, and motivates future experimental work in this direction.
>
> > *Response to W2*
>
> The primary practical insights from our theory are in what and how to explore. Our theoretical analysis provides a precise quantification of which parameters of the system are most relevant to learn if our goal is to find a good policy, and therefore provide an exploration objective: explore so as to minimize the estimation error in these parameters. Furthermore, our analysis provides insight into how to achieve this: running the DynamicOED procedure will ensure that the estimation error in relevant parameters is minimized as much as possible.
>
> The particulars of a given application can be easily encoded in our cost. For example, one could add a penalty to the cost to incentivize safe behavior, or could choose the cost to track a particular trajectory. In such settings, our approach will adapt to direct exploration to whatever parameters are most relevant to the particular cost. In addition, as our exploration policy set is generic, it could be restricted to only include “safe” exploration policies (for example, policies guaranteed to maintain stability of the drone during exploration)---our theoretical results still provide an end-to-end guarantee in such settings. Moreover, the learned features in [1] exactly matches our setting in (1.1), so our theory applies directly to guide practice in such applications.
>
> > *"The authors mentioned in line 378 that they believe the approach will also scale to deep model-based RL settings. Could the authors explain how their approach can be scaled with deep RL?"*
>
> Our approach is very modular—for example, the policy parameterization and optimizer used, as well as the regret minimization oracle for DynamicOED could be replaced without changing our end-to-end guarantee. One could therefore instantiate these with a variety of methods, including deep RL approaches (e.g. PPO with neural network function approximation could be run on the estimated system to obtain a good policy). While these approaches may lack the theoretical guarantees necessary for our formal results to hold, we expect that, if they are used effectively, they could still yield strong empirical results.

---

> > ### Comment · Reviewer_KGe1 · 2023-08-17
> > **Thank you for the clarifications**
> >
> > Thank you for your answers and clarifications. I recommend that you add the practical insights to the paper to improve its presentation.

---

### Official Review · Reviewer_srf1 · 2023-07-28

**Soundness:** 3 good
**Presentation:** 3 good
**Contribution:** 3 good
**Rating:** 7
**Confidence:** 3

**Summary:**


In this paper, the authors propose a method for learning an optimal controller that is able to focus the exploration so as to minimize the estimation error of relevant parameters. They show that the proposed method is indeed optimal with tight upper and lower bounds on the gap. Experiments show that the proposed algorithm performs better compared to the baseline reaching lower loss at any given number of learning episodes.

**After authors' rebuttal:** Thanks for the added clarification. I agree that the paper's contributions are largely theoretical and there's enough here to merit a publication. The authors mention their intention for addition of proof sketches and intuitive discussion which will improve the readability of the paper further. Given these, I am keeping my rating at 7!

**Strengths:**

The paper is clearly-written, well-motivated and systematically-structured. The assumptions are explicit and the theorems are described to develop intuition. While the idea of certain parameters dictating the system dynamics might make abstract sense, having the motivating example in section 1.1 made the follow-up analysis more grounded for me. Using the second order moment to quantify parameter relevance is neat but that appears to be adopted from previous work. In general, the theory part of the paper is crisp and laid out to facilitate understanding.


**Weaknesses:**

On the other hand, the experiment section is quite thin. However, as a largely theoretical contribution, this is acceptable. I didn’t look through the appendix so I missed out on the proofs, and the paper could have benefitted from some proof sketches without hashing out all the details.

**Questions:**

NA

---

> ### Author Rebuttal · Authors · 2023-08-09
>
> We would like to thank the reviewer for their comments and will work to incorporate the feedback into the final version. We address specific comments below.
>
> > *"On the other hand, the experiment section is quite thin. However, as a largely theoretical contribution, this is acceptable."*
>
> We refer the reviewer to points 1. and 3. of our response to all reviewers. As stated there (and as the reviewer and other reviewers also state) our contribution is primarily theoretical, but we believe our experimental results do provide empirical evidence that task-driven exploration yields non-trivial gains on realistic systems, and motivates future experimental work in this direction.
>
> > *"the paper could have benefitted from some proof sketches without hashing out all the details"*
>
> We agree that proof sketches would be helpful and will plan to include in the final version with the extra page of space. In particular, we will include intuition on how the lower bound was proved, and additional intuition on the exploration procedure (DynamicOED).

---

### Author Rebuttal · Authors · 2023-08-09

We would like to thank the reviewers for their feedback and will do our best to incorporate as much of it as possible into the final version. We address several key points of clarification below:

1. **Primary Contributions:** We want to emphasize that the primary contribution of this paper is theoretical. To our knowledge, ours is the first work to provide instance-optimal guarantees for nonlinear control. We also want to emphasize that the extension from linear to nonlinear systems introduces significant technical difficulties (e.g. in linear systems, random excitation allows for efficient learning, which is not the case here), which our work overcomes. Furthermore, the study of instance-optimality in decision making is of much interest to the theory community and is a very active area of research (see e.g. [1,2,3] below and references therein)—we believe our results are an important contribution in this direction.

2. **System Model:** Several reviewers remarked that the class of systems we consider is too restrictive. We have several comments on this. First, as is stated in bullet point 1, our setting can model any continuous dynamics model given expressive enough $\phi$ (which can be accomplished without prior knowledge of the system by, for example, using random fourier features [7]), and is therefore quite general. In particular, our setting encompasses many standard control settings: linear systems, control affine systems, etc. Second, this setting is standard in the theory community, and has seen much recent work [4,5]. Finally, similar settings have been considered in a variety of empirical studies, demonstrating its applicability to a wide range of real-world problems in robotics and model-based RL [6,7,8].

3. **Experiments:** Several reviewers suggested that we provide additional experimental results, or compare against more baselines. We first want to reiterate that our primary contribution is theoretical. Our experiments, then, serve as a proof-of-concept of the theory. They are also, however, the first experimental results, to our knowledge, to demonstrate that task-driven exploration can yield non-trivial improvements over more naive exploration in realistic systems. We believe this motivates future empirical research extending this to higher-dimensional systems, but this is beyond the scope of the current work. We also want to reiterate that our method is agnostic to the choice of policy parameterization and optimizer, and could be combined with many different approaches—our method provides a statistically optimal exploration routine for any given policy parameterization and optimization method. Given this, for the sake of clarity in our experiments we chose to use relatively straightforward control methods, but the extension to more general methods is straightforward. On our choice of baselines, we want to emphasize that the method referred to as “Uniform” (the approach of [4]) is a very strong baseline as it explores in a targeted manner to optimally reduce uncertainty in the estimated model. It is essentially performing a variant of maximum entropy exploration specialized to the class of systems we consider—we would not expect any existing approaches for MBRL (which either rely on random exploration or entropy-driven exploration) to improve on this baseline.

4. **Computational Efficiency:** Several reviewers brought up the computational efficiency of our approach. While our approach is not computationally efficient in general, we want to emphasize it is statistically optimal. In addition, existing work on learning in systems of the form (1.1) are also computationally inefficient as efficient planning in systems of the form (1.1) is in general not tractable. For special cases—such as linear systems—however, where efficient planning is possible, our algorithm is computationally efficient. Furthermore, as we demonstrate experimentally, there exist computationally efficient approximations (e.g., using a Jacobian approximation to the Hessian computation) to our algorithm which allow us to run it on a range of systems.

[1] Wagenmaker, Andrew J., and Dylan J. Foster. "Instance-optimality in interactive decision making: Toward a non-asymptotic theory." The Thirty Sixth Annual Conference on Learning Theory. PMLR, 2023.

[2] Kirschner, Johannes, et al. "Asymptotically optimal information-directed sampling." Conference on Learning Theory. PMLR, 2021.

[3] Xu, Haike, Tengyu Ma, and Simon Du. "Fine-grained gap-dependent bounds for tabular mdps via adaptive multi-step bootstrap." Conference on Learning Theory. PMLR, 2021.

[4] Mania, Horia, Michael I. Jordan, and Benjamin Recht. "Active learning for nonlinear system identification with guarantees." arXiv preprint arXiv:2006.10277 (2020).

[5] Kakade, Sham, et al. "Information theoretic regret bounds for online nonlinear control." Advances in Neural Information Processing Systems 33 (2020): 15312-15325.

[6] Richards, Spencer M., et al. "Adaptive-control-oriented meta-learning for nonlinear systems." arXiv preprint arXiv:2103.04490 (2021).

[7] Boffi, Nicholas M., Stephen Tu, and Jean-Jacques E. Slotine. "Regret bounds for adaptive nonlinear control." Learning for Dynamics and Control. PMLR, 2021.

[8] O’Connell, Michael, et al. "Neural-fly enables rapid learning for agile flight in strong winds." Science Robotics 7.66 (2022): eabm6597.

---

### Decision · Program_Chairs · 2023-09-21

**Decision:**

Accept (spotlight)

**Comment:**

This paper considers the setting of nonlinear dynamical systems and seeks to formally quantify which parameters are most relevant to learning a good controller, and how one can best explore so as to minimize uncertainty in such parameters. The proposed algorithm is supported by theoretical guarantees and shows significant improvement over simple exploration mechanisms in a number of simulated environments. The reviewers agree that the paper is well-written, clear, technically strong, and the algorithm is novel.